# Implicit Bias and Fast Convergence Rates for Self-attention

**Bhavya Vasudeva**$^*$        *bvasudev@usc.edu*
*University of Southern California*

**Puneesh Deora**$^*$        *puneeshdeora@ece.ubc.ca*
*University of British Columbia*

**Christos Thrampoulidis**        *cthrampo@ece.ubc.ca*
*University of British Columbia*

**Reviewed on OpenReview:** *https://openreview.net/forum?id=pKilnjQsbO*

## Abstract

We study the fundamental optimization principles of self-attention, the defining mechanism of transformers, by analyzing the implicit bias of gradient-based optimizers in training a self-attention layer with a linear decoder in binary classification. Building on prior studies in linear logistic regression, recent findings demonstrate that the key-query matrix $\boldsymbol{W}_t$ from gradient-descent (GD) converges in direction towards $\boldsymbol{W}_{\mathrm{mm}}$, which maximizes the margin between optimal and non-optimal tokens across sequences. However, this convergence is local, dependent on initial conditions, only holds asymptotically as the number of iterations increases, and leaves questions about the potential benefits of adaptive step-size rules unaddressed. To bridge this gap, we first establish scenarios for which convergence is provably *global*. We then analyze two adaptive step-size strategies: normalized GD and Polyak step-size, demonstrating *finite-time* convergence rates for $\boldsymbol{W}_t$ to $\boldsymbol{W}_{\mathrm{mm}}$, and quantifying the sparsification rate of the attention map. These findings not only show that these strategies can accelerate parameter convergence over standard GD in a non-convex setting but also deepen the understanding of the implicit bias in self-attention, linking it more closely to the phenomena observed in linear logistic regression despite its intricate non-convex nature.

## 1 Introduction

Self-attention serves as the fundamental building block of transformers, distinguishing them from traditional neural networks (Vaswani et al., 2017) and driving their outstanding performance across various applications, including natural language processing and generation (Devlin et al., 2019; Brown et al., 2020; Raffel et al., 2020), as well as computer vision (Dosovitskiy et al., 2021; Radford et al., 2021; Touvron et al., 2021). With transformers establishing themselves as the de-facto deep-learning architecture, driving advancements in applications seamlessly integrated into society's daily life at an unprecedented pace (OpenAI, 2022), there has been a surge of recent interest in the mathematical study of the fundamental optimization and statistical principles of the self-attention mechanism; see Section 6 on related work for an overview.

In pursuit of this objective, Tarzanagh et al. (2023b;a) have initiated an investigation into the *implicit bias* of gradient descent (GD) in training a self-attention layer with fixed linear decoder in a binary classification task. Concretely, the study paradigm of implicit bias seeks to characterize structural properties of the weights learned by GD when the training objective has multiple solutions. The prototypical instance of this paradigm is GD training of linear logistic regression on separable data: among infinitely many possible solutions to logistic-loss minimization (each linear separator defines one such solution), GD learns weights that converge in direction to the (unique) max-margin class separator (Soudry et al., 2018; Ji & Telgarsky, 2018). Notably, convergence is global, holding irrespective of the initial weights' direction, and comes with explicit rates that

---
$^*$Equal Contribution.

Figure 1: Comparison of train and test dynamics of various optimizers—SGD, stochastic normalized GD (SNGD), stochastic Polyak step (SPS), and Adam—while fine-tuning a pre-trained BERT model on the MNLI dataset; see App. C for details. SNGD and SPS, employing adaptive step-size rules, demonstrate significantly faster training, closely resembling the performance of Adam. Motivated in part by this observation, our work establishes fast convergence rates for NGD and PS for single-layer self-attention.

characterize its speed with respect to the number of iterations. Drawing an analogy to this prototypical instance, when training self-attention with linear decoder in a binary classification task, Tarzanagh et al. (2023a) defines a hard-margin SVM problem (W-SVM) that separates, with maximal margin, *optimal* input tokens from non-optimal ones based on their respective softmax logits. For this, they show that the key-query weights $\boldsymbol{W}_t$ found by GD converge locally to the solution $\boldsymbol{W}_{\mathrm{mm}}$ of (W-SVM) as the number of iterations $t$ grows to infinity.

Despite the intriguing analogy between the two settings, the findings of Tarzanagh et al. (2023a) are highly non-trivial not only because the nature of the max-margin solution differs, but also because of the intricate non-convex optimization landscape introduced by the presence of self-attention. The non-convexity, induced by the softmax operator in the self-attention layer, complicates the analysis and is the reason why the convergence result of Tarzanagh et al. (2023a) is: (i) *local*, holding only for an appropriate initialization direction, and (ii) *asymptotic*, applicable only as iterations $t$ approach infinity.

Identifying these limitations, this work is motivated by the following questions:

Q1: *Are there settings under which GD iterates converge* globally *to the solution $\boldsymbol{W}_{mm}$ of* (W-SVM)?

Q2: *Is it possible to obtain* finite-time *rates of convergence to $\boldsymbol{W}_{mm}$?*

Additionally, motivated by the practical benefit of using adaptive learning rates in transformer optimization (see for example Fig. 1), we pose an additional open question:

Q3: *Can using* adaptive learning rates *in self-attention optimization accelerate the convergence to $\boldsymbol{W}_{mm}$?*

**Contributions.** Our work addresses the above open questions, thus contributing fundamental insights on the optimization properties of the self-attention mechanism.

Concretely, we study a single-layer self-attention model $\Phi(\boldsymbol{X}, \boldsymbol{\theta}) = \boldsymbol{u}^\top \boldsymbol{X}^\top \boldsymbol{\varphi}(\boldsymbol{XW}\boldsymbol{x}_1)$, where, $\boldsymbol{\varphi}(\cdot)$ is the softmax nonlinearity, $\boldsymbol{X} = [\boldsymbol{x}_1, \cdots, \boldsymbol{x}_T]$ is the sequence of input tokens, $\boldsymbol{W}$ is the key-query matrix, and $\boldsymbol{u}$ is a linear decoder. Following the setup of Tarzanagh et al. (2023a), for each sequence $\boldsymbol{X}$, we associate each token $\boldsymbol{x}_\tau$ with a score $\gamma_\tau := y\boldsymbol{u}^\top \boldsymbol{x}_\tau$ and let $\mathtt{opt} \in [\mathrm{T}]$ be the index of the token with the largest score. Given a training set of $n$ sequences $\boldsymbol{X}_i$, this defines a hard-margin SVM problem with solution $\boldsymbol{W}_{\mathrm{mm}}$ that separates the optimal tokens with maximum margin; see Section 2 for details.

Motivated by question Q3, we study here the optimization properties of training $\Phi(\boldsymbol{X}; \boldsymbol{\theta})$ with exponential loss using *normalized GD*, which sets the learning rate $\eta_t$ at iteration $t$ adaptively as $\eta_t = \frac{\eta}{\|\nabla_{\boldsymbol{\theta}} \widehat{L}(\boldsymbol{\theta}_t)\|}$, for some constant $\eta > 0$. Our results also extend to Polyak step-size, another adaptive step-size rule.

Our first set of results (Section 3), for fixed decoder $\boldsymbol{u}$ (similar to Tarzanagh et al. (2023a)), answer questions Q1-Q3 as follows. In response to Q1, we begin by identifying sufficient conditions on the input data under which GD converges *globally*, in direction, to $\boldsymbol{W}_{\mathrm{mm}}$. Then, simultaneously addressing Q2 and Q3, we establish fast finite-time rates of convergence for normalized GD by proving that the iterates $\boldsymbol{W}_t$, at any time $t$, satisfy $\left\| \frac{\boldsymbol{W}_t}{\|\boldsymbol{W}_t\|} - \frac{\boldsymbol{W}_{\mathrm{mm}}}{\|\boldsymbol{W}_{\mathrm{mm}}\|} \right\| \leq \mathcal{O}\left(t^{-1/2}\right)$. We identify two key ingredients towards establishing these results. First, we

show that the Euclidean norm $\|\boldsymbol{W}_t\|$ of the iterates grows at a rate $\Theta(t)$. Second, and more intricate, we demonstrate that even if the iterate at any time $t$ violates the constraints of (W-SVM) corresponding to any training sample, the softmax score of the optimal token for that sample must be non-decreasing. In turn, this establishes that the iterates, initialized in any direction, remain inside a cone around $\boldsymbol{W}_{\mathrm{mm}}$. Our convergence results also imply an explicit rate at which softmax-attention gets sparsified as softmax scores of optimal tokens converge to 1 at an exponential rate $\mathcal{O}(\exp(-\eta t))$.

Our second set of results (Section 4), raises the limitation of fixed linear decoder of prior work and applies to joint training of $\boldsymbol{u}$ and of the attention weights $\boldsymbol{W}$. In response to Q1, we construct a representative data model with Gaussian-distributed tokens and prove that GD converges *globally*, in direction, to $\boldsymbol{W}_{\mathrm{mm}}$. To address Q2, we show that for normalized GD with an aggressive step-size, the iterates $\boldsymbol{W}_t$, at any time $t$, satisfy $\left\| \frac{\boldsymbol{W}_t}{\|\boldsymbol{W}_t\|} - \frac{\boldsymbol{W}_{\mathrm{mm}}}{\|\boldsymbol{W}_{\mathrm{mm}}\|} \right\| \leq \mathcal{O}\left(1/\log t\right).^*$ Further, we prove that the linear decoder $\boldsymbol{u}$ converges, in direction, to $\boldsymbol{u}_{\mathrm{mm}}$, the solution of the hard-margin SVM problem (u-SVM) that separates the examples with maximal margin, using only the *optimal* input tokens at a $\mathcal{O}(t^{-\eta})$ rate. Finally, to completely characterize the training dynamics, we show fast train loss convergence at a $\mathcal{O}(\exp(-t^{1/3}))$ rate. We highlight three key technical contributions towards proving these results: (i) a growing token score gap $\gamma_{\mathtt{opt},t} - \gamma_{\tau,t} > 0$, between optimal $\mathtt{opt}$ and non-optimal $\tau \neq \mathtt{opt}$ tokens, for any time $t > 0$; (ii) non-decreasing softmax scores of the optimal tokens with time; and (iii) a constant loss ratio across all training sequences valid for any time. These properties pave way for a PL-like inequality which is crucial to show the loss convergence rates.

Throughout, we validate our findings on both synthetic and real datasets (see Section 5). We end with Section 7 discussing the implications of our results and the open questions they raise.

## 2 Preliminaries

Lowercase and uppercase bold letters represent vectors and matrices, respectively. $\|\boldsymbol{a}\|$ and $\|\boldsymbol{A}\|$ denote Euclidean norms. $\bar{\boldsymbol{a}}$ and $\bar{\boldsymbol{A}}$ denote $\frac{\boldsymbol{a}}{\|\boldsymbol{a}\|}$ and $\frac{\boldsymbol{A}}{\|\boldsymbol{A}\|}$, respectively. $[\boldsymbol{a}_1, \boldsymbol{a}_2] = \{\boldsymbol{a} : \boldsymbol{a} = \beta \boldsymbol{a}_1 + (1-\beta)\boldsymbol{a}_2, \beta \in [0,1]\}$ denotes the line segment between $\boldsymbol{a}_1$ and $\boldsymbol{a}_2$, and $\mathrm{concat}(\cdot, \cdot)$ denotes the concatenation operation. $a \wedge b$ denotes the minimum of numbers $a$ and $b$. $a \vee b$ denotes their maximum. We use standard notation $\mathcal{O}, \Omega$ to hide absolute constants, and $\widetilde{\mathcal{O}}, \widetilde{\Omega}$ to hide poly-logarithmic factors. All logarithms are natural logarithms.

The output of a single-head self-attention layer, parameterized by key, query and value matrices $\boldsymbol{W}_Q, \boldsymbol{W}_K \in \mathbb{R}^{d \times d_1}$, $\boldsymbol{W}_V \in \mathbb{R}^{d \times d_2}$, is given by $\boldsymbol{\varphi}(\boldsymbol{X}\boldsymbol{W}_Q\boldsymbol{W}_K^\top\boldsymbol{X}^\top)\boldsymbol{X}\boldsymbol{W}_V$, where $\boldsymbol{X} = [\boldsymbol{x}_1, \cdots, \boldsymbol{x}_\mathrm{T}] \in \mathbb{R}^{\mathrm{T} \times d}$ is the input sequence, and the softmax map $\boldsymbol{\varphi}(\cdot) : \mathbb{R}^\mathrm{T} \to \mathbb{R}^\mathrm{T}$ is applied row-wise. We can compose the output of the attention layer with a linear projection head to obtain the final prediction as

$$\Phi(\boldsymbol{X}; \boldsymbol{\theta}) \coloneqq \boldsymbol{u}^\top\boldsymbol{X}^\top\boldsymbol{\varphi}(\boldsymbol{X}\boldsymbol{W}\boldsymbol{x}_1), \tag{1}$$

where $\boldsymbol{\theta} \coloneqq \mathrm{concat}(\boldsymbol{u}, \boldsymbol{W})$ denotes the set of trainable parameters. Here, similar to Tarzanagh et al. (2023a); Deora et al. (2023); Tian et al. (2023a); Oymak et al. (2023) we reparameterize the key-query matrix as $\boldsymbol{W} \coloneqq \boldsymbol{W}_Q\boldsymbol{W}_K^\top \in \mathbb{R}^{d \times d}$, use the first token for prediction$^\dagger$ and subsume the value weights $\boldsymbol{W}_V$ within the prediction head $\boldsymbol{u} \in \mathbb{R}^d$. Let $\boldsymbol{a}(\boldsymbol{W}) \coloneqq \boldsymbol{X}\boldsymbol{W}\boldsymbol{x}_1$ denote the vector of softmax logits, then $\boldsymbol{\varphi}'(\boldsymbol{a}(\boldsymbol{W})) \in \mathbb{R}^{\mathrm{T} \times \mathrm{T}}$ denotes the Jacobian at $\boldsymbol{a}(\boldsymbol{W})$.

Given training data $\{(\boldsymbol{X}_i, y_i)\}_{i=1}^n$ with labels $y_i \in \{\pm 1\}$ and $\boldsymbol{X}_i \coloneqq [\boldsymbol{x}_{i,1}, \boldsymbol{x}_{i,2}, \ldots, \boldsymbol{x}_{i,\mathrm{T}}]$, and decreasing loss $\ell : \mathbb{R} \to \mathbb{R}_+$, the empirical risk is $\widehat{L}(\boldsymbol{\theta}) \coloneqq \frac{1}{n}\sum_{i \in [n]} \ell(y_i \Phi(\boldsymbol{X}_i; \boldsymbol{\theta}))$. We focus on GD optimization with adaptive time-dependent step-size $\eta_t$. Concretely, we study two variants: normalized gradient descent (NGD) (Hazan et al., 2015; Nacson et al., 2019), and Polyak step-size (PS) (Loizou et al., 2021), which set

$$\eta_t \propto 1/\|\nabla_{\boldsymbol{\theta}}\widehat{L}(\boldsymbol{\theta}_t)\|, \quad \text{and} \quad \eta_t = \left(\widehat{L}(\boldsymbol{\theta}_t) - \widehat{L}^*\right)/\left(2\|\nabla_{\boldsymbol{\theta}}\widehat{L}(\boldsymbol{\theta}_t)\|^2\right) \text{ with } \widehat{L}^* = \min_{\boldsymbol{\theta}} \widehat{L}(\boldsymbol{\theta}),$$

respectively. Our results are applicable to both update rules. For specificity, we present them for NGD and include remarks for PS.

---

$^*$Note the rate here is slower compared to the fixed-decoder case. As detailed in Section 4, this is due to the additional $(t+1)^{-1}$ factor in the step-size, which results in a slow down of the rate of growth of $\|\boldsymbol{W}_t\|$. Our proof requires this additional factor to account for the non-smooth objective.

$^\dagger$This is without loss of generality as our results hold for any token $\tau \in [\mathrm{T}]$.

We follow Tarzanagh et al. (2023b;a) in defining token scores as follows.

**Definition 1** (Token scores and Optimality). *Given a fixed prediction head $\boldsymbol{u}_* \in \mathbb{R}^d$, the token score vector for a sample $(y, \boldsymbol{X})$ is given by $\boldsymbol{\gamma} = y\boldsymbol{X}\boldsymbol{u}_*$. The optimal token index[‡] is $\boldsymbol{opt} = \arg\max_{\tau \in [\mathrm{T}]} \gamma_\tau$, where $\gamma_\tau = y\boldsymbol{x}_\tau^\top \boldsymbol{u}_*$ denotes the token score for the token $\boldsymbol{x}_\tau$.*

Similarly, $\mathtt{opt}_i$ denotes the optimal token index for a sample $(y_i, \boldsymbol{X}_i)$, $i \in [n]$. Intuitively, these are the tokens that minimize the training loss upon selection (Lemma 2 in Tarzanagh et al. (2023a)). Given a set of optimal token indices $\mathtt{OPT} := \{\mathtt{opt}_i\}_{i=1}^n$, define the following hard-margin SVM problem, which separates, with maximal margin, optimal tokens from the other tokens for every input sequence:

$$\boldsymbol{W}_{\mathrm{mm}} = \underset{\boldsymbol{W}}{\arg\min} \|\boldsymbol{W}\| \quad \text{s.t.} \ (\boldsymbol{x}_{i,\mathtt{opt}_i} - \boldsymbol{x}_{i,\tau})^\top \boldsymbol{W} \boldsymbol{x}_{i,1} \geq 1 \ \forall i \in [n], \tau \in [\mathrm{T}] \setminus \{\mathtt{opt}_i\}. \tag{W-SVM}$$

Throughout, we assume that (W-SVM) is feasible,[§] *i.e.* softmax logits $\boldsymbol{x}_{i,\mathtt{opt}_i}^\top \boldsymbol{W} \boldsymbol{x}_{i,1}$ of optimal tokens can be separated from logits $\boldsymbol{x}_{i,\tau}^\top \boldsymbol{W} \boldsymbol{x}_{i,1}$ of the other tokens $\tau \neq \mathtt{opt}_i$.

# 3 Training Dynamics of $W$

Here, let fixed prediction head $\boldsymbol{u} = \boldsymbol{u}_*$. This allows focusing first on the dynamics of token selection induced by training key-query weight matrix $\boldsymbol{W}$, which is the only trainable parameter:

$$\boldsymbol{W}_{t+1} = \boldsymbol{W}_t - \eta \frac{\nabla_{\boldsymbol{W}} \widehat{L}(\boldsymbol{W}_t)}{\|\nabla_{\boldsymbol{W}} \widehat{L}(\boldsymbol{W}_t)\|}, \ t > 0. \tag{2}$$

We start by setting up (mild) assumptions on the data and initialization, and then present our main results.

## 3.1 Setup

For convenience, first define $\Lambda := \|\boldsymbol{W}_{\mathrm{mm}}\|$, $B := \max_{i,\tau} \|\boldsymbol{x}_{i,\tau}\|$, $\gamma_i := \min_{\tau \neq \mathtt{opt}_i} \gamma_{i,\tau}$, $\kappa_+ := \max_i \exp(\gamma_{i,\mathtt{opt}_i} - \gamma_i)$, $\kappa_- := \min_i \exp(\gamma_{i,\mathtt{opt}_i} - \gamma_i)$, and $\Upsilon = \kappa_+ \cdot \frac{\log(\kappa_+)}{\log(\kappa_-)}$. Note $1/\Lambda$ is the margin achieved by $\boldsymbol{W}_{\mathrm{mm}}$ separating optimal tokens from the rest, and $B$ is a uniform bound on the tokens' magnitudes. The parameters $\kappa_\pm$ represent the largest and smallest degradation factors across sequences for each sequence's individual loss term when suboptimal tokens are selected. Respectively, $\Upsilon$ can be interpreted as a conditioning parameter for the problem, measuring the variability in token gaps across different sequences.

Our first technical assumption towards ensuring global convergence requires that tokens are nearly orthogonal.[¶] This is often the case in high-dimensional settings; see Example 1 and the references below.

**Assumption 1** (Nearly-orthogonal Tokens). *For any $i, j, k \in [n]$, and for any $\tau, \tau' \in [\mathrm{T}]$,*

$$\|\boldsymbol{x}_{i,\tau}\|^2 \geq 4n\Upsilon\mathrm{T}|\langle \boldsymbol{x}_{i,\tau}, \boldsymbol{x}_{j,\tau'} \rangle|, \quad j \neq i, \tau' \neq \boldsymbol{opt}_j,$$

$$\langle \boldsymbol{x}_{i,\boldsymbol{opt}_i}, \boldsymbol{x}_{j,\boldsymbol{opt}_j} \rangle \geq 4n\Upsilon\mathrm{T}|\langle \boldsymbol{x}_{i,\tau}, \boldsymbol{x}_{k,\tau'} \rangle|, \quad y_i = y_j \neq y_k.$$

We will use Ass. 1 to prove that softmax scores of optimal tokens are lower bounded by a constant throughout the optimization trajectory. Note that Ass. 1 itself makes *no* further guarantees on softmax scores of optimal tokens approaching 1 or even increasing during training, which is essential for global convergence and we prove separately. Furthermore, we impose no assumptions on the direction of initialization $\boldsymbol{W}_0$. In particular, our main results hold even when $\boldsymbol{W}_0$ is aligned with "bad" stationary directions $\overline{\boldsymbol{W}}$ that could asymptotically saturate non-optimal tokens, i.e. $\nabla_{\boldsymbol{W}} \widehat{L}(\alpha\overline{\boldsymbol{W}}) \to \boldsymbol{0}$ and $\varphi_{i,\mathtt{opt}_i}(\alpha\overline{\boldsymbol{W}}) \to 0$ as $\alpha \to \infty$. Thus, the only mild requirement on the initialization $\boldsymbol{W}_0$ regards its scale, rather than its direction, and is formalized in Lem. 2 in the App. See Fig. 8 in the App. for numerical validation of global convergence despite initializing in a "bad" stationary direction. The following example illustrates the above assumption.

---

[‡] Similar to Tarzanagh et al. (2023a), we assume unique optimal token which holds for almost all datasets.

[§] This is analogous to linear separability assumptions in linear models, which are fundamental to understand the implicit bias in GD-based methods. For linear models, feasibility of hard-margin SVM is guaranteed when the ambient dimension exceeds the number of samples. Similarly, for our self-attention model, (W-SVM) feasibility is guaranteed when the ambient dimension exceeds both the number of samples and the context window size (as shown in Theorem 1 of Tarzanagh et al. (2023a)).

[¶] Our results also extend to cases where OPT tokens are antipodal and the remaining tokens are nearly orthogonal.

**Example 1.** *Let $\boldsymbol{\mu}_\pm \in \mathbb{R}^d$, $\boldsymbol{\mu}_+ \perp \boldsymbol{\mu}_-$, $\|\boldsymbol{\mu}_+\| = \|\boldsymbol{\mu}_-\| = U$, and data generated as follows:*

$$y \sim \mathrm{Unif}(\{\pm 1\}), \quad \boldsymbol{opt} \sim \mathrm{Unif}([\mathrm{T}]), \quad \boldsymbol{\nu} \sim \mathcal{N}(\mathbf{0}, \sigma^2 \boldsymbol{\Sigma}),$$

$$\boldsymbol{x}_{\boldsymbol{opt}} = \boldsymbol{\nu} + \begin{cases} \boldsymbol{\mu}_+ & , y = 1 \\ \boldsymbol{\mu}_- & , y = -1 \end{cases}, \quad \boldsymbol{x}_\tau \sim \mathcal{N}(\mathbf{0}, \rho^2 \boldsymbol{\Sigma}), \, \forall \tau \in [\mathrm{T}] \smallsetminus \boldsymbol{opt},$$

*where $\boldsymbol{\Sigma} := \boldsymbol{I}_d - U^{-2}\boldsymbol{\mu}_+\boldsymbol{\mu}_+^\top - U^{-2}\boldsymbol{\mu}_-\boldsymbol{\mu}_-^\top$. It is easy to show that when $d = \widetilde{\Omega}(n^2\mathrm{T}^2)$, the tokens are nearly orthogonal with high probability. Further let high enough signal-to-noise ratio $U^2 \geq \widetilde{\Omega}((\sigma \vee \rho)^2 n\mathrm{T}\sqrt{d})$ and appropriate $\boldsymbol{u}_*$ such that the token scores satisfy $\Upsilon = \mathcal{O}(1)$, then, the data model satisfies Ass. 1 (see Lem. 6 in the App. for details). We remark that similar data models have been considered in prior works on the analysis of linear overparameterized models (Muthukumar et al., 2019; Chatterji & Long, 2021; Wang & Thrampoulidis, 2022; Cao & Gu, 2019; Wang et al., 2021), NNs with Leaky ReLU activation (Frei et al., 2022a; 2023), CNNs (Cao et al., 2022; Kou et al., 2023a), and self-attention models (Deora et al., 2023; Li et al., 2023a).*

Our second technical assumption is similar to Tarzanagh et al. (2023a) with two key distinctions: Firstly, it applies to self-attention rather than their simplified attention model. Secondly, and most importantly, under this same assumption, we will prove a stronger results for global convergence and finite-time rates.

**Assumption 2.** *For any $i \in [n]$, and any $\tau \neq \boldsymbol{opt}_i \in [\mathrm{T}]$, the token scores satisfy $\gamma_{i,\tau} = \gamma_i$.*

Note that in Ex. 1, Ass. 2 is satisfied by choosing an appropriate $\boldsymbol{u}_*$ (see Lem. 6 in the App. for details). It has been recently shown that the optimization landscape of self-attention can lead GD to converge to local directions that are different from $\boldsymbol{W}_{\mathrm{mm}}$ (Tarzanagh et al., 2023a). Thus, global convergence necessitates additional conditions. Assumptions 1 and 2, as outlined above, serve this purpose. While being only sufficient conditions, they lead to the first-known global convergence result with finite-time guarantees, contrasting with the local and asymptotic convergence in Tarzanagh et al. (2023a).

## 3.2 Main Results

We now present our main results. For ease of exposition, we use exponential loss, but our results directly extend to continuously differentiable decreasing loss functions. First, we establish the rate of NGD's directional convergence to $\overline{\boldsymbol{W}}_{\mathrm{mm}}$. A proof sketch is given in Sec. 3.4.

**Theorem 1** (IB rate). *Under small initialization scale (Lem. 2 in the App.) and Ass. 1-2, using the NGD updates in Eq. (2) with $\eta = \widetilde{\mathcal{O}}(B^{-2})$, it holds for any $t \geq t_0 = \mathrm{poly}\left(\eta, B, \Lambda, \mathrm{T}, n\Upsilon\right)$,*

$$\langle \overline{\boldsymbol{W}}_t, \overline{\boldsymbol{W}}_{mm} \rangle \geq 1 - C\,\frac{(\log t)^2}{t},$$

*where $C := C(\eta, B, \Lambda, t_0) = \mathrm{poly}\left(\eta, B, \Lambda, t_0\right)$. In particular, assuming $B = \mathcal{O}(1), T = \mathcal{O}(1)$ and $\Upsilon = \mathcal{O}(1)$, we have $\Lambda = \mathcal{O}(1)$. Thus, for any $t \geq t_0 = \Omega(n)$, it holds $\left\|\overline{\boldsymbol{W}}_t - \overline{\boldsymbol{W}}_{mm}\right\| \leq \tilde{\mathcal{O}}\left(t^{-1/2}\right)$.*

Thm. 1 establishes that normalized iterates $\overline{\boldsymbol{W}}_t$ of NGD converge *globally* to $\overline{\boldsymbol{W}}_{\mathrm{mm}}$, starting from *any* initialization direction. It also provides a lower bound on the rate of $\overline{\boldsymbol{W}}_t$ approaching $\overline{\boldsymbol{W}}_{\mathrm{mm}}$. In Sec. 3.3, we establish an upper bound for the rate of convergence of GD, demonstrating that NGD's adaptive step-size provably accelerates convergence. This is the first global convergence result and finite-time convergence rate for self-attention using either GD or NGD. To the best of our knowledge, this is also the first demonstration of directional convergence of NGD and its superiority over GD in a non-convex setting (Sec. 3.3 for comparisons with existing results in convex linear settings).

We now discuss an implication of Thm. 1 concerning the evolution of the attention map during NGD training. Specifically, we demonstrate that as training progresses, the attention map increasingly becomes sparse, selecting optimal tokens at an exponential rate.

**Lemma 1** (Softmax score rate). *Under the setting of Thm. 1, it holds for any $i \in [n]$ and any $t \geq \mathrm{poly}\left(\eta, B, \Lambda, t_0\right)$ that $\varphi_{i,\boldsymbol{opt}_i}^t \geq \left(1 + (T-1)e^{-\eta(8B^2\Lambda^2)^{-1}t}\right)^{-1}$.*

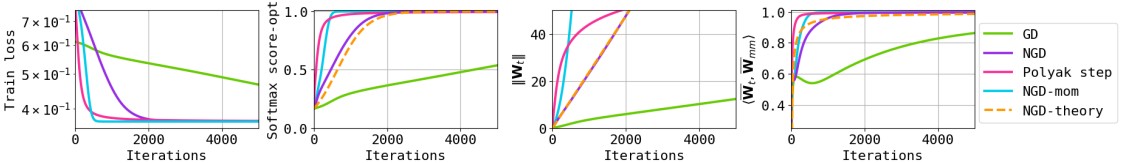

Figure 2: Training dynamics of a single-head self-attention model (Eq. (1)) when optimizing only $\boldsymbol{W}$ on synthetic data with nearly orthogonal tokens (Example 1 with $\sigma = 0$). The observed softmax score saturation, norm growth of $\boldsymbol{W}$ and directional alignment with $\boldsymbol{W}_{\mathrm{mm}}$ closely match with our theoretical results (Lemma 1, Eq. (9) and Theorem 1, respectively).

To understand why the attention map becomes sparser during NGD training, recall from Thm. 1 that for sufficiently large iterations, $\boldsymbol{W}_t \approx \frac{\|\boldsymbol{W}_t\|}{\|\boldsymbol{W}_{\mathrm{mm}}\|} \boldsymbol{W}_{\mathrm{mm}}$, and that $\boldsymbol{W}_{\mathrm{mm}}$ separates optimal tokens from others. In proving Thm. 1, we also show a norm-growth rate $\|\boldsymbol{W}_t\| = \Theta(t)$. Combining these yields that softmax scores of optimal tokens $\varphi_{i,\mathtt{opt}_i}^t$ approach 1, while the rest approach 0. Concretely, given $\boldsymbol{W}_t \approx \frac{\|\boldsymbol{W}_t\|}{\|\boldsymbol{W}_{\mathrm{mm}}\|} \boldsymbol{W}_{\mathrm{mm}}$, the $\mathtt{opt}$ softmax score $\varphi_{i,\mathtt{opt}_i}^t \geq \frac{1}{1 + \sum\limits_{\tau \neq \mathtt{opt}_i} \exp(-\frac{\|\boldsymbol{W}_t\|}{\|\boldsymbol{W}_{\mathrm{mm}}\|})}$ approaches 1 as $\|\boldsymbol{W}_t\|$ grows. Lem. 1 formalizes this intuition and pairs it with an explicit sparsification rate.

Finally, we validate the predictions of the above results using synthetic data generated as per Ex. 1 with $\sigma = 0$ (see App. C for details). In Fig. 2, we plot the train loss, the norm growth of $\boldsymbol{W}_t$, the softmax score $\varphi_{i,\mathtt{opt}_i}$ for the $\mathtt{opt}$ token (averaged over $i \in [n]$), and the alignment of $\boldsymbol{W}_t$ with $\boldsymbol{W}_{\mathrm{mm}}$. Note that since $\boldsymbol{u}_*$ is fixed, the train loss does not converge to 0. Observe that the directional alignment for NGD is closely predicted by Thm. 1. Similarly, the softmax-score saturation rate in Lem. 1 closely aligns with the empirical rate in Fig. 2. Moreover, note that NGD is significantly faster than standard GD, which we formally prove below.

### 3.3 Remarks

**NGD is faster than GD.** We demonstrate that there exists a dataset, satisfying the assumptions of Thm. 1, for which the correlation of vanilla GD parameters to $\boldsymbol{W}_{\mathrm{mm}}$, $\langle \overline{\boldsymbol{W}}_t, \overline{\boldsymbol{W}}_{\mathrm{mm}} \rangle$, is upper bounded by $\leq 1 - \Omega((\log t)^{-2})$. In contrast, according to Thm. 1, the rate for NGD is $\geq 1 - \mathcal{O}((\log t)^2/t)$, thereby highlighting a significant gap between the two.

To construct the dataset, let $n = 1$, T $= 2$, $y = 1$, $\boldsymbol{x}_1 = [1,0]^\top$, $\boldsymbol{x}_2 = [0,0]^\top$ and $\boldsymbol{u}_* = [1,0]^\top$. Clearly, Ass. 2 holds since there is only two tokens; specifically, $\mathtt{opt} = 1$, $\gamma_{\mathtt{opt}} - \gamma = 1$. Also, Ass. 1 holds trivially since $n = 1$. For this dataset, the max-margin classifier can be easily expressed in closed form as $\boldsymbol{W}_{\mathrm{mm}} = (\boldsymbol{x}_1 - \boldsymbol{x}_2)\boldsymbol{x}_1^\top$. Using this, we observe that $\nabla_{\boldsymbol{W}} \widehat{L}(\boldsymbol{W}_t) = -\widehat{L}(\boldsymbol{W}_t)\varphi_{\mathtt{opt}}^t(1 - \varphi_{\mathtt{opt}}^t)\boldsymbol{W}_{\mathrm{mm}}$, which implies

$$\boldsymbol{W}_{t+1} = \boldsymbol{W}_0 + \eta G_t \boldsymbol{W}_{\mathrm{mm}}, \quad \text{where we set } G_t := \sum_{t'=0}^{t} \|\nabla_{\boldsymbol{W}}\widehat{L}(\boldsymbol{W}_{t'})\|.$$

Leveraging this, we now show that $\|\boldsymbol{W}_t\| \leq \mathcal{O}(\log t)$: Note that $\|\boldsymbol{W}_t\| \to \infty$ as $t \to \infty$ which leads to softmax saturation $\varphi_{\mathtt{opt}}^t \to 1$ and hence, loss minimization. This implies that $G_t$ grows with $t$, and there exists some $t_0 > 0$, such that for any $t \geq t_0$, it holds that $\|\boldsymbol{W}_{t+1}\| \approx \eta G_t + W_0^{11}$, where $W_0^{11}$ is the $(1,1)$ entry of the initialization matrix. Using this, we can show for $t \geq t_0$ that $1 - \varphi_{\mathtt{opt}}^{t+1} = \frac{1}{1 + \exp((\boldsymbol{x}_1 - \boldsymbol{x}_2)^\top \boldsymbol{W}_{t+1}\boldsymbol{x}_1)} \leq \exp(-(\boldsymbol{x}_1 - \boldsymbol{x}_2)^\top \boldsymbol{W}_{t+1}\boldsymbol{x}_1) = \exp(-\eta G_t - W_0^{11}) \approx \exp(-\|\boldsymbol{W}_{t+1}\|)$. Thus,

$$\|\boldsymbol{W}_{t+1}\| \leq \|\boldsymbol{W}_0\| + \eta \sum_{t'=0}^{t} \widehat{L}(\boldsymbol{W}_{t'})\exp(-\|\boldsymbol{W}_{t'}\|) \leq \mathcal{O}(\sum_{t'=0}^{t} \exp(-\|\boldsymbol{W}_{t'}\|)),$$

where we use $\widehat{L}(\boldsymbol{W}_t) \leq \mathcal{O}(1)$, since the decoder is fixed. This proves that $\|\boldsymbol{W}_t\| \leq \mathcal{O}(\log t)$, which we use to upper bound the correlation as follows:

$$\langle \overline{\boldsymbol{W}}_t, \overline{\boldsymbol{W}}_{\mathrm{mm}} \rangle = \frac{\eta G_t + W_0^{11}}{\sqrt{(\eta G_t + W_0^{11})^2 + \|\boldsymbol{W}_0\|^2 - (W_0^{11})^2}} \approx 1 - \frac{\|\boldsymbol{W}_0\|^2 - (W_0^{11})^2}{2(\eta G_t + W_0^{11})^2} \approx 1 - \frac{\|\boldsymbol{W}_0\|^2 - (W_0^{11})^2}{2\|\boldsymbol{W}_t\|^2}. \quad (3)$$

Thus, $\langle \overline{\boldsymbol{W}}_t, \overline{\boldsymbol{W}}_{\mathrm{mm}} \rangle \leq 1 - \Omega((\log t)^{-2})$, or equivalently $\|\overline{\boldsymbol{W}}_t - \overline{\boldsymbol{W}}_{\mathrm{mm}}\| \geq \Omega((\log t)^{-1})$. This shows that using an adaptive step-size accelerates convergence to $\boldsymbol{W}_{\mathrm{mm}}$, compared to standard GD.

**Extension to Polyak step-size.** In Thm. 7 in the App., we establish an analogue of Thm. 1 specifically for the Polyak step-size rule. This extension required us to further establish a lower bound on the gradient norm. This is because PS has an additional $\|\nabla_{\boldsymbol{\theta}} \widehat{L}(\boldsymbol{\theta}_t)\|^{-1}$ factor compared to NGD. To accomplish this, we used the variational form of the Euclidean norm for a "good" direction $\boldsymbol{W}_{\mathrm{mm}}$, yielding $\|\nabla_{\boldsymbol{\theta}} \widehat{L}(\boldsymbol{\theta}_t)\| \geq \mathrm{poly}\left(\Lambda^{-1}\right) \widehat{L}(\boldsymbol{\theta}_t)$.

As seen in Fig. 2, PS leads to even faster convergence than NGD across all metrics, including the train loss. In our implementation, we use a hyperparameter $\eta_{\max}$ as a threshold on the step-size value, which has been shown to stabilize training (Loizou et al., 2021). Further, since the selection of opt tokens minimizes the loss, $\widehat{L}^*$ is calculated by setting $\varphi_{i,\mathrm{opt}_i} = 1$ for all $i \in [n]$. Note that after some time, when $\eta_t \geq \eta_{\max}$, the PS updates used in practice are identical to standard GD with step-size $\eta_{\max}$. Hence, the rates are much faster in the beginning, but slow down as training progresses.

### 3.4 Proof Sketch of Theorem 1

The first key intermediate result we need to show is that after sufficient iterations, the negative gradient of the loss at $\boldsymbol{W}_t$ is more correlated with $\boldsymbol{W}_{\mathrm{mm}}$ compared to $\boldsymbol{W}_t$ itself. Formally, we show that for all $\epsilon > 0$, there exists $R_\epsilon := 2\Lambda^{-1} \log(C\epsilon^{-1})$ with $C = \mathrm{poly}\left(\mathrm{T}, B^2\Lambda, n\Upsilon\right)$, such that for every $t$ for which $\|\boldsymbol{W}_t\| \geq R_\epsilon$, it holds that

$$\left\langle -\nabla_{\boldsymbol{W}} \widehat{L}(\boldsymbol{W}_t), \overline{\boldsymbol{W}}_{\mathrm{mm}} \right\rangle \geq (1-\epsilon) \left\langle -\nabla_{\boldsymbol{W}} \widehat{L}(\boldsymbol{W}_t), \overline{\boldsymbol{W}}_t \right\rangle. \tag{4}$$

A proof sketch of this follows; see Lem. 7 in the App. for details. First, under Ass. 2, for any $\boldsymbol{W}$:

$$\left\langle -\nabla_{\boldsymbol{W}} \widehat{L}(\boldsymbol{W}_t), \boldsymbol{W} \right\rangle = \frac{1}{n} \sum_{i=1}^n \ell_{t,i} (\gamma_{i,\mathrm{opt}_i} - \gamma_i) \varphi_{i,\mathrm{opt}_i}^t (1 - \varphi_{i,\mathrm{opt}_i}^t) h_{t,i}, \tag{5}$$

where $h_{t,i} = a_{i,\mathrm{opt}_i} - \frac{\sum_{\tau \neq \mathrm{opt}_i} \varphi_{i,\tau}^t a_{i,\tau}}{\sum_{\tau \neq \mathrm{opt}_i} \varphi_{i,\tau}^t}$, $\boldsymbol{a}_i = \boldsymbol{X}_i \boldsymbol{W} \boldsymbol{x}_{i,1}$. Analogously, define $\tilde{h}, h^*$, using $\tilde{\boldsymbol{a}} = \Lambda \boldsymbol{X} \overline{\boldsymbol{W}}_t \boldsymbol{x}_1$, $\boldsymbol{a}^* = \boldsymbol{X} \boldsymbol{W}_{\mathrm{mm}} \boldsymbol{x}_1$, respectively. Then, to prove Eq. (4), it suffices that

$$\tilde{h}_{t,i} \leq (1+\nu) h_{t,i}^*, \quad \text{where } \nu = \epsilon(1-\epsilon)^{-1}. \tag{6}$$

Suppose that $\|\overline{\boldsymbol{W}}_t - \overline{\boldsymbol{W}}_{\mathrm{mm}}\| > \frac{\nu}{2B^2\Lambda}$, since otherwise the claim follows easily. We categorize the training samples into three subsets based on the satisfaction of the constraints in (W-SVM). Concretely, define the "minimum SVM-gap per sample" $i \in [n]$ as $\delta_i^{\min} := \min_{\tau \notin \{\mathrm{opt}_i\}} \tilde{a}_{i,\mathrm{opt}_i} - \tilde{a}_{i,\tau} - 1$ and based on this consider three sets of samples as follows:

- $\mathcal{I}_1 := \{i \in [n] \mid \delta_i^{\min} \leq 0.3\nu\}$, for samples where the constraints are violated or weakly satisfied,

- $\mathcal{I}_2 := \{i \in [n] \mid 0.3\nu < \delta_i^{\min} \leq 0.8\nu\}$, for samples where constraints are satisfied with small gap,

- $\mathcal{I}_3 := \{i \in [n] \mid \delta_i^{\min} \geq 0.8\nu\}$, for samples where the constraints are well-satisfied.

Since $\|\overline{\boldsymbol{W}}_t - \overline{\boldsymbol{W}}_{\mathrm{mm}}\|$ is large, there is at least one point $i \in [n]$ that violates the constraints, i.e. $\mathcal{I}_1$ is non-empty. We first show the following for examples in $\mathcal{I}_1$ and $\mathcal{I}_2$

$$\tilde{h}_{t,i} \leq (1 + 0.5\nu) h_{t,i}^*, \ i \in \mathcal{I}_1, \quad \text{and} \quad \tilde{h}_{t,i} \leq (1+\nu) h_{t,i}^*, \ i \in \mathcal{I}_2. \tag{7}$$

Note that we prove a tighter inequality for $\mathcal{I}_1$ than required by (6) in order to use this residual to get the desired inequality for examples in $\mathcal{I}_3$. Specifically, when $R_\epsilon$ is large enough, we wish to show (following Eq. (5)) that

$$\sum_{i \in \mathcal{I}_3} \ell_{t,i} (\gamma_{i,\mathrm{opt}_i} - \gamma_i) \varphi_{i,\mathrm{opt}_i}^t (1 - \varphi_{i,\mathrm{opt}_i}^t) \leq \frac{0.5\nu}{2B^2\Lambda} \sum_{i \in \mathcal{I}_1} \ell_{t,i} (\gamma_{i,\mathrm{opt}_i} - \gamma_i) \varphi_{i,\mathrm{opt}_i}^t (1 - \varphi_{i,\mathrm{opt}_i}^t). \tag{8}$$

Intuitively, for the LHS, any sample $i \in \mathcal{I}_3$ satisfies (W-SVM) constraints with large gap giving $\varphi_{i,\mathtt{opt}_i}$ close to 1 and growing with $\|\boldsymbol{W}_t\|$. Similarly, for the RHS, we expect $\varphi_{i,\mathtt{opt}_i}$ to be smaller. Thus, for a sufficiently large $\|\boldsymbol{W}_t\|$ and controlling $\varphi_{i,\mathtt{opt}_i}$ for any $i \in \mathcal{I}_1$ and $t > 0$ would allow us to show the above inequality. This is achieved by establishing a non-trivial lower bound on the softmax scores for $i \in \mathcal{I}_1$ that holds throughout training (Lem. 3 in the App.). Finally, since $h^* \geq 1$ and $\tilde{h} \leq 2B^2\Lambda$, we combine Eqs. (7) and (8) using the formulation in (5) and complete the proof of Eq. (4).

The next key ingredient in the proof of Thm. 1 is showing that the iterate norm grows at rate $\Theta(t)$. Specifically, we prove in Lem. 5 in the App. that

$$2\eta t \geq \|\boldsymbol{W}_t\| \geq \eta(4B^2\Lambda)^{-1}t. \tag{9}$$

The upper bound follows by applying Cauchy-Schwarz. For the lower bound, we write the iterate $\boldsymbol{W}_t$ using its gradient updates, use the dual norm characterization of the $\ell_2$-norm, and select the $\boldsymbol{W}_{\mathrm{mm}}$ direction to get

$$\|\boldsymbol{W}_t\| \geq \left\langle \boldsymbol{W}_0, \overline{\boldsymbol{W}}_{\mathrm{mm}} \right\rangle + \sum_{t'=0}^{t-1} \eta \left\langle -\frac{\nabla_{\boldsymbol{W}}\widehat{L}(\boldsymbol{W}_t)}{\|\nabla_{\boldsymbol{W}}\widehat{L}(\boldsymbol{W}_t)\|}, \frac{\boldsymbol{W}_{\mathrm{mm}}}{\|\boldsymbol{W}_{\mathrm{mm}}\|} \right\rangle,$$

where we use the key property that for any $\boldsymbol{W}$ the gradient correlates positively with $\boldsymbol{W}_{\mathrm{mm}}$ (formalized in Lem. 4 in the App.), i.e. $\left\langle -\frac{\nabla_{\boldsymbol{W}}\widehat{L}(\boldsymbol{W})}{\|\nabla_{\boldsymbol{W}}\widehat{L}(\boldsymbol{W})\|}, \frac{\boldsymbol{W}_{\mathrm{mm}}}{\|\boldsymbol{W}_{\mathrm{mm}}\|} \right\rangle \geq (2B^2\Lambda)^{-1} > 0$.

Next, using Eq. (2) to substitute the gradient update in Eq. (4), we can show that

$$\left\langle \boldsymbol{W}_{t+1} - \boldsymbol{W}_t, \frac{\boldsymbol{W}_{\mathrm{mm}}}{\|\boldsymbol{W}_{\mathrm{mm}}\|} \right\rangle \geq (1-\epsilon_t) \left\langle \boldsymbol{W}_{t+1} - \boldsymbol{W}_t, \frac{\boldsymbol{W}_t}{\|\boldsymbol{W}_t\|} \right\rangle = \frac{\|\boldsymbol{W}_{t+1}\|^2 - \|\boldsymbol{W}_t\|^2 - \|\boldsymbol{W}_{t+1} - \boldsymbol{W}_t\|^2}{2\|\boldsymbol{W}_t\|} - \epsilon_t \left\langle \boldsymbol{W}_{t+1} - \boldsymbol{W}_t, \frac{\boldsymbol{W}_t}{\|\boldsymbol{W}_t\|} \right\rangle$$

$$\geq \|\boldsymbol{W}_{t+1}\| - \|\boldsymbol{W}_t\| - \frac{\eta^2}{2\|\boldsymbol{W}_t\|} - \epsilon_t\eta.$$

We then select $\epsilon_t = (80B^2\Lambda^2 \log t)(\eta t)^{-1}$, and use Eq. (9) and Lem. 7 to obtain a $\mathcal{O}((\log t)^2\|\boldsymbol{W}_t\|^{-1})$ rate. Using Eq. (9) then finishes the proof.

## 4 Training Dynamics for Joint Optimization

We now study training dynamics when jointly optimizing prediction head $\boldsymbol{u}$ and attention weights $\boldsymbol{W}$. Compared to Sec. 3, the key challenge is that the token scores $\boldsymbol{\gamma}^t$ evolve with time, driven by the changing nature of $\boldsymbol{u}_t$. Additionally, the objective function becomes non-smooth in this context, given the dynamic changes in $\boldsymbol{u}_t$. Addressing these challenges necessitates additional technical considerations compared to Sec. 3, which we also discuss in this section.

### 4.1 Setup

We consider the following updates for $\boldsymbol{u}_t$ and $\boldsymbol{W}_t$.

$$\boldsymbol{u}_{t+1} = \boldsymbol{u}_t - \eta_t^u \nabla_{\boldsymbol{u}}\widehat{L}(\boldsymbol{\theta}_t), \quad \eta_t^u = \frac{\eta(t+1)^{-2/3}}{\|\nabla_{\boldsymbol{u}}\widehat{L}(\boldsymbol{\theta}_t)\|}, \quad \boldsymbol{W}_{t+1} = \boldsymbol{W}_t - \eta_t^W \nabla_{\boldsymbol{W}}\widehat{L}(\boldsymbol{\theta}_t), \quad \eta_t^W = \frac{\eta(t+1)^{-1}}{\|\nabla_{\boldsymbol{W}}\widehat{L}(\boldsymbol{\theta}_t)\|}. \tag{10}$$

Here, we focus on exponential loss. Note that in $\eta_t^u$, the factor $(t+1)^{-2/3}$ can be replaced with $(t+1)^{-p}$, where $p \in [2/3, 1)$. In our results, we present findings using $p = 2/3$, as it yields the fastest rate. We also remark that common adaptive learning rates like Adam (Kingma & Ba, 2014), AdaGrad (Duchi et al., 2011), etc. similarly set per-parameter learning rates that vary dynamically.

Similar to Def. 1, given a trainable prediction head $\boldsymbol{u}_t$ at any iteration $t > 0$, the token score vector for a sample $(y, \boldsymbol{X})$ is given by $\boldsymbol{\gamma}^t = y\boldsymbol{X}\boldsymbol{u}_t$.

**Data Model.** We study the following data model. Within each example $\boldsymbol{X}$, a single $\mathtt{opt} \sim \mathrm{Unif}([T])$ token and an additional $T-1$ tokens are sampled i.i.d. Gaussian as follows:

$$\boldsymbol{x}_{\mathtt{opt}} \sim \mathcal{N}(\boldsymbol{0}, \alpha^2\rho^2\boldsymbol{I}_d), \quad \boldsymbol{x}_\tau \sim \mathcal{N}(\boldsymbol{0}, \rho^2\boldsymbol{I}_d), \ \forall \tau \in \mathcal{S}, \quad y = \mathtt{sign}(\boldsymbol{u}_*^\top\boldsymbol{x}_{\mathtt{opt}}), \tag{DM}$$

where $\mathcal{S}:=[\mathrm{T}]\smallsetminus\{\mathtt{opt}\}$, $\boldsymbol{u}_* \in \mathbb{R}^d$ is a fixed vector that generates the label $y \in \{\pm 1\}$. Here, $\rho$ controls the token norms, and $\alpha > 1$ separates the $\mathtt{opt}$ token from the other tokens. We consider the following mild conditions on the initialization, token norms, overparameterization and step-size.

**Condition 1.** *Let $B = \alpha\rho\sqrt{1.5d} \geq 1$, and $C_0 > 1$, $C_1, C_2 > 0$ be some absolute constants. We consider the following conditions for any $\delta \in (0,1)$: i) Zero initialization: $\|\boldsymbol{u}_0\| = 0, \|\boldsymbol{W}_0\| = 0$; ii) opt token has larger norm: $\alpha \geq C_0$; iii) Sufficient overparameterization: $d \geq C_1\alpha^4 n^4 \log\left(\frac{10n^2}{\delta}\right)$; iv) Small step-size: $\eta \leq (C_2\sqrt{n}B^4(B \vee d))^{-1}$.*

The condition on the initialization is for ease of exposition; our analysis and results extend to random initialization with small scale (similar to the conditions of Lem. 2 in Sec. 3). We also remark that under Cond. 1 and data model DM, the tokens are nearly orthogonal, similar to Ass. 1 in Sec. 3. We formalize this in Lem. 9 in the Appendix.

## 4.2 Main Results

We now present our main results for joint training. For ease of exposition, we consider $\mathrm{T}=2$, but our results directly extend to $\mathrm{T}>2$ assuming token scores $\boldsymbol{\gamma}^t$ satisfy Ass. 2 at every iteration $t$. It is also possible to extend our results to the case where Ass. 2 is not exactly satisfied, but the non-$\mathtt{opt}$ token scores show small deviation.

Our first result shows that train loss goes to 0 at $\mathcal{O}(\exp(-t^{1/3}))$ rate; see Sec. 4.3 for a proof sketch.

**Theorem 2** (Train loss convergence)**.** *Under Cond. 1, data model DM, using the updates in Eq. (10), it holds for any $t>0$ that $\widehat{L}(\boldsymbol{\theta}_{t+1}) \leq \mathcal{O}\left(\exp\left(-\frac{\eta BC_0}{\sqrt{n}}(t+1)^{1/3}\right)\right)$.*

The next theorem illustrates that $\boldsymbol{W}_t$ converges to $\boldsymbol{W}_{\mathrm{mm}}$ in direction at an $\mathcal{O}(1/\log t)$ rate.

**Theorem 3** (IB rate of $\boldsymbol{W}$)**.** *Under Cond. 1 and the data model DM, using the updates in Eq. (10), for any $t \geq t_\epsilon = \exp(\mathrm{poly}\left(\eta, B, \Lambda, n, d, \log \delta^{-1}, \epsilon^{-1}\right))$,*

$$\left\langle \overline{\boldsymbol{W}}_t, \overline{\boldsymbol{W}}_{mm} \right\rangle \geq 1 - \epsilon - \mathrm{poly}\left(\eta, B, \Lambda, \epsilon\right)\frac{1}{\log t}.$$

In particular, assuming $\rho = \mathcal{O}\left(\eta^{1/2}d^{-3/4}(n\Lambda)^{-1}\right)$, for any $t \geq t_\epsilon = \Omega(\exp(\epsilon^{-4/3}))$, $\left\|\overline{\boldsymbol{W}}_t - \overline{\boldsymbol{W}}_{\mathrm{mm}}\right\| \leq \mathcal{O}\left(\epsilon \vee (\log t)^{-1}\right)$. Intuitively, since the updates for $\boldsymbol{W}_t$ have an additional $(t+1)^{-1}$ factor due to the smaller step-size $\eta_t$, the iterate norm $\|\boldsymbol{W}_t\|$ grows as $\Theta(\log t)$ in contrast to the $\Theta(t)$ rate in Sec. 3, where we were only optimizing $\boldsymbol{W}_t$. Consequently, the convergence to implicit bias is slower — $\mathcal{O}(1/\log t)$ instead of $\tilde{\mathcal{O}}(t^{-1})$. The additional $(t+1)^{-1}$ factor in the updates for $\boldsymbol{W}_t$ helps to account for the non-smoothness of the objective, which will become apparent in Sec. 4.3.

We now discuss the implicit bias and convergence of $\boldsymbol{u}_t$. From prior work (Soudry et al., 2018; Ji & Telgarsky, 2019), one could guess that $\boldsymbol{u}_t$ converges to the max-margin predictor separating the set of samples $\{(\varphi_{i,\mathtt{opt}_t}^t \boldsymbol{x}_{i,\mathtt{opt}_i} + (1 - \varphi_{i,\mathtt{opt}_i}^t)\boldsymbol{x}_{i,\tau}, y_i)_{i=1}^n\}$. But as shown above, when $t \to \infty$, $\varphi_{\mathtt{opt}} \to 1$. This motivates the following hard-margin SVM problem,

$$\boldsymbol{u}_{\mathrm{mm}} = \underset{\boldsymbol{u}}{\mathrm{argmin}}\|\boldsymbol{u}\| \text{ s.t. } y_i\boldsymbol{u}^\top\boldsymbol{x}_{i,\mathtt{opt}_i} \geq 1 \ \forall i \in [n]. \tag{u-SVM}$$

The following result confirms that the learned model attains margin that is asymptotically a constant factor $(1/4)$ of the maximum margin $\overline{\gamma} := \max_{\boldsymbol{u}:\|\boldsymbol{u}\|\leq 1} \min_i y_i\boldsymbol{u}^\top\boldsymbol{x}_{i,\mathtt{opt}_i}$.

**Theorem 4** (IB rate of $\boldsymbol{u}$)**.** *Under Cond. 1 and the data model DM, using the updates in Eq. (10), for any $t \geq t_\epsilon \vee \exp(C(\eta, B, \Lambda, \epsilon)(\epsilon^{-1} \vee (8B^2\Lambda)^4))$, it holds that*

$$\min_i y_i\overline{\boldsymbol{u}_t}^\top\boldsymbol{x}_{i,opt_i} \geq \frac{\overline{\gamma}}{4} - \frac{1}{1 + \exp(\eta(8B^2\Lambda^2)^{-1}\log t)}.$$

## 4.3 Proof Sketch of Theorem 2

The main challenge in establishing the loss convergence rate in Thm. 2 is the non-smoothness of the objective function. To overcome this, we show three key properties that hold throughout training: i) constant loss ratio

for any two samples, ii) growing token score gap between `opt` and non-`opt` tokens for every sample, and iii) a non-decreasing `opt`-token softmax score. We formalize these in Lem. 11 in the App. Property (ii) is crucial in proving the implicit bias rate in Thm. 3. Property (iii) is crucial in obtaining a PL-like inequality which is the key challenging step in proving Thm. 2. Specifically, using properties (i) and (iii) we demonstrate that the training loss satisfies the following (see Lem. 13 and Rem. 3 in the App. for details):

1. PL-inequality-like result: $\|\nabla_{\boldsymbol{u}}\widehat{L}(\boldsymbol{\theta}_t)\| \geq \Omega(Bn^{-1/2})\widehat{L}(\boldsymbol{\theta}_t)$,

2. controlled loss between current and next iterate: $\max_{\boldsymbol{\theta}'\in[\boldsymbol{\theta}_t,\boldsymbol{\theta}_{t+1}]}\widehat{L}(\boldsymbol{\theta}') \leq 8\widehat{L}(\boldsymbol{\theta}_t)$,

3. second-order self-boundedness: $\max_{\boldsymbol{\theta}'\in[\boldsymbol{\theta}_t,\boldsymbol{\theta}_{t+1}]}\|\nabla_{\boldsymbol{\theta}}^2\widehat{L}(\boldsymbol{\theta}')\| \leq 8(\omega(\boldsymbol{\theta}_t)\vee\omega(\boldsymbol{\theta}_{t+1}))\widehat{L}(\boldsymbol{\theta}_t)$,

where $\omega(\boldsymbol{\theta}_t) \coloneqq 13B^5(B\vee d)(\|\boldsymbol{u}_t\|\vee 1)^2$.

To prove the first point, we use the dual norm characterization of the $\ell_2$-norm,

$$\|\nabla_{\boldsymbol{u}}\widehat{L}(\boldsymbol{\theta}_t)\| = \sup_{\boldsymbol{v}:\|\boldsymbol{v}\|=1}\left\langle\frac{1}{n}\sum_{i=1}^n|\ell_{i,t}'|y_i\nabla_{\boldsymbol{u}}\Phi(\boldsymbol{u}_t,\boldsymbol{X}_i),\boldsymbol{v}\right\rangle \geq \widehat{L}(\boldsymbol{\theta}_t)\sup_{\boldsymbol{v}:\|\boldsymbol{v}\|=1}\min_i y_i(\varphi_{i,\mathtt{opt}_i}^t\boldsymbol{x}_{i,\mathtt{opt}_i} + (1-\varphi_{i,\mathtt{opt}_i}^t)\boldsymbol{x}_{i,\tau})^\top\boldsymbol{v}.$$

To proceed, we lower bound the softmax scores for the `opt` tokens for every $i \in [n]$ and $t > 0$ as

$$\varphi_{i,\mathtt{opt}_i}^t \geq \tfrac{1}{2}. \tag{11}$$

Using this, we show that the set of samples $\{(0.5(\boldsymbol{x}_{i,\mathtt{opt}_i} + \boldsymbol{x}_{i,\tau}), y_i)\}_{i=1}^n$, where $\tau \neq \mathtt{opt}_i$, are separable using $\tilde{\boldsymbol{u}} = \sum_{i\in[n]}y_i\boldsymbol{x}_{i,\mathtt{opt}_i}$ (formalized in Lem. 12 in the App.). Then, selecting $\boldsymbol{v} = \tilde{\boldsymbol{u}}$ proves the first point.

We now briefly discuss the process to prove Eq. (11). We can show this by induction. Since $\boldsymbol{W}_0 = \boldsymbol{0}$, then for any $i \in [n]$, $\varphi_{i,\mathtt{opt}_i}^0 = 1/2$ at initialization. To show the inductive step, note that for any $i \in [n]$, if

$$(\boldsymbol{x}_{i,\mathtt{opt}_i} - \boldsymbol{x}_{i,\tau})^\top(-\nabla_{\boldsymbol{W}}\widehat{L}(\boldsymbol{\theta}_t))\boldsymbol{x}_{i,1} > 0, \tag{12}$$

$$\implies \frac{\varphi_{j,\mathtt{opt}_j}^{t+1}/\varphi_{j,\tau}^{t+1}}{\varphi_{j,\mathtt{opt}_j}^t/\varphi_{j,\tau}^t} = \exp\left((\boldsymbol{x}_{j,\mathtt{opt}_j} - \boldsymbol{x}_{j,\tau})^\top(\boldsymbol{W}_{t+1} - \boldsymbol{W}_t)\boldsymbol{x}_{j,1}\right) \geq 1.$$

We consider two cases. If $\varphi_{i,\mathtt{opt}_i}^t \geq 3/4$, Eq. (12) follows as $\eta$ is small. Specifically, for any $j \in [n]$, $\tau \neq \mathtt{opt}_j$,

$$\varphi_{j,\mathtt{opt}_j}^t = \frac{1}{1 + \exp((\boldsymbol{x}_{i,\tau} - \boldsymbol{x}_{j,\mathtt{opt}_j})^\top\boldsymbol{W}_t\boldsymbol{x}_{j,1})} > 3/4 \implies (\boldsymbol{x}_{j,\tau} - \boldsymbol{x}_{j,\mathtt{opt}_j})^\top\boldsymbol{W}_{t-1}\boldsymbol{x}_{j,1} \leq -\log(3).$$

After the gradient step, since $\eta \leq (2B^2)^{-1}\log(3)$, we have

$$(\boldsymbol{x}_{j,\tau} - \boldsymbol{x}_{j,\mathtt{opt}_j})^\top\boldsymbol{W}_{t+1}\boldsymbol{x}_{j,1} = (\boldsymbol{x}_{j,\tau} - \boldsymbol{x}_{j,\mathtt{opt}_j})^\top\boldsymbol{W}_t\boldsymbol{x}_{j,1} - \eta_{t-1}(\boldsymbol{x}_{j,\mathtt{opt}_j} - \boldsymbol{x}_{j,\tau})^\top(-\nabla_{\boldsymbol{W}}\widehat{L}(\boldsymbol{W}_t))\boldsymbol{x}_{j,1} \leq 2\eta B^2 - \log(3) \leq 0.$$

If $\varphi_{i,\mathtt{opt}_i}^t \leq 3/4$, we proceed as follows. Consider the expansion of the LHS of Eq. (12),

$$(\boldsymbol{x}_{i,\mathtt{opt}_i} - \boldsymbol{x}_{i,\tau})^\top(-\nabla_{\boldsymbol{W}}\widehat{L}(\boldsymbol{\theta}_t))\boldsymbol{x}_{i,1} = \frac{\ell_{t,i}}{n}(\gamma_{i,\mathtt{opt}_i}^t - \gamma_i^t)\varphi_{i,\mathtt{opt}_i}^t(1-\varphi_{i,\mathtt{opt}_i}^t)\|\boldsymbol{x}_{i,1}\|^2\|\boldsymbol{x}_{i,\mathtt{opt}_i} - \boldsymbol{x}_{i,\tau}\|^2$$
$$- \frac{1}{n}\sum_{j\neq i}\ell_{t,j}(\gamma_{j,\mathtt{opt}_j}^t - \gamma_j^t)\varphi_{j,\mathtt{opt}_j}^t(1-\varphi_{j,\mathtt{opt}_j}^t)\boldsymbol{x}_{i,1}^\top\boldsymbol{x}_{j,1}(\boldsymbol{x}_{j,\mathtt{opt}_j} - \boldsymbol{x}_{j,\tau})^\top(\boldsymbol{x}_{i,\mathtt{opt}_i} - \boldsymbol{x}_{i,\tau'}).$$

To use this to prove Eq. (12), we first show that the token scores $\boldsymbol{\gamma}^t$ satisfy Defn. 1 ($\gamma_{\mathtt{opt}_i}^t > \gamma_i^t$ for any $i \in [n]$), and that the loss ratio $\max_{i,j\in[n]}\frac{\ell_{t,i}}{\ell_{t,j}}$ is bounded by a constant throughout training, *i.e.*, for any $i \in [n]$,

$$\gamma_{i,\mathtt{opt}_i}^t - \gamma_i^t \geq \Omega(t^{1/3}) \quad \text{and} \quad \max_{i,j\in[n]}\frac{\ell_{t,i}}{\ell_{t,j}} \leq C, \tag{13}$$

where $C > 0$ is a universal constant. This is formalized in Lem. 11 in the App. We then prove Eqs. (11) and (13) jointly by induction.

For the second point, we first show that for any $\boldsymbol{\theta}, \boldsymbol{\theta}'$,

$$
\begin{aligned}
|y\Phi(\boldsymbol{\theta}, \boldsymbol{X}) - y\Phi(\boldsymbol{\theta}', \boldsymbol{X})| &= |\boldsymbol{u}^\top \boldsymbol{X}^\top \boldsymbol{\varphi}(\boldsymbol{X}\boldsymbol{W}^\top \boldsymbol{x}_1) - \boldsymbol{u}'^\top \boldsymbol{X}^\top \boldsymbol{\varphi}(\boldsymbol{X}\boldsymbol{W}'^\top \boldsymbol{x}_1)| \\
&\leq \|\boldsymbol{X}\|_{2,\infty} \|\boldsymbol{u} - \boldsymbol{u}'\| + 2\|\boldsymbol{X}\|_{2,\infty}^3 \|\boldsymbol{u}'\| \|\boldsymbol{W} - \boldsymbol{W}'\|.
\end{aligned}
$$

Since we are using exponential loss, we use this to get,

$$
\max_{\lambda \in [0,1]} \frac{\ell(y\Phi(\boldsymbol{\theta}_t + \lambda(\boldsymbol{\theta}_{t+1} - \boldsymbol{\theta}_t), \boldsymbol{X}))}{\ell(y\Phi(\boldsymbol{\theta}_t, \boldsymbol{X}))} \leq \max_{\lambda \in [0,1]} \exp\left(2\lambda \|\boldsymbol{X}\|_{2,\infty}^3 \|\boldsymbol{u}_t\| \|\boldsymbol{W}_{t+1} - \boldsymbol{W}_t\| + \lambda \|\boldsymbol{X}\|_{2,\infty} \|\boldsymbol{u}_{t+1} - \boldsymbol{u}_t\|\right).
$$

Further, we show that the iterate norm $\|\boldsymbol{u}_t\|$ grows as $\mathcal{O}(t^{1/3})$ (formalized in Lem. 10 in the App.). Then, using the updates in Eq. (10), we get

$$
\max_{i \in [n]} \max_{\lambda \in [0,1]} \frac{\ell(y_i \Phi(\boldsymbol{\theta}_t + \lambda(\boldsymbol{\theta}_{t+1} - \boldsymbol{\theta}_t), \boldsymbol{X}_i))}{\ell(y_i \Phi(\boldsymbol{\theta}_t, \boldsymbol{X}_i))} \leq \exp\left(6\eta^2 B^3 + \eta B\right).
$$

Then, using Cond. 1 for $\eta$ finishes the proof for the second point. For the third point, we first upper bound $\|\nabla_{\boldsymbol{\theta}}^2 \Phi\|$ as $\|\nabla_{\boldsymbol{\theta}}^2 \Phi(\boldsymbol{\theta}, \boldsymbol{X})\| \leq 6dB^5 \|\boldsymbol{u}\| + 2\sqrt{d}B^3$. Using this and the upper bound on $\|\nabla_{\boldsymbol{\theta}} \Phi\|$, we show that for any $\boldsymbol{\theta}' \in [\boldsymbol{\theta}_t, \boldsymbol{\theta}_{t+1}]$,

$$
\|\nabla_{\boldsymbol{\theta}}^2 \widehat{L}(\boldsymbol{\theta}')\| \leq \max_{i \in [n]} (\|\nabla_{\boldsymbol{\theta}} \Phi(\boldsymbol{\theta}_t, \boldsymbol{X}_i)\|^2 + \|\nabla_{\boldsymbol{\theta}}^2 \Phi(\boldsymbol{\theta}_t, \boldsymbol{X}_i)\|) \widehat{L}(\boldsymbol{\theta}') \leq \omega(\boldsymbol{\theta}') \widehat{L}(\boldsymbol{\theta}').
$$

Then, we leverage the second point to finish proving the third point.

With these ingredients, we prove Thm. 2 by working with the second-order Taylor expansion of $\widehat{L}(\boldsymbol{\theta}_{t+1})$. Specifically, using the updates in Eq. (10) and the three key properties of the training loss established above, we show that

$$
\widehat{L}(\boldsymbol{\theta}_{t+1}) \leq \widehat{L}(\boldsymbol{\theta}_t) - \frac{\eta B}{10\sqrt{n}(t+1)^{2/3}} \widehat{L}(\boldsymbol{\theta}_t) + \frac{8\eta^2}{(t+1)^{4/3}} (\omega(\boldsymbol{\theta}_t) \vee \omega(\boldsymbol{\theta}_{t+1})) \widehat{L}(\boldsymbol{\theta}_t).
$$

Further, since $\|\boldsymbol{u}_t\| \leq \mathcal{O}(t^{1/3})$, we show that $\omega(\boldsymbol{\theta}_t) \leq \mathcal{O}(t^{2/3})$. Using a small enough $\eta$, and telescoping, gives the advertised result.

## 5 Experimental Results

To complement our theory, we present experiments on synthetic/ real-world data demonstrating that (S)NGD-based training leads to faster convergence for various metrics compared to vanilla (S)GD.

**Synthetic Data.** We first consider the self-attention model in Eq. (1) and data generated as per data model DM. Fig. 3 shows the train loss, iterate norms $\|\boldsymbol{u}_t\|$ and $\|\boldsymbol{W}_t\|$, average softmax score for the opt tokens, and alignment of iterates $\boldsymbol{u}_t$ and $\boldsymbol{W}_t$ to the respective max-margin solutions $\boldsymbol{W}_{\mathrm{mm}}$ and $\boldsymbol{u}_{\mathrm{mm}}$. We compare standard GD, NGD updates in Eq. (10) (without the additional $t^{-2/3}, t^{-1}$ factors in the step-size), PS updates, and joint NGD updates. We observe that all other algorithms are faster than GD, with PS being the fastest. We also see that the train loss converges at a similar rate for NGD and joint-NGD, while for the other metrics, NGD is slightly faster than joint-NGD.

**Language and Vision Datasets.** Fig. 1 compares the train and test dynamics of various optimizers while fine-tuning a pre-trained BERT (Devlin et al., 2019) model on the MNLI (Williams et al., 2018) dataset. As evident, both gradient normalized step-sizes SNGD and SPS train significantly faster than SGD. Fig. 4 in the App. shows similar results for the CivilComments dataset (Borkan et al., 2019). In App. C we present additional results on vision datasets. Interestingly, in this setting, while SPS still outperforms SGD, SNGD trains slower, which could be explored further in future work. It is also worth noting that in this setting we see no significant gap between Adam and SGD; this is consistent with the observations reported in Xie et al. (2023); Kunstner et al. (2023).

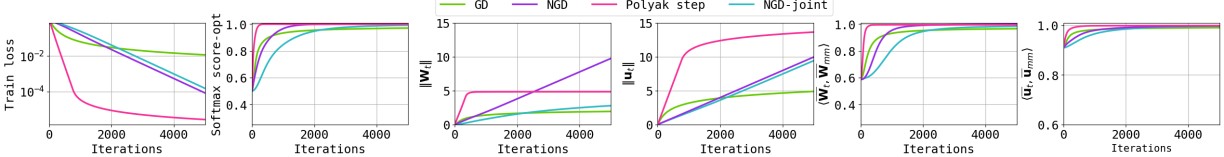

Figure 3: Training dynamics of a self-attention model (Eq. (1)) with data generated using model DM.

## 6   Related Work

**Implicit bias of NNs.** There has been extensive work on the implicit bias of GD for both linear predictors (see Sec. 1) and NNs. Despite non-convexity, our results are more closely related to those for linear predictors since we give an explicit formulation of the solution in the convergence limit (compare to implicit formulations in terms of KKT points for nonlinear NNs; see App. D for a review).

Thm. 1 provides a non-trivial extension to self-attention of corresponding results for linear logistic regression. Specifically, Ji & Telgarsky (2021) demonstrate that NGD on logistic loss under separable data converges globally, in direction, at rate $\widetilde{\mathcal{O}}(1/t)$ to the max-margin separator of data points belonging to distinct classes. While the convergence rate of Thm. 1 for self-attention is slower, it holds under a more intricate non-convex landscape induced by the softmax nonlinearity; see App. A.4 for a discussion on the tightness of our results. As we have seen, in addition to NGD, our convergence results also apply to Polyak step-size, which empirically outperforms NGD (see Fig. 2). We also present results for NGD with momentum in Fig. 2, which results in a performance boost. In the simpler setting of linear logistic regression, this method is proven to have a faster implicit bias rate of $\widetilde{\mathcal{O}}(1/t^2)$ (Ji et al., 2021). Extending our theory to incorporate momentum in the self-attention setting is an interesting direction for future work. The loss convergence rate in Thm. 2 is analogous to the $\mathcal{O}(\exp(-t))$ loss rate for linear predictors on separable data with $\eta_t = \eta/\|\nabla \widehat{L}(\boldsymbol{\theta}_t)\|$ (Ji & Telgarsky, 2019). The difference $\mathcal{O}(\exp(-t^{1/3}))$ vs. $\mathcal{O}(\exp(-t))$ shows up as we use smaller step-size. Nacson et al. (2019) similarly show a slower rate of $\mathcal{O}(\exp(-t^{1/2}))$ for linear predictors with $\eta_t = \eta(t+1)^{-1/2}/\|\nabla \widehat{L}(\boldsymbol{\theta}_t)\|$. Contrary to all these works, to the best of our knowledge, we are the first to prove parameter convergence of NGD and PS in non-convex settings. See App. D for a discussion.

**Transformers theory.** To understand the optimization and generalization dynamics, (Jelassi et al., 2022) shows that Vision Transformers (ViTs) learn spatially localized patterns in a binary classification task using gradient-based methods, while Li et al. (2023b) shows that attention maps sparsify as SGD training progresses. Oymak et al. (2023) studies the initial trajectory of GD for the closely related prompt-attention. Additionally, Tian et al. (2023a) studies SGD-dynamics for the next-token prediction task in a one-layer transformer with a linear decoder. More recently, Tian et al. (2023b) extends this analysis to the joint training of multi-layer transformer with an MLP. As detailed in Sec. 1, our work is most closely related and improves upon Tarzanagh et al. (2023b;a), which adopt an implicit-bias view to attention training. After completion of this work, we became aware of contemporaneous work (Sheen et al., 2024) that also studies implicit bias in the same problem setting (single-layer self-attention, fixed decoder, Ass. 2). Unlike ours, their results hold only for gradient flow (i.e. GD with infinitesimal step-size), and are asymptotic. On the other hand, they study separate optimization of key-query matrices, albeit with appropriate initialization assumptions. See App. D for a full review.

## 7   Conclusion and Future Work

We studied implicit optimization bias of GD for self-attention in binary classification. For both fixed and trainable decoder settings, we identified data conditions under which global convergence holds and characterized, for the first time, the convergence rates. Our convergence rates hold for NGD and PS, which are shown both theoretically and empirically to outperform vanilla GD. In future work, we aim to further relax Ass. 2 and identify and investigate other sufficient conditions for global convergence, such as overparameterization. Additionally, extending our theory to incorporate momentum and explain its fast convergence is also an interesting future direction. Finally, motivated by our experiments, we want to further investigate why NGD trains faster on language datasets, yet appears to be slower on vision datasets. This may involve extending our findings to the next-token prediction setting, which is also of interest.

## Acknowledgements

PD and CT were supported by NSERC Discovery Grant No. 2021-03677, the Alliance Grant ALLRP 581098-22 and a CIFAR AI Catalyst grant. BV was supported by NSF CAREER Award CCF-2239265. The authors acknowledge the use of the Discovery cluster by USC CARC.

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

## Appendix

## A   Training only $W$

### A.1   Preliminaries

$\|\boldsymbol{A}\|_2$ denotes the spectral norm and $\lambda_{\min}(\boldsymbol{A})$ denotes the minimum eigenvalue. Also, $\|\boldsymbol{A}\|_{2,\infty} = \max_i \|\boldsymbol{a}_i\|$, where $\boldsymbol{a}_i$ is the $i^{\text{th}}$ row of $\boldsymbol{A}$. We list the useful notations used in the Appendix in Table 1 for convenience. Further, let $\boldsymbol{\varphi}'(\boldsymbol{v}) := \nabla\boldsymbol{\varphi}(\boldsymbol{v}) = \operatorname{diag}(\boldsymbol{\varphi}(\boldsymbol{v})) - \boldsymbol{\varphi}(\boldsymbol{v})\boldsymbol{\varphi}(\boldsymbol{v})^\top$ denote the Jacobian of the softmax vector $\boldsymbol{\varphi}(\boldsymbol{v})$ at

| | |
|---|---|
| $\boldsymbol{x}_{i,\tau}$ | $\tau^{\text{th}}$ token in the $i^{\text{th}}$ sample |
| $\boldsymbol{a}(\boldsymbol{W})$ | $\boldsymbol{X}\boldsymbol{W}\boldsymbol{x}_1$ |
| $\boldsymbol{\gamma}$ | $y\boldsymbol{X}\boldsymbol{u}_*$ |
| $\boldsymbol{\varphi}(\cdot)$ | softmax vector |
| $\varphi_{i,\tau}^t$ | $\boldsymbol{\varphi}(\cdot)\vert_\tau,\ \tau \in [\mathrm{T}]$ for sample $i$ at time $t$ |
| $\Lambda$ | $\|\boldsymbol{W}_{\mathrm{mm}}\|$ |
| $B$ | $\max_i \|\boldsymbol{X}_i\|_{2,\infty}$ |
| $\kappa$ | $\dfrac{\max_i(\gamma_{i,\mathrm{opt}_i}-\gamma_i)}{\min_i(\gamma_{i,\mathrm{opt}_i}-\gamma_i)}$ |
| $\Upsilon$ | $\kappa\dfrac{\min_i \exp(\gamma_{i,\mathrm{opt}_i})}{\max_i \exp(\gamma_i)}$ |
| $\zeta$ | $\dfrac{\max_i(\gamma_{i,\mathrm{opt}_i}-\gamma_i)}{\max_i \exp(\gamma_i)}$ |

Table 1: Useful Notation.

$\boldsymbol{v} \in \mathbb{R}^{\mathrm{T}}$. We note that under Assumption 2, for any $\boldsymbol{W}$, we have

$$
\begin{aligned}
\langle -\nabla_{\boldsymbol{W}} \widehat{L}(\boldsymbol{W}_t), \boldsymbol{W} \rangle &= -\frac{1}{n} \sum_{i=1}^{n} \ell'_{t,i} \langle y_i \nabla_{\boldsymbol{W}} \Phi(\boldsymbol{W}_t, \boldsymbol{X}_i), \boldsymbol{W} \rangle \\
&= \frac{1}{n} \sum_{i=1}^{n} \ell_{t,i} (\gamma_{i,\mathrm{opt}_i} - \gamma_i) \varphi^t_{i,\mathrm{opt}_i} (1 - \varphi^t_{i,\mathrm{opt}_i}) \left[ a_{i,\mathrm{opt}_i} - \frac{\sum_{\tau \neq \mathrm{opt}_i} \varphi^t_{i,\tau} a_{i,\tau}}{\sum_{\tau \neq \mathrm{opt}_i} \varphi^t_{i,\tau}} \right],
\end{aligned}
\tag{14}
$$

where $\boldsymbol{a}_i = \boldsymbol{X}_i \boldsymbol{W} \boldsymbol{x}_{i,1}$.

**Intuition for the Constraints in Equation (W-SVM).** The intuition behind the constraints in Equation (W-SVM) is as follows. Consider the loss for the $i^{\mathrm{th}}$ sample:

$$
\ell_i = \ell(y_i \boldsymbol{u}_\star^\top \boldsymbol{X}_i^\top \boldsymbol{\varphi}(\boldsymbol{X}_i \boldsymbol{W} \boldsymbol{x}_{i,1})) = \ell(\boldsymbol{\gamma}_i^\top \boldsymbol{\varphi}(\boldsymbol{X}_i \boldsymbol{W} \boldsymbol{x}_{i,1})) = \ell(\gamma_{i,\mathrm{opt}_i} \varphi_{i,\mathrm{opt}_i} + \gamma_i(1 - \varphi_{i,\mathrm{opt}_i})).
$$

Since $\ell$ is a decreasing function, it is minimized when $\varphi_{i,\mathrm{opt}_i} = 1$.

Next, consider the softmax weight for the optimal token:

$$
\frac{\exp(\boldsymbol{x}_{\mathrm{opt}}^\top \boldsymbol{W}_t \boldsymbol{x}_1)}{\sum_\tau \exp(\boldsymbol{x}_\tau^\top \boldsymbol{W}_t \boldsymbol{x}_1)} = \frac{1}{1 + \sum_{\tau \neq opt} \exp((\boldsymbol{x}_\tau - \boldsymbol{x}_{\mathrm{opt}})^\top \boldsymbol{W}_t \boldsymbol{x}_1)}.
$$

For loss minimization, this score should tend to 1 as the iterate norm $\|\boldsymbol{W}_t\|$ tends to infinity. This can only happen when the exponent is negative, that is, $\boldsymbol{W}_t$ satisfies $(\boldsymbol{x}_\tau - \boldsymbol{x}_{\mathrm{opt}})^\top \boldsymbol{W}_t \boldsymbol{x}_1 < 0$. Then, analogous to translating the constraints to the SVM for training linear models on linearly separable data, we can obtain Equation (W-SVM) for self-attention.

## A.2 Key Lemmas

We first prove some key Lemmas that are useful in the proof of Theorem 1.

**Lemma 2.** Let $w^0_{u,v} \sim \mathcal{N}(0, \sigma_0^2)$ for all $u, v \in [d]$, with the initialization scale

$$
\sigma_0 \leq B^{-2} d^{-1/2} \left( C'^{-1} \log \left( \frac{k\mathrm{T} - 1}{\mathrm{T} - 1} \right) \wedge \eta d^{-1/2} (8\Lambda)^{-1} \right),
$$

for some $k > 1$, and absolute constant $C' > 0$. Further, if $d \geq \log(2/\delta)$ for any $\delta \in (0, 1)$, then with probability at least $1 - \delta$, we have $\varphi^0_{i,opt_i} \geq \frac{1}{k\mathrm{T}}$ for every $i \in [n]$.

*Proof.* Let $w^0_{u,v} \sim \mathcal{N}(0, \sigma_0^2)$, for any $u, v \in [d]$. Then for any $i \in [n]$ and $\tau \in [T]$, for any $\delta \in (0, 1)$, with probability at least $1 - \delta$ we have

$$
\begin{aligned}
(\boldsymbol{x}_{i,\tau} - \boldsymbol{x}_{i,\mathrm{opt}_i})^\top \boldsymbol{W}_0 \boldsymbol{x}_{i,1} &\leq \left| (\boldsymbol{x}_{i,\tau} - \boldsymbol{x}_{i,\mathrm{opt}_i})^\top \boldsymbol{W} \boldsymbol{x}_{i,1} \right| \\
&\leq \|\boldsymbol{x}_{i,\tau} - \boldsymbol{x}_{i,\mathrm{opt}_i}\| \|\boldsymbol{W}_0\|_2 \|\boldsymbol{x}_{i,1}\| \\
&\leq 2B^2 (\sigma_0 C(2\sqrt{d} + \sqrt{\log(2/\delta)})) \\
&\leq 6CB^2 \sigma_0 \sqrt{d} =: B^2 \sigma_0 \sqrt{d} C'.
\end{aligned}
$$

Using this, we have

$$
\begin{aligned}
\varphi^0_{i,\mathrm{opt}_i} &= \frac{1}{1 + \sum_{\tau \neq \mathrm{opt}_i} \exp((\boldsymbol{x}_{i,\tau} - \boldsymbol{x}_{i,\mathrm{opt}_i})^\top \boldsymbol{W}_0 \boldsymbol{x}_{i,1})} \\
&\geq \frac{1}{1 + (T - 1) \exp(B^2 \sigma_0 \sqrt{d} C')} \\
&\geq \frac{1}{kT} \qquad\qquad\qquad \text{(using } \sigma_0 \leq C'^{-1} B^{-2} d^{-1/2} \log \left( \frac{k\mathrm{T} - 1}{\mathrm{T} - 1} \right) )
\end{aligned}
$$

Also, $\|\boldsymbol{W}_0\| \leq 2\sigma_0 d \leq \eta(4B^2\Lambda)^{-1}$. $\qquad\qquad\qquad\qquad\qquad\qquad\qquad\qquad\qquad\qquad\qquad\qquad \square$

**Lemma 3.** *Under the conditions of Lemma 2, Assumptions 1 and 2, using the updates in Eq. (2) with $\eta \leq (2B^2)^{-1} \log(2\mathrm{T} - 1)$, at any time $t \geq 0$, for any sample $j \in [n]$, the softmax weight for the $\boldsymbol{opt}_j$ token satisfies*

$$\varphi^t_{j,\boldsymbol{opt}_j} \geq \varphi^0_{j,\boldsymbol{opt}_j} \wedge \frac{1}{\mathrm{T}}. \tag{15}$$

*Proof.* We prove this by induction. Clearly, Eq. (15) is true for $t = 0$. Next, we assume that it is true at time $t - 1$. Combining this with the conditions of Lemma 2, we have $\varphi^{t-1}_{j,\mathsf{opt}_j} \geq \frac{1}{2\mathrm{T}}$.

Further, for any $j \in [n]$ we can have one of the following two scenarios:

**Scenario 1:** At time $t - 1$, $\varphi^{t-1}_{j,\mathsf{opt}_j} > 1 - \frac{1}{2\mathrm{T}}$. This implies

$$\varphi^{t-1}_{j,\mathsf{opt}_j} = \frac{1}{1 + \sum_{\tau \neq \mathsf{opt}_i} \exp((\boldsymbol{x}_{i,\tau} - \boldsymbol{x}_{j,\mathsf{opt}_j})^\top \boldsymbol{W}_{t-1} \boldsymbol{x}_{j,1})} > 1 - \frac{1}{2\mathrm{T}}$$

$$\implies \sum_{\tau \neq \mathsf{opt}_j} \exp((\boldsymbol{x}_{j,\tau} - \boldsymbol{x}_{j,\mathsf{opt}_j})^\top \boldsymbol{W}_{t-1} \boldsymbol{x}_{j,1}) < \frac{1}{2\mathrm{T} - 1}$$

$$\implies (\boldsymbol{x}_{j,\tau} - \boldsymbol{x}_{j,\mathsf{opt}_j})^\top \boldsymbol{W}_{t-1} \boldsymbol{x}_{j,1} \leq -\log(2\mathrm{T} - 1), \quad \forall \tau \neq \mathsf{opt}_j.$$

After the gradient step at time $t - 1$, for any $\tau \neq \mathsf{opt}_j$, we have,

$$(\boldsymbol{x}_{j,\tau} - \boldsymbol{x}_{j,\mathsf{opt}_j})^\top \boldsymbol{W}_t \boldsymbol{x}_{j,1} = (\boldsymbol{x}_{j,\tau} - \boldsymbol{x}_{j,\mathsf{opt}_j})^\top \boldsymbol{W}_{t-1} \boldsymbol{x}_{j,1} - \eta_{t-1}(\boldsymbol{x}_{j,\mathsf{opt}_j} - \boldsymbol{x}_{j,\tau})^\top(-\nabla_{\boldsymbol{W}} \widehat{L}(\boldsymbol{W}_{t-1})) \boldsymbol{x}_{j,1}$$

$$\leq -\log(2\mathrm{T} - 1) + \eta \|\boldsymbol{x}_{j,\tau} - \boldsymbol{x}_{j,\mathsf{opt}_j}\| \|\boldsymbol{x}_{j,1}\|$$

$$\leq -\log(2\mathrm{T} - 1) + 2\eta B^2$$

$$\leq 0,$$

where the last step follows since $\eta \leq (2B^2)^{-1} \log(2\mathrm{T} - 1)$. This gives

$$\sum_{\tau \neq \mathsf{opt}_j} \exp((\boldsymbol{x}_{j,\tau} - \boldsymbol{x}_{j,\mathsf{opt}_j})^\top \boldsymbol{W}_t \boldsymbol{x}_{j,1}) \leq \mathrm{T} - 1$$

$$\implies \varphi^t_{j,\mathsf{opt}_j} \geq \frac{1}{\mathrm{T}}.$$

**Scenario 2:** At time $t - 1$, $\varphi^{t-1}_{j,\mathsf{opt}_j} \leq 1 - \frac{1}{2\mathrm{T}}$. This gives

$$\varphi^{t-1}_{j,\mathsf{opt}_j}\left(1 - \varphi^{t-1}_{j,\mathsf{opt}_j}\right) \geq \frac{2\mathrm{T} - 1}{4\mathrm{T}^2} \geq \frac{1}{4\mathrm{T}}. \tag{16}$$

We will use this to show that

$$(\boldsymbol{x}_{j,\mathsf{opt}_j} - \boldsymbol{x}_{j,\tau})^\top(-\nabla_{\boldsymbol{W}} \widehat{L}(\boldsymbol{W}_t)) \boldsymbol{x}_{j,1} \geq 0. \tag{17}$$

Let $\boldsymbol{a}' = \boldsymbol{a}((\boldsymbol{x}_{j,\mathsf{opt}_j} - \boldsymbol{x}_{j,\tau})\boldsymbol{x}_{j,1}^\top) = \boldsymbol{X}(\boldsymbol{x}_{j,\mathsf{opt}_j} - \boldsymbol{x}_{j,\tau})\boldsymbol{x}_{j,1}^\top \boldsymbol{x}_1$. Using similar calculations as Eq. (14), we get

$$(\boldsymbol{x}_{j,\mathsf{opt}_j} - \boldsymbol{x}_{j,\tau})^\top(-\nabla_{\boldsymbol{W}} \widehat{L}(\boldsymbol{W}_{t-1})) \boldsymbol{x}_{j,1}$$

$$= \frac{1}{n} \sum_{i=1}^n \ell_{t-1,i}(\gamma_{i,\mathsf{opt}_i} - \gamma_i) \varphi^{t-1}_{i,\mathsf{opt}_i}(1 - \varphi^{t-1}_{i,\mathsf{opt}_i})\left[a'_{i,\mathsf{opt}_i} - \frac{\sum_{\tau \neq \mathsf{opt}_i} \varphi^{t-1}_{i,\tau} a'_{i,\tau}}{\sum_{\tau \neq \mathsf{opt}_i} \varphi^{t-1}_{i,\tau}}\right]. \tag{18}$$

There are two cases:

**Case 1:** $\text{opt}_j \neq 1$. In this case, by splitting the sum over $i \in [n]$ in Eq. (18) into $i = j$ and $i \neq j$, and using non-negativity of the softmax weights, we get

$$
(\boldsymbol{x}_{j,\text{opt}_j} - \boldsymbol{x}_{j,\tau})^\top (-\nabla_{\boldsymbol{W}} \widehat{L}(\boldsymbol{W}_{t-1})) \boldsymbol{x}_{j,1}
$$
$$
\geq \frac{\ell_{t-1,j}}{n} (\gamma_{j,\text{opt}_j} - \gamma_j) \varphi_{j,\text{opt}_j}^{t-1} (1 - \varphi_{j,\text{opt}_j}^{t-1}) \|\boldsymbol{x}_{j,1}\|^2 \min_{\tau' \neq \text{opt}_j} (\boldsymbol{x}_{j,\text{opt}_j} - \boldsymbol{x}_{j,\tau})^\top (\boldsymbol{x}_{j,\text{opt}_j} - \boldsymbol{x}_{j,\tau'})
$$
$$
- \max_{i \neq j} \ell_{t-1,i} (\gamma_{i,\text{opt}_i} - \gamma_i) \varphi_{i,\text{opt}_i}^{t-1} (1 - \varphi_{i,\text{opt}_i}^{t-1}) |\boldsymbol{x}_{i,1}^\top \boldsymbol{x}_{j,1}| \max_{\tau' \neq \text{opt}_i} |(\boldsymbol{x}_{j,\text{opt}_j} - \boldsymbol{x}_{j,\tau})^\top (\boldsymbol{x}_{i,\text{opt}_i} - \boldsymbol{x}_{i,\tau'})|. \tag{19}
$$

Next, we use Assumption 1 and that $n\Upsilon\mathrm{T} \geq 1$ to get

$$
\max_{i \neq j} \max_{\tau' \neq \text{opt}_i} (\boldsymbol{x}_{j,\text{opt}_j} - \boldsymbol{x}_{j,\tau})^\top (\boldsymbol{x}_{i,\text{opt}_i} - \boldsymbol{x}_{i,\tau'}) \leq \max_{i \neq j} |\boldsymbol{x}_{j,\text{opt}_j}^\top \boldsymbol{x}_{i,\text{opt}_i}| + 3 \max_{\substack{i,j:i\neq j, \tau \in [\mathrm{T}], \\ \tau' \in [\mathrm{T}] \setminus \{\text{opt}_j\}}} |\boldsymbol{x}_{i,\tau}^\top \boldsymbol{x}_{j,\tau'}|
$$
$$
\leq 4 \frac{\|\boldsymbol{x}_{j,\text{opt}_j}\|^2}{4} = \|\boldsymbol{x}_{j,\text{opt}_j}\|^2. \tag{20}
$$

Similarly,

$$
\min_{\tau' \neq \text{opt}_j} (\boldsymbol{x}_{j,\text{opt}_j} - \boldsymbol{x}_{j,\tau})^\top (\boldsymbol{x}_{j,\text{opt}_j} - \boldsymbol{x}_{j,\tau'}) \geq \|\boldsymbol{x}_{j,\text{opt}_j}\|^2 - 3 \max_{\substack{\tau \in [\mathrm{T}], \\ \tau' \in [\mathrm{T}] \setminus \{\text{opt}_j\}}} |\boldsymbol{x}_{j,\tau}^\top \boldsymbol{x}_{j,\tau'}|
$$
$$
\geq \|\boldsymbol{x}_{j,\text{opt}_j}\|^2 - 3 \frac{\|\boldsymbol{x}_{j,\text{opt}_j}\|^2}{4} = \frac{1}{4} \|\boldsymbol{x}_{j,\text{opt}_j}\|^2. \tag{21}
$$

Combining Eqs. (16), (20) and (21), we have

$$
\zeta := \frac{n \max_{i \neq j} \ell_{t,i} (\gamma_{i,\text{opt}_i} - \gamma_i) \varphi_{i,\text{opt}_i}^{t-1} (1 - \varphi_{i,\text{opt}_i}^{t-1})}{\ell_{t,j} (\gamma_{j,\text{opt}_j} - \gamma_j) \varphi_{j,\text{opt}_j}^{t-1} (1 - \varphi_{j,\text{opt}_j}^{t-1})} \frac{\max_{i,\tau' \neq \text{opt}_i} (\boldsymbol{x}_{j,\text{opt}_j} - \boldsymbol{x}_{j,\tau})^\top (\boldsymbol{x}_{i,\text{opt}_i} - \boldsymbol{x}_{i,\tau'})}{\min_{\tau' \neq \text{opt}_j} (\boldsymbol{x}_{j,\text{opt}_j} - \boldsymbol{x}_{j,\tau})^\top (\boldsymbol{x}_{j,\text{opt}_j} - \boldsymbol{x}_{j,\tau'})}
$$
$$
\leq n \frac{\max_i \ell_{t-1,i} (\gamma_{i,\text{opt}_i} - \gamma_i)}{\min_i \ell_{t-1,i} (\gamma_{i,\text{opt}_i} - \gamma_i)} \frac{1/4}{1/(4\mathrm{T})} \frac{\|\boldsymbol{x}_{j,\text{opt}_j}\|^2}{\|\boldsymbol{x}_{j,\text{opt}_j}\|^2/4}
$$
$$
\leq 4n\mathrm{T}\kappa \frac{\max_i \ell_{t-1,i}}{\min_i \ell_{t-1,i}}
$$
$$
\leq 4n\mathrm{T}\kappa \frac{\max_i \exp(-(\varphi_{i,\text{opt}_i}^{t-1} \gamma_{i,\text{opt}_i} + (1 - \varphi_{i,\text{opt}_i}^{t-1})\gamma_i))}{\min_i \exp(-(\varphi_{i,\text{opt}_i}^{t-1} \gamma_{i,\text{opt}_i} + (1 - \varphi_{i,\text{opt}_i}^{t-1})\gamma_i))}
$$
$$
\leq 4n\mathrm{T}\kappa \frac{\max_i \exp(-\gamma_i)}{\min_i \exp(-\gamma_{i,\text{opt}_i})} = 4n\Upsilon\mathrm{T}. \tag{22}
$$

Using Assumption 1 and Eq. (22), we have

$$
\|\boldsymbol{x}_{j,1}\|^2 \geq \zeta \max_{i \neq j} |\langle \boldsymbol{x}_{i,1}, \boldsymbol{x}_{j,1} \rangle|,
$$

which implies that Eq. (19) is non-negative.

**Case 2:** $\text{opt}_j = 1$. In this case, let $\mathcal{I} := \{i \in [n] : y_i = y_j \land \text{opt}_i = 1\}$. Using Eq. (18), we get

$$
(\boldsymbol{x}_{j,\text{opt}_j} - \boldsymbol{x}_{j,\tau})^\top (-\nabla_{\boldsymbol{W}} \widehat{L}(\boldsymbol{W}_{t-1})) \boldsymbol{x}_{j,1}
$$
$$
\geq \frac{\ell_{t-1,j}}{n} (\gamma_{j,\text{opt}_j} - \gamma_j) \varphi_{j,\text{opt}_j}^{t-1} (1 - \varphi_{j,\text{opt}_j}^{t-1}) \|\boldsymbol{x}_{j,1}\|^2 \min_{\tau' \neq \text{opt}_j} (\boldsymbol{x}_{j,\text{opt}_j} - \boldsymbol{x}_{j,\tau})^\top (\boldsymbol{x}_{j,\text{opt}_j} - \boldsymbol{x}_{j,\tau'})
$$
$$
+ \min_{i \neq j, i \in \mathcal{I}} \ell_{t-1,i} (\gamma_{i,\text{opt}_i} - \gamma_i) \varphi_{i,\text{opt}_i}^{t-1} (1 - \varphi_{i,\text{opt}_i}^{t-1}) \boldsymbol{x}_{i,1}^\top \boldsymbol{x}_{j,1} \min_{\tau' \neq \text{opt}_i} (\boldsymbol{x}_{j,\text{opt}_j} - \boldsymbol{x}_{j,\tau})^\top (\boldsymbol{x}_{i,\text{opt}_i} - \boldsymbol{x}_{i,\tau'})
$$
$$
- \max_{i \neq j, i \notin \mathcal{I}} \ell_{t-1,i} (\gamma_{i,\text{opt}_i} - \gamma_i) \varphi_{i,\text{opt}_i}^{t-1} (1 - \varphi_{i,\text{opt}_i}^{t-1}) |\boldsymbol{x}_{i,1}^\top \boldsymbol{x}_{j,1}| \max_{\tau' \neq \text{opt}_i} |(\boldsymbol{x}_{j,\text{opt}_j} - \boldsymbol{x}_{j,\tau})^\top (\boldsymbol{x}_{i,\text{opt}_i} - \boldsymbol{x}_{i,\tau'})|. \tag{23}
$$

Using Assumption 1, the second term is non-negative. For the remaining two terms, we can proceed in a similar way as the previous case to show that using Assumption 1, these terms are also non-negative.

Using Eq. (17), we get

$$(\boldsymbol{x}_{j,\mathsf{opt}_j} - \boldsymbol{x}_{j,\tau})^\top (\boldsymbol{W}_t - \boldsymbol{W}_{t-1})\boldsymbol{x}_{j,1} = \eta \frac{(\boldsymbol{x}_{j,\mathsf{opt}_j} - \boldsymbol{x}_{j,\tau})^\top (-\nabla_{\boldsymbol{W}}\widehat{L}(\boldsymbol{W}_{t-1}))\boldsymbol{x}_{j,1}}{\|\nabla_{\boldsymbol{W}}\widehat{L}(\boldsymbol{W}_{t-1})\|} \geq 0.$$

Then, for any $\tau \neq \mathsf{opt}_j$, we have

$$\frac{\varphi_{j,\mathsf{opt}_j}^t / \varphi_{j,\tau}^t}{\varphi_{j,\mathsf{opt}_j}^{t-1} / \varphi_{j,\tau}^{t-1}} = \exp\left((\boldsymbol{x}_{j,\mathsf{opt}_j} - \boldsymbol{x}_{j,\tau})^\top (\boldsymbol{W}_t - \boldsymbol{W}_{t-1})\boldsymbol{x}_{j,1}\right) \geq 1.$$

By telescoping, we get $\frac{\varphi_{j,\mathsf{opt}_j}^t}{\varphi_{j,\tau}^t} \geq \frac{\varphi_{j,\mathsf{opt}_j}^0}{\varphi_{j,\tau}^0}$, which implies that $\varphi_{j,\mathsf{opt}_j}^t \geq \varphi_{j,\mathsf{opt}_j}^0$.

$\square$

**Lemma 4.** *Under Assumption 2, for $\boldsymbol{W}_{mm}$ as defined in* (W-SVM) *and any $\boldsymbol{W}$,*

$$\left\langle -\frac{\nabla_{\boldsymbol{W}}\widehat{L}(\boldsymbol{W})}{\|\nabla_{\boldsymbol{W}}\widehat{L}(\boldsymbol{W})\|_F}, \frac{\boldsymbol{W}_{mm}}{\|\boldsymbol{W}_{mm}\|}\right\rangle \geq (2B^2\Lambda)^{-1} > 0.$$

*Proof.* First, using Eq. (14) for $\boldsymbol{W}_{\mathrm{mm}}$, we have

$$\langle -\nabla_{\boldsymbol{W}}\widehat{L}(\boldsymbol{W}), \boldsymbol{W}_{\mathrm{mm}}\rangle \geq \min_i(a_{i,\mathsf{opt}_i}^* - \max_{\tau \neq \mathsf{opt}_i} a_{i,\tau}^*)\frac{1}{n}\sum_{i=1}^n \ell_i(\gamma_{i,\mathsf{opt}_i} - \gamma_i)\varphi_{i,\mathsf{opt}_i}(1 - \varphi_{i,\mathsf{opt}_i})$$

$$\geq \frac{1}{n}\sum_{i=1}^n \ell_i(\gamma_{i,\mathsf{opt}_i} - \gamma_i)\varphi_{i,\mathsf{opt}_i}(1 - \varphi_{i,\mathsf{opt}_i}), \tag{24}$$

where $\boldsymbol{a}_i^* = \boldsymbol{X}_i \boldsymbol{W}_{\mathrm{mm}} \boldsymbol{x}_{i,1}$, and the second inequality follows by the definition of $\boldsymbol{W}_{\mathrm{mm}}$.

Let $\hat{\boldsymbol{W}} := -\frac{\nabla_{\boldsymbol{W}}\widehat{L}(\boldsymbol{W})}{\|\nabla_{\boldsymbol{W}}\widehat{L}(\boldsymbol{W})\|_F}$, and $\hat{\boldsymbol{a}}_i = \boldsymbol{X}_i \hat{\boldsymbol{W}} \boldsymbol{x}_{i,1}$, then by similar calculations as Eq. (14), we have

$$\|\nabla_{\boldsymbol{W}}\widehat{L}(\boldsymbol{W})\|_F = \langle -\nabla_{\boldsymbol{W}}\widehat{L}(\boldsymbol{W}), \hat{\boldsymbol{W}}\rangle$$

$$= \frac{1}{n}\sum_{i=1}^n \ell_i(\gamma_{i,\mathsf{opt}_i} - \gamma_i)\varphi_{i,\mathsf{opt}_i}(1 - \varphi_{i,\mathsf{opt}_i})\left[\hat{a}_{i,\mathsf{opt}_i} - \frac{\sum_{\tau \neq \mathsf{opt}_i}\varphi_{i,\tau}\hat{a}_{i,\tau}}{\sum_{\tau \neq \mathsf{opt}_i}\varphi_{i,\tau}}\right]$$

$$\leq \max_i(\hat{a}_{i,\mathsf{opt}_i} - \min_{\tau \neq \mathsf{opt}_i}\hat{a}_{i,\tau})\frac{1}{n}\sum_{i=1}^n \ell_i(\gamma_{i,\mathsf{opt}_i} - \gamma_i)\varphi_{i,\mathsf{opt}_i}(1 - \varphi_{i,\mathsf{opt}_i})$$

$$\leq 2B^2 \frac{1}{n}\sum_{i=1}^n \ell_i(\gamma_{i,\mathsf{opt}_i} - \gamma_i)\varphi_{i,\mathsf{opt}_i}(1 - \varphi_{i,\mathsf{opt}_i}), \tag{25}$$

where for the last step we use

$$\max_i(\hat{a}_{i,\mathsf{opt}_i} - \min_{\tau \neq \mathsf{opt}_i}\hat{a}_{i,\tau}) \leq 2\max_{i,\tau}|\hat{a}_{i,\tau}| = 2\max_{i,\tau}|\boldsymbol{x}_{i,\tau}^\top \hat{\boldsymbol{W}}\boldsymbol{x}_{i,1}|$$

$$\leq 2\max_i\|\boldsymbol{X}_i\|_{2,\infty}\|\boldsymbol{x}_{i,1}\|\|\hat{\boldsymbol{W}}\| \leq 2B^2.$$

Using Eqs. (24) and (25), we get

$$\left\langle -\frac{\nabla_{\boldsymbol{W}}\widehat{L}(\boldsymbol{W})}{\|\nabla_{\boldsymbol{W}}\widehat{L}(\boldsymbol{W})\|_F}, \frac{\boldsymbol{W}_{\mathrm{mm}}}{\|\boldsymbol{W}_{\mathrm{mm}}\|}\right\rangle \geq (2B^2\Lambda)^{-1}.$$

$\square$

**Lemma 5** (Iterate Norm). *Using the updates in Eq. (2), under the conditions of Lemma 2 and Assumption 2, at any time $t > 0$, we have*

$$2\eta((4B^2\Lambda)^{-1} \vee 1)t \geq \|\boldsymbol{W}_t\|_F \geq \eta(4B^2\Lambda)^{-1}t. \tag{26}$$

*Proof.* Using Eq. (2) and Lemma 4 with $\boldsymbol{W} = \boldsymbol{W}_t$, we have

$$\begin{aligned}
\|\boldsymbol{W}_t\| &= \left\|\boldsymbol{W}_0 - \sum_{t'=0}^{t-1} \eta_{t'} \nabla_{\boldsymbol{W}} \widehat{L}(\boldsymbol{W}_t)\right\| \\
&\geq \langle \boldsymbol{W}_0, \overline{\boldsymbol{W}}_{\mathrm{mm}} \rangle + \sum_{t'=0}^{t-1} \eta \left\langle -\frac{\nabla_{\boldsymbol{W}} \widehat{L}(\boldsymbol{W}_t)}{\|\nabla_{\boldsymbol{W}} \widehat{L}(\boldsymbol{W}_t)\|_F}, \frac{\boldsymbol{W}_{\mathrm{mm}}}{\|\boldsymbol{W}_{\mathrm{mm}}\|} \right\rangle \\
&\geq \eta(2B^2\Lambda)^{-1}t - \left|\langle \boldsymbol{W}_0, \overline{\boldsymbol{W}}_{\mathrm{mm}} \rangle\right|.
\end{aligned}$$

Under the conditions of Lemma 2,

$$\left|\langle \boldsymbol{W}_0, \overline{\boldsymbol{W}}_{\mathrm{mm}} \rangle\right| \leq \|\boldsymbol{W}_0\|_F \leq \eta(4B^2\Lambda)^{-1},$$

which gives the required result for $t > 0$.

Similarly, for the upper bound we have

$$\begin{aligned}
\|\boldsymbol{W}_t\|_F &= \left\|\boldsymbol{W}_0 - \sum_{t'=0}^{t-1} \eta_{t'} \nabla_{\boldsymbol{W}} \widehat{L}(\boldsymbol{W}_t)\right\|_F \\
&\leq \|\boldsymbol{W}_0\| + \sum_{t'=0}^{t-1} \eta \\
&\leq \eta(4B^2\Lambda)^{-1} + \eta t.
\end{aligned}$$

$\square$

**Lemma 6** (Data Example 1). *Using $\boldsymbol{u}_* = \frac{1}{U^2}(\boldsymbol{\mu}_+ - \boldsymbol{\mu}_-)$ and $d \gtrsim \Omega(n^2 \mathrm{T}^2 \log\left(\frac{4n}{\delta}\right))$, data generated as per Example 1 satisfies Ass. 1 and 2 with probability at least $1 - \delta$, for any $\delta \in (0,1)$.*

*Proof.* Using $\boldsymbol{u}_* = \frac{1}{U^2}(\boldsymbol{\mu}_+ - \boldsymbol{\mu}_-)$, for all $i \in [n]$ and $\tau \neq \mathtt{opt}_i$, we have

$$\gamma_{i,\mathtt{opt}_i} = y\boldsymbol{u}_*^\top \boldsymbol{x}_{i,\mathtt{opt}_i} = 1, \quad \gamma_{i,\tau} = y\boldsymbol{u}_*^\top \boldsymbol{x}_{i,\tau} = 0.$$

Using this, we have

$$\Upsilon = \frac{\max_i(\gamma_{i,\mathtt{opt}_i} - \gamma_i)}{\min_i(\gamma_{i,\mathtt{opt}_i} - \gamma_i)} \frac{\min_i \exp(\gamma_{i,\mathtt{opt}_i})}{\max_i \exp(\gamma_i)} = e.$$

Next, by using Bernstein inequality, we know with probability at least $1 - \frac{\delta}{2n}$, for every $i \in [n]$ and $\tau \neq \mathtt{opt}_i$ we have

$$\begin{aligned}
\left|\|\boldsymbol{x}_{i,\mathtt{opt}_i}\|^2 - \sigma^2 d - U^2\right| &\leq c\sigma^2\sqrt{d\log\left(\frac{4n}{\delta}\right)} \leq \frac{\sigma^2 d}{2}, \\
\left|\|\boldsymbol{x}_{i,\tau}\|^2 - \rho^2 d\right| &\leq c\rho^2\sqrt{d\log\left(\frac{4n}{\delta}\right)} \leq \frac{\rho^2 d}{2}.
\end{aligned}$$

where $c > 0$ is an absolute constant and we use $d \geq 4c^2 \log\left(\frac{4n}{\delta}\right)$.

Further, for any $i, j \in [n], i \neq j$ and any $\tau, \tau' \in [\mathrm{T}]$. By applying Bernstein's inequality, each of the following is true with probability at least $1 - \frac{\delta}{2n^2}$,

$$\left|\langle \boldsymbol{x}_{i,\mathtt{opt}_i}, \boldsymbol{x}_{j,\mathtt{opt}_j} \rangle - \langle \boldsymbol{\mu}_{y_i}, \boldsymbol{\mu}_{y_j} \rangle\right| \leq 2\sigma^2\sqrt{d\log\left(\frac{4n}{\delta}\right)}, \quad \left|\langle \boldsymbol{x}_{i,\tau'}, \boldsymbol{x}_{j,\tau} \rangle\right| \leq 2\rho^2\sqrt{d\log\left(\frac{4n}{\delta}\right)},$$

$$\left|\langle \boldsymbol{x}_{i,\mathtt{opt}_i}, \boldsymbol{x}_{j,\tau} \rangle\right| \leq 2\sigma\rho\sqrt{d\log\left(\frac{4n}{\delta}\right)},$$

where $\tau' \neq \mathtt{opt}_i, \tau \neq \mathtt{opt}_j$.

In order to satisfy Assumption 1, ignoring the log factors, we require

$$(U^2 + \sigma^2 d) \wedge \rho^2 d \gtrsim nT(\rho^2\sqrt{d} \vee \sigma\rho\sqrt{d})$$
$$U^2 + \sigma^2\sqrt{d} \gtrsim nT(\sigma^2\sqrt{d} \vee \rho^2\sqrt{d} \vee \sigma\rho\sqrt{d})$$

The above inequalities hold if we have $d = \widetilde{\Omega}(n^2\mathrm{T}^2)$ and $U^2 \geq \widetilde{\Omega}((\sigma \vee \rho)^2 n\mathrm{T}\sqrt{d})$. The proof finishes by the application of a union bound. □

We now state the key Lemma used in the proof of Thm. 1 below, with the expression for $R_\epsilon$.

**Lemma 7.** *Under the conditions of Lemma 2 and Assumptions 1 and 2, for any $\epsilon \in (0,1)$, there exists*

$$R_\epsilon := 2\Lambda\epsilon^{-1}(\log(4n(B^2\Lambda)^{-1}\Upsilon\mathrm{T}^3\epsilon^{-1}) \vee 5\log(20\mathrm{T}B^2\Lambda\epsilon^{-1})),$$

*such that for every t where $\|\boldsymbol{W}_t\| \geq R_\epsilon$,*

$$\left\langle -\nabla_{\boldsymbol{W}}\widehat{L}(\boldsymbol{W}_t), \frac{\boldsymbol{W}_{mm}}{\|\boldsymbol{W}_{mm}\|} \right\rangle \geq (1-\epsilon)\left\langle -\nabla_{\boldsymbol{W}}\widehat{L}(\boldsymbol{W}_t), \frac{\boldsymbol{W}_t}{\|\boldsymbol{W}_t\|} \right\rangle. \tag{27}$$

**Remark 1.** *Lem. 7 improves Lem. 10 of (Tarzanagh et al., 2023a), which only holds for $n=1$, i.e., for a single training sample (same restriction as (Tarzanagh et al., 2023b)). Our key idea to extend the result to $n \geq 1$ is to divide the samples into three sets, based on whether the constraints in (W-SVM) are violated ($\mathcal{I}_1$), satisfied with a small margin ($\mathcal{I}_2$) or well satisfied ($\mathcal{I}_3$). We prove a tighter version for samples in $\mathcal{I}_1$ and the residual allows us to prove the result for the sum over all samples.*

*Proof.* Let $\tilde{\boldsymbol{a}}_i := \boldsymbol{X}_i\widetilde{\boldsymbol{W}}_t\boldsymbol{x}_{i,1}$, $\boldsymbol{a}_i^* := \boldsymbol{X}_i\boldsymbol{W}_{mm}\boldsymbol{x}_{i,1}$. We consider two scenarios based on whether $\|\widetilde{\boldsymbol{W}}_t - \boldsymbol{W}_{mm}\| \leq \frac{\nu}{2B^2}$ (Scenario 1) or not (Scenario 2), where

$$\widetilde{\boldsymbol{W}}_t := \frac{\boldsymbol{W}_t}{\|\boldsymbol{W}_t\|}\|\boldsymbol{W}_{mm}\|, \ \nu := \frac{\epsilon}{(1-\epsilon)}. \tag{28}$$

Scenario 1 of the proof is the same as (Tarzanagh et al., 2023a). Below, we consider Scenario 2.

Using the definition of $\nu$ and similar calculations to Eq. (14), Eq. (27) translates to

$$\frac{1}{n}\sum_{i=1}^n \ell_{t,i}(\gamma_{i,\mathtt{opt}_i} - \gamma_i)\varphi_{i,\mathtt{opt}_i}^t(1 - \varphi_{i,\mathtt{opt}_i}^t)\tilde{h}_{t,i} \leq (1+\nu)\frac{1}{n}\sum_{i=1}^n \ell_{t,i}(\gamma_{i,\mathtt{opt}_i} - \gamma_i)\varphi_{i,\mathtt{opt}_i}^t(1 - \varphi_{i,\mathtt{opt}_i}^t)h_{t,i}^*, \tag{29}$$

where $\tilde{h}_{t,i} := \tilde{a}_{i,\mathtt{opt}_i} - \frac{\sum_{\tau \neq \mathtt{opt}_i}\varphi_{i,\tau}^t\tilde{a}_{i,\tau}}{\sum_{\tau \neq \mathtt{opt}_i}\varphi_{i,\tau}^t}$ and $h_{t,i}^*$ is defined similarly.

For any sample $i \in [n]$, let

$$\delta_i^{\max}(\epsilon,\omega) := \max_{\tau \notin \{\mathtt{opt}_i\}} \tilde{a}_{i,\mathtt{opt}_i} - \tilde{a}_{i,\tau} - 1,$$
$$\delta_i^{\min}(\epsilon,\omega) := \min_{\tau \notin \{\mathtt{opt}_i\}} \tilde{a}_{i,\mathtt{opt}_i} - \tilde{a}_{i,\tau} - 1.$$

Define the following sets

$$\mathcal{I}_1 := \{i \in [n] \mid \delta_i^{\min} \leq 0.3\nu\},$$
$$\mathcal{I}_2 := \{i \in [n] \mid 0.3\nu < \delta_i^{\min} \leq 0.8\nu\},$$
$$\mathcal{I}_3 := \{i \in [n] \mid \delta_i^{\min} \geq 0.8\nu\}.$$

Intuitively, $\mathcal{I}_1$ is the set of samples for which some (W-SVM) constraints are either violated or all constraints are barely satisfied. Similarly, $\mathcal{I}_3$ contains samples where all the (W-SVM) constraints are satisfied. Finally,

$\mathcal{I}_2$ makes up the rest of the samples.

First, we will show that for any $i \in \mathcal{I}_1$,

$$\left[ \tilde{a}_{i,\mathtt{opt}_i} - \frac{\sum_{\tau \neq \mathtt{opt}_i} \varphi_{i,\tau}^t \tilde{a}_{i,\tau}}{\sum_{\tau \neq \mathtt{opt}_i} \varphi_{i,\tau}^t} \right] \leq (1 + 0.5\nu) \left[ a_{i,\mathtt{opt}_i}^* - \frac{\sum_{\tau \neq \mathtt{opt}_i} \varphi_{i,\tau}^t a_{i,\tau}^*}{\sum_{\tau \neq \mathtt{opt}_i} \varphi_{i,\tau}^t} \right]. \tag{30}$$

Next, we will show that for any $i \in \mathcal{I}_2$, we have $\tilde{h}_{t,i} \leq (1 + \nu) h_{t,i}^*$, i.e.

$$\left[ \tilde{a}_{i,\mathtt{opt}_i} - \frac{\sum_{\tau \neq \mathtt{opt}_i} \varphi_{i,\tau}^t \tilde{a}_{i,\tau}}{\sum_{\tau \neq \mathtt{opt}_i} \varphi_{i,\tau}^t} \right] \leq (1 + \nu) \left[ a_{i,\mathtt{opt}_i}^* - \frac{\sum_{\tau \neq \mathtt{opt}_i} \varphi_{i,\tau}^t a_{i,\tau}^*}{\sum_{\tau \neq \mathtt{opt}_i} \varphi_{i,\tau}^t} \right]. \tag{31}$$

Finally, notice that in order to complete the proof for Eq. (29), using Eqs. (30) and (31) it suffices to show that

$$\sum_{i \in \mathcal{I}_3} \ell_{t,i}(\gamma_{i,\mathtt{opt}_i} - \gamma_i)\varphi_{i,\mathtt{opt}_i}^t(1 - \varphi_{i,\mathtt{opt}_i}^t)(\tilde{h}_{t,i} - (1 + \nu)h_{t,i}^*)$$
$$\leq 0.5\nu \sum_{i \in \mathcal{I}_1} \ell_{t,i}(\gamma_{i,\mathtt{opt}_i} - \gamma_i)\varphi_{i,\mathtt{opt}_i}^t(1 - \varphi_{i,\mathtt{opt}_i}^t)h_{t,i}^*. \tag{32}$$

Using $h_{t,i}^* \geq 1$, Eq. (32) follows when

$$\sum_{i \in \mathcal{I}_3} \ell_{t,i}(\gamma_{i,\mathtt{opt}_i} - \gamma_i)\varphi_{i,\mathtt{opt}_i}^t(1 - \varphi_{i,\mathtt{opt}_i}^t)(\tilde{h}_{t,i} - 1 - \nu) \leq 0.5\nu \sum_{i \in \mathcal{I}_1} \ell_{t,i}(\gamma_{i,\mathtt{opt}_i} - \gamma_i)\varphi_{i,\mathtt{opt}_i}^t(1 - \varphi_{i,\mathtt{opt}_i}^t). \tag{33}$$

**Case 1:** Sample $i \in \mathcal{I}_1$, i.e. $\delta_i^{\min} \leq 0.3\nu$.

If $\delta_i^{\max} < 0.5\nu$, (30) follows directly. For the rest of the samples $i$ where $\delta_i^{\max} \geq 0.5\nu$, let

$$\mathcal{N} := \{\tau \in [\mathrm{T}] : a_{i,\mathtt{opt}_i} - a_{i,\tau} \leq 1 + 0.4\nu\}.$$

Let $R' := \frac{\|\boldsymbol{W}_t\|}{\|\boldsymbol{W}_{\mathrm{mm}}\|}$, then by definition of $\mathcal{N}, \delta_{\min}$, we have

$$\frac{\sum_{\tau \neq \mathtt{opt}_i, \tau \notin \mathcal{N}} \varphi_{i,\tau}^t}{\sum_{\tau \neq \mathtt{opt}_i} \varphi_{i,\tau}^t} \leq \frac{\mathrm{T} \max_{\tau \neq \mathtt{opt}_i, \tau \notin \mathcal{N}} \varphi_{i,\tau}^t}{\varphi_{i,\tau}^t} \leq \frac{\mathrm{T} \exp(-R'(1 + 0.4\nu))}{\exp(-R'(1 + \delta_i^{\min}))}$$
$$\leq \mathrm{T} \exp(-0.1R'\nu).$$

Combining these, we have

$$\frac{\sum_{\tau \neq \mathtt{opt}_i} (\tilde{a}_{i,\mathtt{opt}_i} - \tilde{a}_{i,\tau})\varphi_{i,\tau}^t}{\sum_{\tau \neq \mathtt{opt}_i} \varphi_{i,\tau}^t} \leq (1 + 0.4\nu)\frac{\sum_{\tau \in \mathcal{N}} \varphi_{i,\tau}^t}{\sum_{\tau \neq \mathtt{opt}_i} \varphi_{i,\tau}^t} + (1 + \delta_i^{\max})\frac{\sum_{\tau \neq \mathtt{opt}_i, \tau \notin \mathcal{N}} \varphi_{i,\tau}^t}{\sum_{\tau \neq \mathtt{opt}_i} \varphi_{i,\tau}^t}$$
$$\leq (1 + 0.4\nu)\frac{\sum_{\tau \neq \mathtt{opt}_i} \varphi_{i,\tau}^t - \sum_{\tau \neq \mathtt{opt}_i \notin \mathcal{N}} \varphi_{i,\tau}^t}{\sum_{\tau \neq \mathtt{opt}_i} \varphi_{i,\tau}^t} + (1 + \delta_i^{\max})\frac{\sum_{\tau \neq \mathtt{opt}_i, \tau \notin \mathcal{N}} \varphi_{i,\tau}^t}{\sum_{\tau \neq \mathtt{opt}_i} \varphi_{i,\tau}^t}$$
$$\leq 1 + 0.4\nu + (\delta_i^{\max} - 0.4\nu)\mathrm{T} \exp(-0.1R'\nu).$$

To satisfy (30), we need $(\delta_i^{\max} - 0.4\nu)\mathrm{T} \exp(-0.1R'\nu) \leq 0.1\nu$, which is true when

$$R' \geq (0.1\nu)^{-1} \log(\mathrm{T}(\delta_i^{\max}/(0.1\nu) - 4)).$$

This is true since $R' \geq 10\epsilon^{-1} \log(10\mathrm{T} \max_{i \in [n]} \delta_i^{\max}/\epsilon)$, where we use the definition of $\nu$ from Eq. (28).

**Case 2:** Sample $i \in \mathcal{I}_2$, i.e. $0.3\nu < \delta^{\min} \leq 0.8\nu$.

If $\delta_i^{\max} < \nu$, (31) follows directly. For the rest of the samples where $\delta_i^{\max} \geq \nu$, we can proceed in a similar way as Case 1. We use a threshold of $0.9\nu$ to split the tokens, and obtain

$$\frac{\sum\limits_{\tau \neq \mathsf{opt}_i} (a_{i,\mathsf{opt}_i} - a_{i,\tau})\varphi_{t,i,\tau}}{\sum\limits_{\tau \neq \mathsf{opt}_i} \varphi_{t,i,\tau}} \leq 1 + \nu,$$

when $R' \geq 10\epsilon^{-1} \log(10\mathrm{T} \max_{i \in [n]} \delta_i^{\max}/\epsilon)$.

**Case 3:** Sample $i \in \mathcal{I}_3$, i.e. $\delta_i^{\min} \geq 0.8\nu$.

As $\|\widetilde{\boldsymbol{W}}_t - \boldsymbol{W}_{\mathrm{mm}}\| \geq \frac{\nu}{2B^2}$, at least one sample $i \in [n]$ violates the (W-SVM) constraints, i.e. $|\mathcal{I}_1| \geq 1$. We have

$$\sum_{i \in \mathcal{I}_1} \ell_{t,i}(\gamma_{i,\mathsf{opt}_i} - \gamma_i)\varphi_{i,\mathsf{opt}_i}^t(1 - \varphi_{i,\mathsf{opt}_i}^t) \geq \min_{i \in \mathcal{I}_1} \ell_{t,i}(\gamma_{i,\mathsf{opt}_i} - \gamma_i)\min_t(\varphi_{i,\mathsf{opt}_i}^t)^2 \sum_{\tau \neq \mathsf{opt}_i} \varphi_{i,\tau}^t/\varphi_{i,\mathsf{opt}_i}^t$$

$$\geq \min_{i \in \mathcal{I}_1} \ell_{t,i}(\gamma_{i,\mathsf{opt}_i} - \gamma_i)(4\mathrm{T})^{-2}\exp(-(1 + 0.3\nu)R'), \tag{34}$$

where the last inequality follows by using Lemmas 2 and 3, and that $\delta_i^{\min} \leq 0.3\nu$ for any $i \in \mathcal{I}_1$. Also,

$$\sum_{i \in \mathcal{I}_3} \ell_{t,i}(\gamma_{i,\mathsf{opt}_i} - \gamma_i)\varphi_{i,\mathsf{opt}_i}^t(1 - \varphi_{i,\mathsf{opt}_i}^t) \leq (n-1)\max_{i \in \mathcal{I}_3} \ell_{t,i}(\gamma_{i,\mathsf{opt}_i} - \gamma_i)\max_{i \in \mathcal{I}_3} \sum_{\tau \neq \mathsf{opt}_i} \varphi_{i,\tau}^t/\varphi_{i,\mathsf{opt}_i}^t$$

$$\leq (n-1)\max_{i \in \mathcal{I}_3} \ell_{t,i}(\gamma_{i,\mathsf{opt}_i} - \gamma_i)(\mathrm{T}-1)\exp(-(1 + 0.8\nu)R'), \tag{35}$$

where the last inequality follows because $0.8\nu \leq \min_{i \in \mathcal{I}_3} \delta_i^{\min}$. Combining Eqs. (34) and (35) gives us (33) when

$$\exp(0.5\nu R') \geq 0.5\nu(2B^2\Lambda)^{-1}(n-1)(\mathrm{T}-1)(4\mathrm{T})^2 \frac{\max_{i \in \mathcal{I}_3} \ell_{t,i}(\gamma_{i,\mathsf{opt}_i} - \gamma_i)}{\min_{i \in \mathcal{I}_1} \ell_{t,i}(\gamma_{i,\mathsf{opt}_i} - \gamma_i)}, \tag{36}$$

which is true when

$$\exp(0.5\nu R') \geq 4n(B^2\Lambda)^{-1}\Upsilon\mathrm{T}^3\epsilon^{-1},$$

which is true when $R' \geq 2\epsilon^{-1}\log(4n(B^2\Lambda)^{-1}\Upsilon\mathrm{T}^3\epsilon^{-1})$.

Combining all conditions on $R'$, and using $\delta_i^{\max} \leq 2B^2\Lambda$ for all $i \in [n]$, we get the desired result since

$$R_\epsilon \geq \Lambda^{-1} = 2\Lambda\epsilon^{-1}(\log(4n(B^2\Lambda)^{-1}\Upsilon\mathrm{T}^3\epsilon^{-1}) \vee 5\log(20\mathrm{T}B^2\Lambda\epsilon^{-1})).$$

$\square$

### A.3 Proof of Theorem 1

We first restate Theorem 1, this time with the exact constants.

**Theorem 5** (IB Rate). *Under the conditions of Lemma 2 and Assumptions 1 and 2, using the updates in Eq. (2) with $\eta \leq (2B^2)^{-1}\log(2\mathrm{T}-1)$, for any $t \geq t_0 := \left(\frac{10B\Lambda}{\sqrt{\eta}}\right)^3 \vee \log\left(\frac{\eta\mathrm{T}(n\Upsilon(B^2\Lambda)^{-2}\mathrm{T}^2\vee 5)}{20\Lambda}\right),$*

$$\left\langle \frac{\boldsymbol{W}_t}{\|\boldsymbol{W}_t\|}, \frac{\boldsymbol{W}_{mm}}{\|\boldsymbol{W}_{mm}\|} \right\rangle \geq 1 - \frac{C(\eta, B, \Lambda, t_0)(\log t)^2}{t},$$

*where $C(\eta, B, \Lambda, t_0) = 4\eta^{-1}B^2\Lambda\left(\|\boldsymbol{W}_{t_0}\| - \langle\boldsymbol{W}_{t_0}, \overline{\boldsymbol{W}}_{mm}\rangle + 2B^2\Lambda(40\Lambda + \eta)\right).$*

**Remark 2** (Comparison to (Tarzanagh et al., 2023a).)**.** *Thm. 1 establishes* finite-time *convergence of NGD to $\boldsymbol{W}_{mm}$ starting from* any *initialization direction. This marks a clear improvement over prior work (Tarzanagh et al., 2023a), which demonstrates that GD converges asymptotically $(t \to \infty)$ to $\boldsymbol{W}_{mm}$ only under an appropriate* local *initialization direction. The only previously known global convergence result in Tarzanagh et al. (2023b;a) requires Ass. 2 and $n = 1$, limiting its applicability to training with a single sample. Thm. 1 addresses this limitation and additionally provides convergence rates. For completeness, we demonstrate in Thm. 6 in the Appendix that GD (with constant step size) also converges globally to $\boldsymbol{W}_{mm}$ under the conditions specified in Thm. 1, without requiring $n = 1$.*

*Proof.* First, we use Lemma 5 to show that Lemma 7 is true for any $t \geq t_0$, by selecting $\epsilon = \epsilon_t :=$ $(80B^2\Lambda^2 \log t)(\eta t)^{-1}$. Since $\log t < 1.25 t^{1/3}$ for $t > 0$, $\epsilon_t \leq 1$ for $t \geq (10B\Lambda\eta^{-1/2})^3$. We also have

$$2\Lambda\epsilon^{-1}(\log(4n\Upsilon(B^2\Lambda)^{-1}\mathrm{T}^3\epsilon^{-1}) \vee 5\log(20\mathrm{T}B^2\Lambda\epsilon^{-1})) \leq 10\Lambda\epsilon^{-1}\log((4n\Upsilon(B^2\Lambda)^{-1}\mathrm{T}^3 \vee 20\mathrm{T}B^2\Lambda)\epsilon^{-1})$$

$$= 10\Lambda \frac{\eta t}{80B^2\Lambda^2 \log t} \log\left((4n\Upsilon(B^2\Lambda)^{-1}\mathrm{T}^3 \vee 20\mathrm{T}B^2\Lambda)\frac{\eta t}{80B^2\Lambda^2 \log t}\right)$$

$$\leq \frac{\eta(4B^2\Lambda)^{-1}t}{\log t}\left(\log t \vee \log\left(\frac{\eta \mathrm{T}(n\Upsilon(B^2\Lambda)^{-2}\mathrm{T}^2 \vee 5)}{20\Lambda}\right)\right)$$

$$\leq \eta(4B^2\Lambda)^{-1}t.$$

Thus, $\|\boldsymbol{W}_t\| \geq R$ for $t \geq t_0$ and we can use Lemma 7 with $\epsilon = \epsilon_t$, *i.e.*,

$$\left\langle -\nabla_{\boldsymbol{W}}\widehat{L}(\boldsymbol{W}_t), \frac{\boldsymbol{W}_{mm}}{\|\boldsymbol{W}_{mm}\|}\right\rangle \geq (1 - \epsilon_t)\left\langle -\nabla_{\boldsymbol{W}}\widehat{L}(\boldsymbol{W}_t), \frac{\boldsymbol{W}_t}{\|\boldsymbol{W}_t\|}\right\rangle.$$

Using the update Eq. (2), we have

$$\left\langle \boldsymbol{W}_{t+1} - \boldsymbol{W}_t, \frac{\boldsymbol{W}_{mm}}{\|\boldsymbol{W}_{mm}\|}\right\rangle \geq (1 - \epsilon_t)\left\langle \boldsymbol{W}_{t+1} - \boldsymbol{W}_t, \frac{\boldsymbol{W}_t}{\|\boldsymbol{W}_t\|}\right\rangle$$

$$= \frac{1}{2\|\boldsymbol{W}_t\|}(\|\boldsymbol{W}_{t+1}\|^2 - \|\boldsymbol{W}_t\|^2 - \|\boldsymbol{W}_{t+1} - \boldsymbol{W}_t\|^2) - \epsilon_t\left\langle \boldsymbol{W}_{t+1} - \boldsymbol{W}_t, \frac{\boldsymbol{W}_t}{\|\boldsymbol{W}_t\|}\right\rangle$$

$$= \frac{1}{2\|\boldsymbol{W}_t\|}(\|\boldsymbol{W}_{t+1}\|^2 - \|\boldsymbol{W}_t\|^2) - \frac{\eta^2}{2\|\boldsymbol{W}_t\|} + \epsilon_t\eta\left\langle \overline{\nabla_{\boldsymbol{W}}\widehat{L}(\boldsymbol{W}_t)}, \overline{\boldsymbol{W}_t}\right\rangle \qquad \text{(using Eq. (2))}$$

$$\geq \|\boldsymbol{W}_{t+1}\| - \|\boldsymbol{W}_t\| - \frac{\eta^2}{2\|\boldsymbol{W}_t\|} - \epsilon_t\eta \qquad \left(\text{since } \frac{a^2 - b^2}{2b} \geq a - b \ \forall \ a, b > 0\right)$$

$$\geq \|\boldsymbol{W}_{t+1}\| - \|\boldsymbol{W}_t\| - \frac{80B^2\Lambda^2 \log t}{t} - 2\eta B^2\Lambda(t+1)^{-1} \qquad \text{(substituting } \epsilon_t, \text{ using Lemma 5).}$$

Summing over $t \geq t_0$, we get

$$\left\langle \boldsymbol{W}_t, \overline{\boldsymbol{W}}_{mm}\right\rangle \geq \|\boldsymbol{W}_t\| - \|\boldsymbol{W}_{t_0}\| + \left\langle \boldsymbol{W}_{t_0}, \overline{\boldsymbol{W}}_{mm}\right\rangle - 80B^2\Lambda^2 \sum_{t'=t_0}^{t}(\log t')(t')^{-1} - 2\eta B^2\Lambda \sum_{t'=t_0}^{t}(t'+1)^{-1}. \qquad (37)$$

We bound the last two terms as follows

$$\sum_{t'=t_0}^{t}(\log t')(t')^{-1} \leq \log t \sum_{t'=t_0}^{t}(t')^{-1} \leq (\log t)^2,$$

$$\sum_{t'=t_0}^{t}(t'+1)^{-1} \leq \log t.$$

Using these in Eq. (37) and dividing by $\|\boldsymbol{W}_t\|$ throughout, we get

$$\left\langle \frac{\boldsymbol{W}_t}{\|\boldsymbol{W}_t\|}, \frac{\boldsymbol{W}_{mm}}{\|\boldsymbol{W}_{mm}\|}\right\rangle \geq 1 - \frac{1}{\|\boldsymbol{W}_t\|}\left(\|\boldsymbol{W}_{t_0}\| - \left\langle \boldsymbol{W}_{t_0}, \overline{\boldsymbol{W}}_{mm}\right\rangle + 2B^2\Lambda(40\Lambda + \eta)(\log t)^2\right)$$

$$\geq 1 - \frac{C(\eta, B, \Lambda, t_0)(\log t)^2}{t},$$

where the last step follows by Lemma 5. $\qquad \square$

## A.4 Optimal Rate for NGD

In Th. 1, we showed an upper bound on the correlation $\langle \overline{\boldsymbol{W}}_t, \overline{\boldsymbol{W}}_{\mathrm{mm}} \rangle \le 1 - \tilde{\Omega}(t^{-1})$ for NGD. In this section, we consider the example used in Sec. 3.3 to derive a lower bound on the correlation.

We restate the example here for convenience. Let $n = 1$, $\mathrm{T} = 2$, $y = 1$, $\boldsymbol{x}_1 = [1, 0]^\top$, $\boldsymbol{x}_2 = [0, 0]^\top$ and $\boldsymbol{u}_* = [1, 0]^\top$. As we saw in Sec. 3.3, $\mathtt{opt} = 1$, $\gamma_{\mathtt{opt}} - \gamma = 1$, $\boldsymbol{W}_{\mathrm{mm}} = (\boldsymbol{x}_1 - \boldsymbol{x}_2)\boldsymbol{x}_1^\top = \boldsymbol{X}$ and $\Lambda = 1$. Using this, we can write

$$\nabla_{\boldsymbol{W}} \widehat{L}(\boldsymbol{W}_t) = -\widehat{L}(\boldsymbol{W}_t)\varphi_{\mathtt{opt}}^t(1 - \varphi_{\mathtt{opt}}^t)\boldsymbol{W}_{\mathrm{mm}}, \text{ and } \boldsymbol{W}_{t+1} = \boldsymbol{W}_0 + \eta G_t \boldsymbol{W}_{\mathrm{mm}},$$

where $G_t \boldsymbol{W}_{\mathrm{mm}} \coloneqq \sum_{t'=0}^t \frac{\nabla_{\boldsymbol{W}} \widehat{L}(\boldsymbol{W}_{t'})}{\|\nabla_{\boldsymbol{W}} \widehat{L}(\boldsymbol{W}_{t'})\|} = (t+1)\boldsymbol{W}_{\mathrm{mm}}$.

Clearly, $\|\boldsymbol{W}_t\| \le \eta t + \|\boldsymbol{W}_0\| \le \mathcal{O}(t)$. Using Eq. (3), we have

$$\langle \overline{\boldsymbol{W}}_t, \overline{\boldsymbol{W}}_{\mathrm{mm}} \rangle \approx 1 - \frac{\|\boldsymbol{W}_0\|^2 - (W_0^{11})^2}{2\|\boldsymbol{W}_t\|^2} \le 1 - \Omega(t^{-2}).$$

Comparing this with the upper bound, we see that the rate in Th. 1 may not be the optimal rate for NGD. However, it is the first finite-time convergence rate for self-attention. Improving this rate is an interesting direction for future work.

## A.5 Proofs for Additional Results in Section 3.2

**Lemma 8** (Softmax score rate). *Under the conditions of Lemma 2 and Assumptions 1 and 2, using the updates in Eq. (2), for any $i \in [n]$ and any $t \ge 2^{11}(B^2\Lambda)^2 C(\eta, B, \Lambda, t_0) \vee t_0$, where $C(\eta, B, \Lambda, t_0) \coloneqq 4\eta^{-1}B^2\Lambda \left( \|\boldsymbol{W}_{t_0}\| - \langle \boldsymbol{W}_{t_0}, \overline{\boldsymbol{W}}_{mm} \rangle + 2B^2\Lambda(40\Lambda + \eta) \right)$, $t_0 \coloneqq \left( \frac{10B\Lambda}{\sqrt{\eta}} \right)^3 \vee \log \left( \frac{\eta \mathrm{T}(n\Upsilon(B^2\Lambda)^{-2}\mathrm{T}^2 \vee 5)}{20\Lambda} \right)$,*

$$\varphi_{i,opt_i}^t \ge \frac{1}{1 + (T-1)\exp(-\eta(8B^2\Lambda^2)^{-1}t)}.$$

*Proof.* Using Theorem 1 and Lemma 5, we have

$$(\boldsymbol{x}_{i,\tau} - \boldsymbol{x}_{i,\mathtt{opt}_i})^\top (\boldsymbol{W}_t - \overline{\boldsymbol{W}}_{\mathrm{mm}}\|\boldsymbol{W}_t\| + \overline{\boldsymbol{W}}_{\mathrm{mm}}\|\boldsymbol{W}_t\|)\boldsymbol{x}_{i,1} \le 2B^2\|\boldsymbol{W}_t\| \left\| \overline{\boldsymbol{W}}_t - \overline{\boldsymbol{W}}_{\mathrm{mm}} \right\| - \frac{1}{\|\boldsymbol{W}_{\mathrm{mm}}\|}\|\boldsymbol{W}_t\|$$

$$\le 2\sqrt{2}B^2 \frac{\sqrt{C(\eta, B, \Lambda, t_0)}}{t^{1/2}}(2\eta t) - \eta(4B^2\Lambda^2)^{-1}t$$

$$= 4\eta B^2 \sqrt{2C(\eta, B, \Lambda, t_0)}\sqrt{t} - \eta(4B^2\Lambda^2)^{-1}t$$

$$\le -\eta(8B^2\Lambda^2)^{-1}t, \tag{38}$$

since $t \ge 2^{11}(B^2\Lambda)^2 C(\eta, B, \Lambda, t_0)$.

We use this to find the softmax rate as follows

$$\varphi_{i,\mathtt{opt}_i}^t = \frac{1}{1 + \sum_{\tau \neq \mathtt{opt}_i} \exp((\boldsymbol{x}_{i,\tau} - \boldsymbol{x}_{i,\mathtt{opt}_i})^\top \boldsymbol{W}_t \boldsymbol{x}_{i,1})}$$

$$= \frac{1}{1 + \sum_{\tau \neq \mathtt{opt}_i} \exp((\boldsymbol{x}_{i,\tau} - \boldsymbol{x}_{i,\mathtt{opt}_i})^\top (\boldsymbol{W}_t - \overline{\boldsymbol{W}}_{\mathrm{mm}}\|\boldsymbol{W}_t\| + \overline{\boldsymbol{W}}_{\mathrm{mm}}\|\boldsymbol{W}_t\|)\boldsymbol{x}_{i,1})}$$

$$\ge \frac{1}{1 + (T-1)\exp(-\eta(8B^2\Lambda^2)^{-1}t)}.$$

$\square$

**Theorem 6** (Asymptotic Convergence of GD). *Under the conditions of Lemma 2 and Assumptions 1 and 2, using the standard GD updates with $\eta \leq \frac{\log(2T-1)}{B^4\zeta}$, for any $t \geq t_\epsilon$ such that $\|\boldsymbol{W}_t\| \geq R_\epsilon \vee 1/2$,*

$$\left\langle \frac{\boldsymbol{W}_t}{\|\boldsymbol{W}_t\|}, \frac{\boldsymbol{W}_{mm}}{\|\boldsymbol{W}_{mm}\|} \right\rangle \geq 1 - \epsilon - \frac{C(\eta, B, \Lambda, \epsilon)}{\|\boldsymbol{W}_t\|},$$

*where $C(\eta, B, \Lambda, \epsilon) = \frac{2B^2\Lambda}{\eta}(1-\epsilon)\|\boldsymbol{W}_{t_\epsilon}\| \left(1 - (1-\epsilon)^{-1}\left\langle \overline{\boldsymbol{W}}_{t_\epsilon}, \overline{\boldsymbol{W}}_{mm}\right\rangle - 2\eta\|\boldsymbol{W}_{t_\epsilon}\|^{-1}\widehat{L}(\boldsymbol{\theta}_{t_\epsilon})\right)$.*

*Proof.* First, we show that Lemma 3 holds for standard GD updates, *i.e.*, for any $t \geq 0$ and any $j \in [n]$,

$$\varphi^t_{j,\mathtt{opt}_j} \geq \varphi^0_{j,\mathtt{opt}_j} \wedge \frac{1}{\mathrm{T}}.$$

Following similar steps as the proof of Lemma 3, we consider two scenarios,
**Scenario 1:** At time $t-1$, $\varphi^{t-1}_{j,\mathtt{opt}_j} > 1 - \frac{1}{2\mathrm{T}}$. This implies

$$\varphi^{t-1}_{j,\mathtt{opt}_j} \leq -\log(2\mathrm{T}-1), \quad \forall \tau \neq \mathtt{opt}_j.$$

After the gradient step at time $t-1$, for any $\tau \neq \mathtt{opt}_j$, we have,

$$
\begin{aligned}
(\boldsymbol{x}_{j,\tau} - \boldsymbol{x}_{j,\mathtt{opt}_j})^\top \boldsymbol{W}_t \boldsymbol{x}_{j,1} &= (\boldsymbol{x}_{j,\tau} - \boldsymbol{x}_{j,\mathtt{opt}_j})^\top \boldsymbol{W}_{t-1} \boldsymbol{x}_{j,1} - \eta(\boldsymbol{x}_{j,\mathtt{opt}_j} - \boldsymbol{x}_{j,\tau})^\top(-\nabla_{\boldsymbol{W}}\widehat{L}(\boldsymbol{W}_{t-1}))\boldsymbol{x}_{j,1} \\
&\leq -\log(2\mathrm{T}-1) + \eta\|\boldsymbol{x}_{j,\tau} - \boldsymbol{x}_{j,\mathtt{opt}_j}\|\|\boldsymbol{x}_{j,1}\|\|\nabla_{\boldsymbol{W}}\widehat{L}(\boldsymbol{W}_{t-1})\| \\
&\leq -\log(2\mathrm{T}-1) + \eta(2B^2)(B^2\widehat{L}(\boldsymbol{W}_{t-1})\max_i(\gamma_{i,\mathtt{opt}_i} - \gamma_i)\varphi^t_{i,\mathtt{opt}_i}(1-\varphi^t_{i,\mathtt{opt}_i})) \\
&\leq -\log(2\mathrm{T}-1) + \eta B^4 \max_i \exp(-\gamma_i)\max_i(\gamma_{i,\mathtt{opt}_i} - \gamma_i) \\
&\leq 0,
\end{aligned}
$$

where the last step follows since $\eta \leq \frac{\log(2\mathrm{T}-1)}{B^4\zeta}$. This gives $\varphi^t_{j,\mathtt{opt}_j} \geq \frac{1}{\mathrm{T}}$.
**Scenario 2:** At time $t-1$, $\varphi^{t-1}_{j,\mathtt{opt}_j} \leq 1 - \frac{1}{2\mathrm{T}}$. The proof for this part is the same as Lemma 3.

Using this, we can easily show that Lemma 7 is true for GD updates, *i.e.*, for any $\epsilon \in (0,1)$, there exists

$$R_\epsilon := 2\Lambda\epsilon^{-1}(\log(4n(B^2\Lambda)^{-1}\Upsilon\mathrm{T}^3\epsilon^{-1}) \vee 5\log(20\mathrm{T}B^2\Lambda\epsilon^{-1})),$$

such that for every $t$ where $\|\boldsymbol{W}_t\| \geq R_\epsilon$,

$$\left\langle -\nabla_{\boldsymbol{W}}\widehat{L}(\boldsymbol{W}_t), \frac{\boldsymbol{W}_{\mathrm{mm}}}{\|\boldsymbol{W}_{\mathrm{mm}}\|} \right\rangle \geq (1-\epsilon)\left\langle -\nabla_{\boldsymbol{W}}\widehat{L}(\boldsymbol{W}_t), \frac{\boldsymbol{W}_t}{\|\boldsymbol{W}_t\|} \right\rangle.$$

Once we have this, the remaining proof is the same as Theorem 4 in (Tarzanagh et al., 2023a). $\qquad\square$

**Theorem 7** (IB rate for Polyak-step). *Under the conditions of Lemma 2 and Assumptions 1 and 2, using GD updates with Polyak-step, with $\eta \leq (2B^2)^{-1}\omega_1\log(2T-1)$, where $\omega_1 = \beta(1-\beta)\Lambda^{-1}\min_i(\gamma_{i,opt_i} - \gamma_i)$, for any $T_0 \geq t \geq t_0 := \omega_2\nu\left(\left(\frac{10B\Lambda}{\sqrt{\eta}}\right)^3 \vee \log\left(\frac{\eta\mathrm{T}(n\Upsilon(B^2\Lambda)^{-2}\mathrm{T}^2\vee 5)}{20\Lambda}\right)\right)$, where $\omega_2 = 0.5B^2\max_i(\gamma_{i,opt_i} - \gamma_i)$, such that for any $i \in [n]$, $\varphi^t_{i,opt_i} \leq \beta$,*

$$\left\langle \frac{\boldsymbol{W}_t}{\|\boldsymbol{W}_t\|}, \frac{\boldsymbol{W}_{mm}}{\|\boldsymbol{W}_{mm}\|} \right\rangle \geq 1 - \frac{C(\eta, B, \Lambda, t_0)(\log t)^2}{t},$$

*where $C(\eta, B, \Lambda, t_0) = 4\eta^{-1}B^2\Lambda\omega_2\nu\left(\|\boldsymbol{W}_{t_0}\| - \left\langle \boldsymbol{W}_{t_0}, \overline{\boldsymbol{W}}_{mm}\right\rangle + 2B^2\Lambda\omega_1^{-1}(40\Lambda + \eta\upsilon\omega_1^{-1}\omega_2)\right)$.*

*Proof.* To extend our analysis and results for NGD updates to the Polyak-step, we mainly use the following lower bound on the gradient norm,

$$
\begin{aligned}
\|\nabla_{\boldsymbol{W}}\widehat{L}(\boldsymbol{W}_t)\| &\geq \widehat{L}(\boldsymbol{W}_t)\min_i\left\langle y_i\nabla_{\boldsymbol{W}}\Phi(\boldsymbol{\theta}_t,\boldsymbol{X}_i),\frac{\boldsymbol{W}_{\mathrm{mm}}}{\|\boldsymbol{W}_{\mathrm{mm}}\|}\right\rangle \\
&= \Lambda^{-1}\widehat{L}(\boldsymbol{W}_t)\min_i(\gamma_{i,\mathrm{opt}_i}-\gamma_i)\varphi^t_{i,\mathrm{opt}_i}(1-\varphi^t_{i,\mathrm{opt}_i})\left[a^*_{i,\mathrm{opt}_i}-\frac{\sum_{\tau\neq\mathrm{opt}_i}\varphi^t_{i,\tau}a^*_{i,\tau}}{\sum_{\tau\neq\mathrm{opt}_i}\varphi^t_{i,\tau}}\right] \\
&\geq \Lambda^{-1}\widehat{L}(\boldsymbol{W}_t)\min_i(\gamma_{i,\mathrm{opt}_i}-\gamma_i)\min_{i,t'\leq t}\varphi^{t'}_{i,\mathrm{opt}_i}(1-\varphi^{t'}_{i,\mathrm{opt}_i}) \\
&\geq \beta(1-\beta)\Lambda^{-1}\min_i(\gamma_{i,\mathrm{opt}_i}-\gamma_i)\widehat{L}(\boldsymbol{W}_t) = \omega_1\widehat{L}(\boldsymbol{W}_t).
\end{aligned}
$$

Note that similar lower bounds have appeared when deriving loss convergence rates of linear predictors trained with NGD on separable data (Nacson et al., 2019).

We also have the following upper bound,

$$
\|\nabla_{\boldsymbol{W}}\widehat{L}(\boldsymbol{W}_t)\| \leq \widehat{L}(\boldsymbol{W}_t)(1/4)\max_i(\gamma_{i,\mathrm{opt}_i}-\gamma_i)(2B^2) = \omega_2\widehat{L}(\boldsymbol{W}_t).
$$

First, we show that Lemma 3 holds for Polyak-step updates, *i.e.*, for any $t\geq 0$ and any $j\in[n]$,

$$
\varphi^t_{j,\mathrm{opt}_j} \geq \varphi^0_{j,\mathrm{opt}_j}\wedge\frac{1}{\mathrm{T}}.
$$

Following similar steps as the proof of Lemma 3, we consider two scenarios,
**Scenario 1:** At time $t-1$, $\varphi^{t-1}_{j,\mathrm{opt}_j}>1-\frac{1}{2\mathrm{T}}$. This implies

$$
\varphi^{t-1}_{j,\mathrm{opt}_j}\leq-\log(2\mathrm{T}-1),\qquad\forall\tau\neq\mathrm{opt}_j.
$$

After the gradient step at time $t-1$, for any $\tau\neq\mathrm{opt}_j$, we have,

$$
\begin{aligned}
(\boldsymbol{x}_{j,\tau}-\boldsymbol{x}_{j,\mathrm{opt}_j})^\top\boldsymbol{W}_t\boldsymbol{x}_{j,1} &= (\boldsymbol{x}_{j,\tau}-\boldsymbol{x}_{j,\mathrm{opt}_j})^\top\boldsymbol{W}_{t-1}\boldsymbol{x}_{j,1}-\eta_{t-1}(\boldsymbol{x}_{j,\mathrm{opt}_j}-\boldsymbol{x}_{j,\tau})^\top(-\nabla_{\boldsymbol{W}}\widehat{L}(\boldsymbol{W}_{t-1}))\boldsymbol{x}_{j,1} \\
&\leq -\log(2\mathrm{T}-1)+\eta\frac{\widehat{L}(\boldsymbol{W}_{t-1})-\widehat{L}^*}{\|\nabla_{\boldsymbol{W}}\widehat{L}(\boldsymbol{W}_{t-1})\|^2}\|\boldsymbol{x}_{j,\tau}-\boldsymbol{x}_{j,\mathrm{opt}_j}\|\|\boldsymbol{x}_{j,1}\|\|\nabla_{\boldsymbol{W}}\widehat{L}(\boldsymbol{W}_{t-1})\| \\
&\leq -\log(2\mathrm{T}-1)+\eta\frac{\widehat{L}(\boldsymbol{W}_{t-1})}{\omega_1\widehat{L}(\boldsymbol{W}_{t-1})}(2B^2) \\
&= -\log(2\mathrm{T}-1)+\eta\frac{2B^2}{\omega_1} \\
&\leq 0,
\end{aligned}
$$

where the last step follows since $\eta\leq\frac{\omega_1\log(2\mathrm{T}-1)}{2B^2}$. This gives $\varphi^t_{j,\mathrm{opt}_j}\geq\frac{1}{\mathrm{T}}$.

**Scenario 2:** At time $t-1$, $\varphi^{t-1}_{j,\mathrm{opt}_j}\leq 1-\frac{1}{2\mathrm{T}}$. The proof for this part is the same as Lemma 3.

Using this, we can easily show that Lemma 7 is true for Polyak-step updates, *i.e.*, for any $\epsilon\in(0,1)$, there exists

$$
R_\epsilon := 2\Lambda\epsilon^{-1}(\log(4n(B^2\Lambda)^{-1}\Upsilon\mathrm{T}^3\epsilon^{-1})\vee 5\log(20\mathrm{T}B^2\Lambda\epsilon^{-1})),
$$

such that for every $t$ where $\|\boldsymbol{W}_t\|\geq R_\epsilon$,

$$
\left\langle-\nabla_{\boldsymbol{W}}\widehat{L}(\boldsymbol{W}_t),\frac{\boldsymbol{W}_{\mathrm{mm}}}{\|\boldsymbol{W}_{\mathrm{mm}}\|}\right\rangle\geq(1-\epsilon)\left\langle-\nabla_{\boldsymbol{W}}\widehat{L}(\boldsymbol{W}_t),\frac{\boldsymbol{W}_t}{\|\boldsymbol{W}_t\|}\right\rangle. \tag{39}
$$

Next, following similar steps as the proof of Lemma 5, the iterate norm growth can be characterized as follows. Using Lemma 4 with $\boldsymbol{W} = \boldsymbol{W}_t$, for any $t \leq T_0$,

$$
\begin{aligned}
\|\boldsymbol{W}_t\| &= \left\| \boldsymbol{W}_0 - \sum_{t'=0}^{t-1} \eta_{t'} \nabla_{\boldsymbol{W}} \widehat{L}(\boldsymbol{W}_t) \right\| \\
&\geq \left\langle \boldsymbol{W}_0, \overline{\boldsymbol{W}}_{\mathrm{mm}} \right\rangle + \sum_{t'=0}^{t-1} \eta \frac{\widehat{L}(\boldsymbol{W}_t) - \widehat{L}^*}{\|\nabla_{\boldsymbol{W}} \widehat{L}(\boldsymbol{W}_t)\|} \left\langle -\frac{\nabla_{\boldsymbol{W}} \widehat{L}(\boldsymbol{W}_t)}{\|\nabla_{\boldsymbol{W}} \widehat{L}(\boldsymbol{W}_t)\|}, \frac{\boldsymbol{W}_{\mathrm{mm}}}{\|\boldsymbol{W}_{\mathrm{mm}}\|} \right\rangle \\
&\geq \eta (2B^2 \Lambda \omega_2 \upsilon)^{-1} t - \left| \left\langle \boldsymbol{W}_0, \overline{\boldsymbol{W}}_{\mathrm{mm}} \right\rangle \right| \\
&\geq \eta (4B^2 \Lambda \omega_2 \upsilon)^{-1} t,
\end{aligned}
$$

since $\omega_2 > 1$. Using this, and following similar steps as the proof of Theorem 1, we can show that Eq. (39) is true for any $t \geq t_0$. Using $\epsilon_t = \epsilon_t := (80 B^2 \Lambda^2 \log t)(\eta t)^{-1}$, we have,

$$
\begin{aligned}
&\left\langle \boldsymbol{W}_{t+1} - \boldsymbol{W}_t, \frac{\boldsymbol{W}_{\mathrm{mm}}}{\|\boldsymbol{W}_{\mathrm{mm}}\|} \right\rangle \geq (1 - \epsilon_t) \left\langle \boldsymbol{W}_{t+1} - \boldsymbol{W}_t, \frac{\boldsymbol{W}_t}{\|\boldsymbol{W}_t\|} \right\rangle \\
&= \frac{1}{2\|\boldsymbol{W}_t\|} (\|\boldsymbol{W}_{t+1}\|^2 - \|\boldsymbol{W}_t\|^2) - \frac{\eta^2}{2\|\boldsymbol{W}_t\|} \frac{(\widehat{L}(\boldsymbol{W}_t) - \widehat{L}^*)^2}{\|\nabla_{\boldsymbol{W}} \widehat{L}(\boldsymbol{W}_t)\|^2} + \epsilon_t \eta \frac{\widehat{L}(\boldsymbol{W}_t) - \widehat{L}^*}{\|\nabla_{\boldsymbol{W}} \widehat{L}(\boldsymbol{W}_t)\|} \left\langle \overline{\nabla_{\boldsymbol{W}} \widehat{L}(\boldsymbol{W}_t)}, \overline{\boldsymbol{W}_t} \right\rangle \\
&\geq \|\boldsymbol{W}_{t+1}\| - \|\boldsymbol{W}_t\| - \frac{\eta^2}{2\omega_1^2 \|\boldsymbol{W}_t\|} - \frac{\epsilon_t \eta}{\omega_1} \\
&\geq \|\boldsymbol{W}_{t+1}\| - \|\boldsymbol{W}_t\| - 80 B^2 \Lambda^2 \omega_1^{-1} (\log t) t^{-1} - 2\eta \upsilon B^2 \Lambda \omega_1^{-2} \omega_2 (t+1)^{-1}.
\end{aligned}
$$

Then, telescoping over $t \geq t_0$, we get

$$
\begin{aligned}
\left\langle \frac{\boldsymbol{W}_t}{\|\boldsymbol{W}_t\|}, \frac{\boldsymbol{W}_{\mathrm{mm}}}{\|\boldsymbol{W}_{\mathrm{mm}}\|} \right\rangle &\geq 1 - \frac{1}{\|\boldsymbol{W}_t\|} \left( \|\boldsymbol{W}_{t_0}\| - \left\langle \boldsymbol{W}_{t_0}, \overline{\boldsymbol{W}}_{\mathrm{mm}} \right\rangle + 2B^2 \Lambda \omega_1^{-1} (40\Lambda + \eta \upsilon \omega_1^{-1} \omega_2)(\log t)^2 \right) \\
&\geq 1 - \frac{C(\eta, B, \Lambda, t_0)(\log t)^2}{t}.
\end{aligned}
$$

$\square$

## B   Joint Optimization

### B.1   Preliminaries

We first state the complete version of Condition 1 below.

**Condition 2** (Complete version of Condition 1). *Let $B = \alpha \rho \sqrt{1.5d}$. We consider the following conditions for any $\delta \in (0, 1)$,*

- *Zero initialization: $\|\boldsymbol{u}_0\| = 0, \|\boldsymbol{W}_0\| = 0$,*

- *$\boldsymbol{opt}$ token has larger norm: $\alpha \geq 6$,*

- *Sufficient overparameterization: $d \geq C_0 \alpha^4 n^4 \log \left( \frac{10n^2}{\delta} \right)$, where $C_0$ is an absolute constant,*

- *Small step-size: $\eta \leq \frac{1}{18\alpha^2 \rho^2 d} \wedge \frac{\alpha \rho}{160n} \wedge \frac{\log(2)}{3B} \wedge \frac{B}{128\omega_0 \sqrt{1.5}n}$, where $\omega_0 = 13(B \vee 1)^5 (B \vee d)$.*

**Lemma 9.** *Under the data model DM and Condition 1, the following events are true with probability at least $1 - \delta$,*

- $E_1 := \left\{ \frac{\alpha^2 \rho^2 d}{2} \leq \|\boldsymbol{x}_{i, \boldsymbol{opt}_i}\|^2 \leq \frac{3\alpha^2 \rho^2 d}{2}, \ \forall i \in [n] \right\}$,

- $E_2 := \left\{ \frac{\rho^2 d}{2} \leq \|\boldsymbol{x}_{i, \tau}\|^2 \leq \frac{3\rho^2 d}{2}, \ \forall i \in [n], \tau \neq \boldsymbol{opt}_i \right\}$,

- $E_3 \coloneqq \{|\langle \boldsymbol{x}_{i,opt_i}, \boldsymbol{x}_{j,opt_j}\rangle| \leq 2\alpha^2\rho^2\sqrt{d\log\left(\frac{10n^2}{\delta}\right)}, \forall i \neq j \in [n]\}$,

- $E_4 \coloneqq \{|\langle \boldsymbol{x}_{i,\tau'}, \boldsymbol{x}_{j,\tau}\rangle| \leq 2\rho^2\sqrt{d\log\left(\frac{10n^2}{\delta}\right)}, \forall i \neq j \in [n], \tau' \neq \boldsymbol{opt}_i, \tau \neq \boldsymbol{opt}_j\}$,

- $E_5 \coloneqq \{|\langle \boldsymbol{x}_{i,opt_i}, \boldsymbol{x}_{j,\tau}\rangle| \leq 2\alpha\rho^2\sqrt{d\log\left(\frac{10n^2}{\delta}\right)}, \forall i \neq j \in [n], \tau \neq \boldsymbol{opt}_j\}$,

*Proof.* First, by applying Bernstein's inequality, with probability at least $1 - \frac{\delta}{5n}$, for every $i \in [n]$,

$$\left|\|\boldsymbol{x}_{i,\mathrm{opt}_i}\|^2 - \alpha^2\rho^2 d\right| \leq c\alpha^2\rho^2\sqrt{d\log\left(\frac{10n}{\delta}\right)} \leq \frac{\alpha^2\rho^2 d}{2},$$

where $c > 0$ is an absolute constant and we use $d \geq (2c)^2\log\left(\frac{10n}{\delta}\right)$ in the second inequality.

Similarly, since $d \geq (2c)^2\log\left(\frac{10n}{\delta}\right)$, we also have that with probability at least $1 - \frac{\delta}{5n}$, for every $i \in [n]$, $\tau \neq \mathrm{opt}_i$,

$$\left|\|\boldsymbol{x}_{i,\tau}\|^2 - \rho^2 d\right| \leq c\rho^2\sqrt{d\log\left(\frac{10n}{\delta}\right)} \leq \frac{\rho^2 d}{2}.$$

Further, for any $i, j \in [n], i \neq j$ and any $\tau, \tau' \in [\mathrm{T}]$ since $\langle \boldsymbol{x}_{i,\tau}, \boldsymbol{x}_{j,\tau'}\rangle$ is zero-mean. By applying Bernstein's inequality, each of the following is true with probability at least $1 - \frac{\delta}{5n^2}$,

$$|\langle \boldsymbol{x}_{i,\mathrm{opt}_i}, \boldsymbol{x}_{j,\mathrm{opt}_j}\rangle| \leq 2\alpha^2\rho^2\sqrt{d\log\left(\frac{10n^2}{\delta}\right)}, \quad |\langle \boldsymbol{x}_{i,\tau'}, \boldsymbol{x}_{j,\tau}\rangle| \leq 2\rho^2\sqrt{d\log\left(\frac{10n^2}{\delta}\right)},$$

$$|\langle \boldsymbol{x}_{i,\mathrm{opt}_i}, \boldsymbol{x}_{j,\tau}\rangle| \leq 2\alpha\rho^2\sqrt{d\log\left(\frac{10n^2}{\delta}\right)},$$

where $\tau' \neq \mathrm{opt}_i, \tau \neq \mathrm{opt}_j$.

Applying a union bound over all these finishes the proof.

$\square$

## B.2 Key Lemmas

We first state some intermediate Lemmas that are useful in the proofs of Theorem 2 and 3.

**Lemma 10** (Iterate Norm). *Using the updates in Eq. (10), at any time $t > 0$,*

$$\|\boldsymbol{u}_t\| \leq 3\eta t^{1/3}.$$

*Proof.* Using Eq. (10) and triangle inequality, we get

$$\|\boldsymbol{u}_t\| \leq \|\boldsymbol{u}_{t-1}\| + \frac{\eta}{t^{2/3}}\frac{\|\nabla_{\boldsymbol{u}}\widehat{L}(\boldsymbol{\theta}_t)\|}{\|\nabla_{\boldsymbol{u}}\widehat{L}(\boldsymbol{\theta}_t)\|} \leq \eta\sum_{t'=0}^{t-1}(1+t')^{-2/3} \leq 3\eta t^{1/3}.$$

$\square$

**Lemma 11.** *Under Condition 1 and the data model DM, for any $j \in [n]$, at any $t > 0$,*

$$\gamma_{j,opt_j}^t - \gamma_j^t \geq \eta\omega_2\sum_{t'=0}^{t-1}(t+1)^{-2/3}, \quad \varphi_{j,opt_j}^t \geq \frac{1}{2} \quad and \quad \max_{i,j\in[n]}\frac{\ell_{t,i}}{\ell_{t,j}} \leq C,$$

*where $\omega_2 = \omega_2(\alpha, \rho, n) \coloneqq \frac{\alpha\rho}{n}\sqrt{\log\left(\frac{10n^2}{\delta}\right)}$ and $C \geq 24$ is a universal constant.*

*Proof.* We will prove this by induction. First, we prove the three parts at $t = 1$ as follows.

Since $\boldsymbol{u}_0 = \boldsymbol{0}$, for any $j \in [n]$, $\ell_{0,j} = 1$. Using this, We have

$$
\begin{aligned}
(\gamma^1_{j,\mathrm{opt}_j} - \gamma^1_j) &= -\eta^u_0 (-\nabla_{\boldsymbol{u}} \widehat{L}(\boldsymbol{\theta}_0))^\top (\boldsymbol{x}_{j,\mathrm{opt}_j} - \boldsymbol{x}_{j,\tau}) \\
&= \frac{\eta^u_0}{n} \sum_{i=1}^n (y_i \boldsymbol{X}_i^\top \boldsymbol{\varphi}(\boldsymbol{a}_i(\boldsymbol{W}_t)))^\top (\boldsymbol{x}_{j,\mathrm{opt}_j} - \boldsymbol{x}_{j,\tau}) \\
&\geq \frac{\eta^u_0}{n} (\varphi^t_{j,\mathrm{opt}_j} \boldsymbol{x}_{j,\mathrm{opt}_j} + (1 - \varphi^t_{j,\mathrm{opt}_j}) \boldsymbol{x}_{j,\tau})^\top (\boldsymbol{x}_{j,\mathrm{opt}_j} - \boldsymbol{x}_{j,\tau}) \\
&\quad + \frac{\eta^u_0}{n} \sum_{i \neq j} \ell_{t,i} y_i y_j (\varphi^t_{i,\mathrm{opt}_i} \boldsymbol{x}_{i,\mathrm{opt}_i} + (1 - \varphi^t_{i,\mathrm{opt}_i}) \boldsymbol{x}_{i,\tau})^\top (\boldsymbol{x}_{j,\mathrm{opt}_j} - \boldsymbol{x}_{j,\tau}) \\
&\geq \frac{\eta^u_0}{n} \left( \frac{1}{2} \frac{\alpha^2 \rho^2 d}{2} - \frac{3\rho^2 d}{2} - \alpha \rho^2 \sqrt{d \log\left(\frac{10n^2}{\delta}\right)} - 2(n-1) M_t \rho^2 \alpha^2 \sqrt{d \log\left(\frac{10n^2}{\delta}\right)} \right) \\
&\geq \frac{\eta^u_0}{n} \frac{\alpha^2 \rho^2 d}{16},
\end{aligned}
$$

since $\alpha \geq 5$, $d \geq (16n)^2 \log\left(\frac{10n^2}{\delta}\right)$. Also, $\|\nabla_{\boldsymbol{u}} \widehat{L}(\boldsymbol{\theta}_0)\| \leq \max_{i \in [n]} \|\boldsymbol{X}_i\|_{2,\infty} \leq \alpha \rho \sqrt{1.5d}$. Using these, we get

$$
\begin{aligned}
(\gamma^1_{j,\mathrm{opt}_j} - \gamma^1_j) &\geq \frac{\frac{\eta}{16n} \alpha^2 \rho^2 \sqrt{d} 16n \sqrt{\log\left(\frac{10n^2}{\delta}\right)}}{\alpha \rho \sqrt{1.5d}} \\
&= \frac{\eta \alpha \rho}{\sqrt{1.5}} \sqrt{\log\left(\frac{10n^2}{\delta}\right)} \geq \eta \omega_2,
\end{aligned}
$$

when $n > 1$. Next, we have

$$
\begin{aligned}
\frac{\varphi^1_{j,\mathrm{opt}_j} / \varphi^1_{j,\tau}}{\varphi^0_{j,\mathrm{opt}_j} / \varphi^0_{j,\tau}} &= \exp\left( (\boldsymbol{x}_{j,\mathrm{opt}_j} - \boldsymbol{x}_{j,\tau})^\top (\boldsymbol{W}_1 - \boldsymbol{W}_0) \boldsymbol{x}_{j,1} \right) \\
&= \exp\left( (\boldsymbol{x}_{j,\mathrm{opt}_j} - \boldsymbol{x}_{j,\tau})^\top (-\nabla_{\boldsymbol{W}} \widehat{L}(\boldsymbol{\theta}_0)) \boldsymbol{x}_{j,1} \right) = 1,
\end{aligned}
$$

since $\boldsymbol{u}_0 = \boldsymbol{0}$. Therefore, $\varphi^1_{j,\mathrm{opt}_j} = \varphi^0_{j,\mathrm{opt}_j} \geq \frac{1}{2}$.

Further, since $\eta \leq \frac{\log(2)}{3\alpha\rho\sqrt{1.5d}}$, we have

$$
\begin{aligned}
\max_{i,j \in [n]} \frac{\ell_{1,i}}{\ell_{1,j}} &\leq \max_i \exp(2|\boldsymbol{u}_1^\top \boldsymbol{X}_i \top \boldsymbol{\varphi}^1_i|) \leq \exp(2\|\boldsymbol{u}_1\| \alpha \rho \sqrt{1.5d}) \\
&\leq \exp(6\eta \alpha \rho \sqrt{1.5d}) \leq \exp(2\log(2)) = 4 \leq C.
\end{aligned}
$$

Next, we assume that these are true at iteration $t$. Let $M_t := \max_i \ell_{t,i}$ and $m_t := \min_i \ell_{t,i}$. We will first show that for any $j \in [n]$,

$$
\begin{aligned}
\Delta \gamma^{t+1}_j &:= (\gamma^{t+1}_{j,\mathrm{opt}_j} - \gamma^{t+1}_j) - (\gamma^t_{j,\mathrm{opt}_j} - \gamma^t_j) \\
&= -\eta^u_t (-\nabla_{\boldsymbol{u}} \widehat{L}(\boldsymbol{\theta}_t))^\top (\boldsymbol{x}_{j,\mathrm{opt}_j} - \boldsymbol{x}_{j,\tau}) \geq \eta (t+1)^{-1/3} \omega_2.
\end{aligned}
\tag{40}
$$

We have

$$
\begin{aligned}
n(-\nabla_{\boldsymbol{u}}\widehat{L}(\boldsymbol{\theta}_t))^\top(\boldsymbol{x}_{j,\mathrm{opt}_j} - \boldsymbol{x}_{j,\tau}) &= \sum_{i=1}^{n}(\ell_{t,i} y_i \boldsymbol{X}_i^\top \boldsymbol{\varphi}(\boldsymbol{a}_i(\boldsymbol{W}_t)))^\top(\boldsymbol{x}_{j,\mathrm{opt}_j} - \boldsymbol{x}_{j,\tau}) \\
&= \ell_{t,j}(\varphi_{j,\mathrm{opt}_j}^t \boldsymbol{x}_{j,\mathrm{opt}_j} + (1 - \varphi_{j,\mathrm{opt}_j}^t)\boldsymbol{x}_{j,\tau})^\top(\boldsymbol{x}_{j,\mathrm{opt}_j} - \boldsymbol{x}_{j,\tau}) \\
&\quad + \sum_{i\neq j}\ell_{t,i} y_i y_j(\varphi_{i,\mathrm{opt}_i}^t \boldsymbol{x}_{i,\mathrm{opt}_i} + (1 - \varphi_{i,\mathrm{opt}_i}^t)\boldsymbol{x}_{i,\tau})^\top(\boldsymbol{x}_{j,\mathrm{opt}_j} - \boldsymbol{x}_{j,\tau}) \\
&\geq m_t\left(\frac{1}{2}\frac{\alpha^2\rho^2 d}{2} - \frac{3\rho^2 d}{2} - \alpha\rho^2\sqrt{d\log\left(\frac{10n^2}{\delta}\right)}\right) - 2(n-1)M_t\rho^2\alpha^2\sqrt{d\log\left(\frac{10n^2}{\delta}\right)} \\
&= 2\rho^2\alpha^2 M_t\sqrt{d\log\left(\frac{10n^2}{\delta}\right)} - 2n\rho^2\alpha^2 m_t\sqrt{d\log\left(\frac{10n^2}{\delta}\right)}\left(\frac{M_t}{m_t} - \frac{\frac{\alpha^2\rho^2 d}{4} - \frac{3\rho^2 d}{2} - \alpha\rho^2\sqrt{d\log\left(\frac{10n^2}{\delta}\right)}}{2n\rho^2\alpha^2\sqrt{d\log\left(\frac{10n^2}{\delta}\right)}}\right) \\
&\geq 2\rho^2\alpha^2\widehat{L}(\boldsymbol{\theta}_t)\sqrt{d\log\left(\frac{10n^2}{\delta}\right)} - 2n\rho^2\alpha^2 m_t\sqrt{d\log\left(\frac{10n^2}{\delta}\right)}\left(\frac{M_t}{m_t} - \frac{1}{16n}\sqrt{\frac{d}{\log\left(\frac{10n^2}{\delta}\right)}}\right) \\
&\geq 2\rho^2\alpha^2\widehat{L}(\boldsymbol{\theta}_t)\sqrt{d\log\left(\frac{10n^2}{\delta}\right)},
\end{aligned}
\tag{41}
$$

where we use the definitions of $M_t$, $m_t$, and the first inequality follows by using Lemma 9 and the second part of the IH, the second inequality follows since $M_t \geq \widehat{L}(\boldsymbol{\theta}_t)$, $\alpha \geq 6$ and $d \geq 4\log\left(\frac{10n^2}{\delta}\right)$, and the final inequality follows by using the third part of the IH and that $d \geq (16Cn)^2\log\left(\frac{10n^2}{\delta}\right)$.

Next, we have $\|\nabla_{\boldsymbol{u}}\widehat{L}(\boldsymbol{\theta}_t)\| \leq \max_{i\in[n]}\|\boldsymbol{X}_i\|_{2,\infty}\widehat{L}(\boldsymbol{\theta}_t) \leq \alpha\rho\sqrt{1.5d}\widehat{L}(\boldsymbol{\theta}_t)$. Combining this with Eq. (41), we get

$$
\frac{-\nabla_{\boldsymbol{u}}\widehat{L}(\boldsymbol{\theta}_t)^\top(\boldsymbol{x}_{j,\mathrm{opt}_j} - \boldsymbol{x}_{j,\tau})}{\|\nabla_{\boldsymbol{u}}\widehat{L}(\boldsymbol{\theta}_t)\|} \geq \frac{2\rho\alpha}{\sqrt{1.5}n}\sqrt{\log\left(\frac{10n^2}{\delta}\right)} \geq \omega_1,
$$

which implies Eq. (40). Using Eq. (40) and the first part of the IH, it follows that

$$
\gamma_{j,\mathrm{opt}_j}^{t+1} - \gamma_j^{t+1} \geq \gamma_{j,\mathrm{opt}_j}^t - \gamma_j^t + \eta_t^u\omega_1 \geq \eta\omega_2\sum_{t'=0}^{t}(t'+1)^{-2/3}.
$$

Next, we will show that for any $j \in [n]$, $\varphi_{j,\mathrm{opt}_j}^{t+1} \geq \frac{1}{2}$. We consider two cases:

**Case 1:** $\varphi_{j,\mathrm{opt}_j}^t \geq \frac{3}{4}$. In this case, we can directly use the steps in Scenario 1 of the proof of Lemma 3, and get that $\varphi_{j,\mathrm{opt}_j}^{t+1} \geq \frac{1}{2}$ since $\eta \leq \frac{\log(3)}{3\alpha^2\rho^2 d}$.

**Case 2:** $\varphi_{j,\mathrm{opt}_j}^t \leq \frac{3}{4}$. Using Lemma 10, we have

$$
\gamma_{i,\mathrm{opt}_i}^t - \gamma_i^t \leq 2\|\boldsymbol{u}_t\|\max_i\|\boldsymbol{X}_i\|_{2,\infty} \leq 6\eta\sum_{t'=0}^{t-1}(1+t')^{-2/3}\alpha\rho\sqrt{1.5d}.
\tag{42}
$$

Next, using similar calculations as Eqs. (18) and (19), for any $j \in [n]$, we have

$$(\boldsymbol{x}_{j,\mathsf{opt}_j} - \boldsymbol{x}_{j,\tau})^\top (-\nabla_{\boldsymbol{W}} \widehat{L}(\boldsymbol{\theta}_t)) \boldsymbol{x}_{j,1}$$

$$\geq \frac{\ell_{t,j}}{n} (\gamma_{j,\mathsf{opt}_j}^t - \gamma_j^t) \varphi_{j,\mathsf{opt}_j}^t (1 - \varphi_{j,\mathsf{opt}_j}^t) \|\boldsymbol{x}_{j,1}\|^2 \|\boldsymbol{x}_{j,\mathsf{opt}_j} - \boldsymbol{x}_{j,\tau}\|^2$$

$$- \frac{(n-1)}{n} \max_{i \neq j} \ell_{t,i} (\gamma_{i,\mathsf{opt}_i}^t - \gamma_i^t) \varphi_{i,\mathsf{opt}_i}^t (1 - \varphi_{i,\mathsf{opt}_i}^t) |\boldsymbol{x}_{i,1}^\top \boldsymbol{x}_{j,1}| |(\boldsymbol{x}_{j,\mathsf{opt}_j} - \boldsymbol{x}_{j,\tau})^\top (\boldsymbol{x}_{i,\mathsf{opt}_i} - \boldsymbol{x}_{i,\tau'})|$$

$$\geq \frac{m_t}{n} (\eta \omega_2 \sum_{t'=0}^{t-1} (t'+1)^{-2/3}) \frac{3}{16} \frac{\rho^2 d}{2} \left( \frac{\alpha^2 \rho^2 d}{2} + \frac{\rho^2 d}{2} - 2\alpha \rho^2 \sqrt{d \log \left( \frac{10n^2}{\delta} \right)} \right)$$

$$- \frac{(n-1)}{n} M_t (7.5 \eta \alpha \rho \sqrt{d} \sum_{t'=0}^{t-1} (t'+1)^{-2/3}) \alpha^2 \rho^2 \sqrt{d \log \left( \frac{10n^2}{\delta} \right)} (\alpha^2 + 2\alpha + 1) \rho^2 \sqrt{d \log \left( \frac{10n^2}{\delta} \right)}$$

$$\geq \frac{\alpha^3 \rho^5 d^{3/2}}{n} (\eta \sum_{t'=0}^{t-1} (t'+1)^{-2/3}) \left( \frac{3}{2^6 n} \sqrt{d \log \left( \frac{10n^2}{\delta} \right)} m_t - 32 \alpha^2 (n-1) M_t \log \left( \frac{10n^2}{\delta} \right) \right)$$

$$\geq \frac{32 \alpha^5 \rho^5 d^{3/2}}{n} (\eta \sum_{t'=0}^{t-1} (t'+1)^{-2/3}) \log \left( \frac{10n^2}{\delta} \right) \left( \widehat{L}(\boldsymbol{\theta}_t) + nm_t \left( \frac{1}{2^{10} \alpha^2 n^2} \sqrt{\frac{d}{\log \left( \frac{10n^2}{\delta} \right)}} - \frac{M_t}{m_t} \right) \right)$$

$$\geq \frac{32 \alpha^5 \rho^5 d^{3/2}}{n} (\eta \sum_{t'=0}^{t-1} (t'+1)^{-2/3}) \log \left( \frac{10n^2}{\delta} \right) \widehat{L}(\boldsymbol{\theta}_t), \tag{43}$$

where we use the definitions of $M_t$ and $m_t$, and the second inequality follows by using Lemma 9, the first two parts of the IH, and the final inequality uses the third part of the IH and follows when $d \geq (2^{10} C \alpha^2 n^2)^2 \log \left( \frac{10n^2}{\delta} \right)$. This implies that

$$\frac{\varphi_{j,\mathsf{opt}_j}^{t+1} / \varphi_{j,\tau}^{t+1}}{\varphi_{j,\mathsf{opt}_j}^t / \varphi_{j,\tau}^t} = \exp \left( (\boldsymbol{x}_{j,\mathsf{opt}_j} - \boldsymbol{x}_{j,\tau})^\top (\boldsymbol{W}_{t+1} - \boldsymbol{W}_t) \boldsymbol{x}_{j,1} \right) \geq 1.$$

Therefore, $\varphi_{j,\mathsf{opt}_j}^{t+1} \geq \varphi_{j,\mathsf{opt}_j}^t \geq \frac{1}{2}$.

Finally, we will show that $A_{t+1}^{\max} := \max_{i,j \in [n]} \frac{\ell_{t+1,i}}{\ell_{t+1,j}} \leq C$. Let $\boldsymbol{\varphi}_i^t := \boldsymbol{\varphi}(\boldsymbol{a}_i(\boldsymbol{W}_t))$. For any $i \in [n]$, we have

$$\ell_{t+1,i} = \exp(-y_i \boldsymbol{u}_{t+1}^\top \boldsymbol{X}_i^\top \boldsymbol{\varphi}_i^{t+1})$$

$$= \exp(-y_i \boldsymbol{u}_{t+1}^\top \boldsymbol{X}_i^\top (\boldsymbol{\varphi}_i^{t+1} - \boldsymbol{\varphi}_i^t)) \exp(-y_i \boldsymbol{u}_t^\top \boldsymbol{X}_i^\top \boldsymbol{\varphi}_i^t) \exp(y_i (\eta_t^u \nabla_{\boldsymbol{u}} \widehat{L}(\boldsymbol{\theta}_t))^\top \boldsymbol{X}_i^\top \boldsymbol{\varphi}_i^t)$$

$$= \ell_{t,i} \exp(-y_i \boldsymbol{u}_{t+1}^\top \boldsymbol{X}_i^\top (\boldsymbol{\varphi}_i^{t+1} - \boldsymbol{\varphi}_i^t)) \exp \left( -\frac{\eta_t^u}{n} \sum_{j=1}^n \ell_{t,j} (y_j \boldsymbol{X}_j^\top \boldsymbol{\varphi}_j^t)^\top y_i \boldsymbol{X}_i^\top \boldsymbol{\varphi}_i^t \right), \tag{44}$$

where $S_{t,i} := \sum_{j=1}^n \ell_{t,j} (y_j \boldsymbol{X}_j^\top \boldsymbol{\varphi}_j^t)^\top y_i \boldsymbol{X}_i^\top \boldsymbol{\varphi}_i^t$ can be expanded as

$$S_{t,i} = \ell_{t,i} \|\varphi_{i,\mathsf{opt}_i}^t \boldsymbol{x}_{i,\mathsf{opt}_i} + (1 - \varphi_{i,\mathsf{opt}_i}^t) \boldsymbol{x}_{i,\tau}\|^2$$

$$+ \sum_{j \neq i} \ell_{t,j} y_i y_j (\varphi_{j,\mathsf{opt}_j}^t \boldsymbol{x}_{j,\mathsf{opt}_j} + (1 - \varphi_{j,\mathsf{opt}_j}^t) \boldsymbol{x}_{j,\tau'})^\top (\varphi_{i,\mathsf{opt}_i}^t \boldsymbol{x}_{i,\mathsf{opt}_i} + (1 - \varphi_{i,\mathsf{opt}_i}^t) \boldsymbol{x}_{i,\tau}). \tag{45}$$

Using Lemma 9 and the second part of the IH, we have

$$\|\varphi_{i,\mathsf{opt}_i}^t \boldsymbol{x}_{i,\mathsf{opt}_i} + (1 - \varphi_{i,\mathsf{opt}_i}^t) \boldsymbol{x}_{i,\tau}\|^2 \leq \frac{3\alpha^2 \rho^2 d}{2}$$

$$\|\varphi_{i,\mathsf{opt}_i}^t \boldsymbol{x}_{i,\mathsf{opt}_i} + (1 - \varphi_{i,\mathsf{opt}_i}^t) \boldsymbol{x}_{i,\tau}\|^2 \geq \frac{1}{2^2} \frac{\alpha^2 \rho^2 d}{2} + \frac{1^2}{2^2} \frac{\rho^2 d}{2} - 2 \frac{1}{4} \alpha \rho^2 \sqrt{d \log \left( \frac{10n^2}{\delta} \right)} \geq \frac{\alpha^2 \rho^2 d}{8},$$

$$|(\varphi_{j,\mathsf{opt}_j}^t \boldsymbol{x}_{j,\mathsf{opt}_j} + (1 - \varphi_{j,\mathsf{opt}_j}^t) \boldsymbol{x}_{j,\tau'})^\top (\varphi_{i,\mathsf{opt}_i}^t \boldsymbol{x}_{i,\mathsf{opt}_i} + (1 - \varphi_{i,\mathsf{opt}_i}^t) \boldsymbol{x}_{i,\tau})| \leq \alpha^2 \rho^2 \sqrt{d \log \left( \frac{10n^2}{\delta} \right)}.$$

Using these in Eq. (45), we get

$$S_{t,i} \le \frac{3\alpha^2\rho^2 d}{2}\ell_{t,i} + n\alpha^2\rho^2\sqrt{d\log\left(\frac{10n^2}{\delta}\right)}M_t, \quad S_{t,i} \ge \frac{\alpha^2\rho^2 d}{8}\ell_{t,i} - n\alpha^2\rho^2\sqrt{d\log\left(\frac{10n^2}{\delta}\right)}M_t. \tag{46}$$

Next, for the second term, using similar calculations as Eq. (53) and Lemma 10, we have

$$\begin{aligned}
R_{t+1,i} &:= -y_i \boldsymbol{u}_{t+1}^\top \boldsymbol{X}_i^\top (\boldsymbol{\varphi}_i^{t+1} - \boldsymbol{\varphi}_i^t) \le |\boldsymbol{u}_{t+1}^\top \boldsymbol{X}_i^\top (\boldsymbol{\varphi}_i^{t+1} - \boldsymbol{\varphi}_i^t)| \\
&\le 2\|\boldsymbol{u}_{t+1}\| \max_i \|\boldsymbol{X}_i\|_{2,\infty}^3 \|\boldsymbol{W}_{t+1} - \boldsymbol{W}_t\| \\
&\le 6\eta(\alpha\rho\sqrt{1.5d})^3(t+1)^{1/3}\eta(t+1)^{-1} = 2\eta^2(\alpha\rho\sqrt{1.5d})^3(t+1)^{-2/3}. \tag{47}
\end{aligned}$$

Consider the loss ratio for any two samples $i,j \in [n]$, $A_t := \frac{\ell_{t,i}}{\ell_{t,j}}$. Using Eqs. (44) and (47), we have

$$A_{t+1} = A_t \frac{\exp(R_{t+1,i})}{\exp(R_{t+1,j})} \frac{\exp(-\frac{\eta_t^u}{n}S_{t,i})}{\exp(-\frac{\eta_t^u}{n}S_{t,j})} \le A_t \exp(12\eta^2(\alpha\rho\sqrt{1.5d})^3(t+1)^{-2/3}) \frac{\exp(-\frac{\eta_t^u}{n}S_{t,i})}{\exp(-\frac{\eta_t^u}{n}S_{t,j})} \tag{48}$$

We consider two cases:

**Case 1:** $A_t < C/2$. In this case, we have

$$\frac{\eta_t^u}{n}S_{t,i} \le \eta(t+1)^{-2/3}\max_i\|\boldsymbol{X}_i\|_{2,\infty} \le \eta\alpha\rho\sqrt{1.5d}(t+1)^{-2/3}.$$

Using this in Eq. (48), and that $\eta \le \frac{\log(2)}{3\alpha\rho\sqrt{1.5d}}$, $\eta \le \frac{1}{18\alpha^2\rho^2 d}$, we get

$$\begin{aligned}
A_{t+1} &\le A_t \exp(12\eta^2(\alpha\rho\sqrt{1.5d})^3)\exp(2\eta\alpha\rho\sqrt{1.5d}) \\
&\le A_t \exp\left(\frac{\log(2)}{3\alpha\rho\sqrt{1.5d}}\frac{1}{18\alpha^2\rho^2 d}(18\alpha^3\rho^3 d\sqrt{1.5d})\right)\exp(2\log(2)/3) \\
&= 2A_t \le C.
\end{aligned}$$

**Case 2:** $A_t \ge C/2$.

In this case, using Eq. (46), we have

$$\begin{aligned}
\frac{\exp(-\frac{\eta_t^u}{n}S_{t,i})}{\exp(-\frac{\eta_t^u}{n}S_{t,j})} &\le \exp\left(-\frac{\eta_t^u}{n}\left(\frac{\alpha^2\rho^2 d}{8}\ell_{t,i} - \frac{3\alpha^2\rho^2 d}{2}\ell_{t,j} - 2n\alpha^2\rho^2\sqrt{d\log\left(\frac{10n^2}{\delta}\right)}M_t\right)\right) \\
&\le \exp\left(-\frac{\eta_t^u\alpha^2\rho^2\ell_{t,j}}{n}\left(\frac{d}{8}A_t - \frac{3d}{2} - 2n\sqrt{d\log\left(\frac{10n^2}{\delta}\right)}A_t^{\max}\right)\right).
\end{aligned}$$

Further,

$$\begin{aligned}
\frac{\eta_t^u\alpha^2\rho^2\ell_{t,j}}{n}\left(\frac{d}{8}A_t - \frac{3d}{2} - 2n\sqrt{d\log\left(\frac{10n^2}{\delta}\right)}A_t^{\max}\right) &\ge \frac{\eta_t^u\alpha^2\rho^2 m_t}{n}\left(\frac{d}{8}C - \frac{3d}{2} - 2n\sqrt{d\log\left(\frac{10n^2}{\delta}\right)}C\right) \\
&\ge \frac{\eta_t^u\alpha^2\rho^2 m_t}{n}\frac{C\sqrt{d}}{16}\left(\sqrt{d} - 32n\sqrt{\log\left(\frac{10n^2}{\delta}\right)}\right) \\
&\ge \frac{\eta_t^u\alpha^2\rho^2\sqrt{d}M_t}{32n} \ge \frac{\eta\alpha^2\rho^2\sqrt{d}}{32n}(t+1)^{-2/3}\frac{\widehat{L}(\boldsymbol{\theta}_t)}{\alpha\rho\sqrt{1.5d}\widehat{L}(\boldsymbol{\theta}_t)} \\
&= \frac{\eta\alpha\rho}{40n}(t+1)^{-2/3},
\end{aligned}$$

as $C \geq 24$, $d \geq (32n)^2 \log\left(\frac{10n^2}{\delta}\right)$ and $\|\nabla_{\boldsymbol{u}}\widehat{L}(\boldsymbol{\theta}_t)\| \leq \alpha\rho\sqrt{1.5d}\widehat{L}(\boldsymbol{\theta}_t)$. Further, since $\eta \leq \frac{\alpha\rho}{160n}$, we have

$$\exp(4\eta^2(t+1)^{-2/3})\exp\left(-\frac{\eta\alpha\rho}{40n}(t+1)^{-2/3}\right) \leq 1,$$

which gives $A_{t+1} \leq A_t \leq C$.

$\square$

**Lemma 12.** *Let* $\tilde{\boldsymbol{u}} := \sum\limits_{i\in[n]} y_i\boldsymbol{x}_{i,\boldsymbol{opt}_i}$. *Under Condition* 1 *and the data model DM, for any* $j \in [n]$, $\tau \neq \boldsymbol{opt}_j$,

$$\left\langle y_j(\boldsymbol{x}_{j,\boldsymbol{opt}_j} + \boldsymbol{x}_{j,\tau}), \frac{\tilde{\boldsymbol{u}}}{\|\tilde{\boldsymbol{u}}\|}\right\rangle \geq \frac{\alpha\rho}{4}\sqrt{\frac{d}{n}}.$$

*Proof.* Using Lemma 9, we have

$$y_j(\boldsymbol{x}_{j,\mathrm{opt}_j} + \boldsymbol{x}_{j,\tau})^\top\left(\sum_{i\in[n]} y_i\boldsymbol{x}_{i,\mathrm{opt}_i}\right) = \|\boldsymbol{x}_{j,\mathrm{opt}_j}\|^2 + y_j\sum_{i\neq j\in[n]} y_i\boldsymbol{x}_{i,\mathrm{opt}_i}^\top\boldsymbol{x}_{j,\mathrm{opt}_j} + y_j\sum_{i\in[n]} y_i\boldsymbol{x}_{i,\mathrm{opt}_i}^\top\boldsymbol{x}_{j,\tau}$$

$$\geq 0.5\alpha^2\rho^2 d - 2(n-1)\alpha^2\rho^2\sqrt{d\log\left(\frac{10n^2}{\delta}\right)} - 2n\alpha\rho^2\sqrt{d\log\left(\frac{10n^2}{\delta}\right)}$$

$$\geq 0.5\alpha^2\rho^2 d - 4\alpha^2\rho^2\sqrt{d\log\left(\frac{10n^2}{\delta}\right)}$$

$$\geq \frac{\alpha^2\rho^2 d}{3},$$

since $d \geq (12n)^2 \log\left(\frac{10n^2}{\delta}\right)$ Similarly, we also have

$$\|\tilde{\boldsymbol{u}}\| \leq \sqrt{n\max_i\|\boldsymbol{x}_{i,\mathrm{opt}_i}\|^2 + n^2\max_{i\neq j}|\langle\boldsymbol{x}_{i,\mathrm{opt}_i}, \boldsymbol{x}_{j,\mathrm{opt}_j}\rangle|}$$

$$\leq \sqrt{n(1.5\alpha^2\rho^2 d) + 2n^2\alpha^2\rho^2\sqrt{d\log\left(\frac{10n^2}{\delta}\right)}}$$

$$\leq \alpha\rho\sqrt{1.75nd},$$

since $d \geq (8n)^2 \log\left(\frac{10n^2}{\delta}\right)$. Combining these and using $3\sqrt{1.75} \leq 4$ finishes the proof.

$\square$

**Lemma 13.** *Under Condition* 1 *and the data model DM, using the updates in Eq.* (10), *exponential loss has the following properties:*

- $\|\nabla_{\boldsymbol{\theta}}\widehat{L}(\boldsymbol{\theta}_t)\| \geq \|\nabla_{\boldsymbol{u}}\widehat{L}(\boldsymbol{\theta}_t)\| \geq \omega_1\widehat{L}(\boldsymbol{\theta}_t)$,

- $\max\limits_{\boldsymbol{\theta}'\in[\boldsymbol{\theta}_t,\boldsymbol{\theta}_{t+1}]} \widehat{L}(\boldsymbol{\theta}') \leq 8\widehat{L}(\boldsymbol{\theta}_t)$,

- $\max\limits_{\boldsymbol{\theta}'\in[\boldsymbol{\theta}_t,\boldsymbol{\theta}_{t+1}]} \|\nabla_{\boldsymbol{\theta}}^2\widehat{L}(\boldsymbol{\theta}')\| \leq 8(\omega(\boldsymbol{\theta}_t) \vee \omega(\boldsymbol{\theta}_{t+1}))\widehat{L}(\boldsymbol{\theta}_t)$,

*where* $\omega_1 = \omega_1(\alpha, \rho, n, d) = \frac{B}{8\sqrt{1.5n}}$, $\omega(\boldsymbol{\theta}_t) := 13(B \vee 1)^5(B \vee d)(\|\boldsymbol{u}_t\| \vee 1)^2$, $B = \alpha\rho\sqrt{1.5d}$.

**Remark 3.** *We note that the first property is the most challenging to show in the analysis and it yields a PL-inequality-like form, e.g. for neural networks (Frei & Gu, 2021; Nguyen & Mondelli, 2020; Liu et al., 2022). Analogous properties but for simpler settings, e.g. to obtain loss convergence results for linear predictors using NGD on separable data have been shown in (Nacson et al., 2019). The second point is an analog of the log-Lipschitzness property shown in (Taheri & Thrampoulidis, 2023a) for the analysis of two-layer neural networks trained with NGD and controls the loss on a line between current iterate* $\boldsymbol{\theta}_t$ *and the next one* $\boldsymbol{\theta}_{t+1}$. *The third property of second-order self-boundedness has been seen previously in the convergence analysis of multi-head self-attention (Deora et al., 2023) and MLPs (Taheri & Thrampoulidis, 2023b) trained with GD.*

*Proof.* We first obtain the lower bound on the gradient norm as follows.

$$\|\nabla_{\boldsymbol{\theta}}\widehat{L}(\boldsymbol{\theta}_t)\| = \sup_{\boldsymbol{v}:\|\boldsymbol{v}\|=1} \left\langle \frac{1}{n}\sum_{i=1}^{n}|\ell'_{i,t}|y_i\nabla_{\boldsymbol{\theta}}\Phi(\boldsymbol{\theta}_t,\boldsymbol{X}_i),\boldsymbol{v}\right\rangle$$

$$\geq \|\nabla_{\boldsymbol{u}}\widehat{L}(\boldsymbol{\theta}_t)\| = \sup_{\boldsymbol{v}:\|\boldsymbol{v}\|=1} \left\langle \frac{1}{n}\sum_{i=1}^{n}|\ell'_{i,t}|y_i\nabla_{\boldsymbol{u}}\Phi(\boldsymbol{\theta}_t,\boldsymbol{X}_i),\boldsymbol{v}\right\rangle$$

$$\geq \widehat{L}(\boldsymbol{\theta}_t)\sup_{\boldsymbol{v}:\|\boldsymbol{v}\|=1}\min_i y_i\nabla_{\boldsymbol{u}}\Phi(\boldsymbol{\theta}_t,\boldsymbol{X}_i)^{\top}\boldsymbol{v}$$

$$\geq \widehat{L}(\boldsymbol{\theta}_t)\min_i\left\langle y_i\nabla_{\boldsymbol{u}}\Phi(\boldsymbol{\theta}_t,\boldsymbol{X}_i),\frac{\tilde{\boldsymbol{u}}}{\|\tilde{\boldsymbol{u}}\|}\right\rangle, \tag{49}$$

where $\tilde{\boldsymbol{u}} = \sum\limits_{i\in[n]} y_i\boldsymbol{x}_{i,\mathrm{opt}_i}$. Using Lemma 11 and 12,

$$\min_i\left\langle y_i\nabla_{\boldsymbol{u}}\Phi(\boldsymbol{\theta}_t,\boldsymbol{X}_i),\frac{\tilde{\boldsymbol{u}}}{\|\tilde{\boldsymbol{u}}\|}\right\rangle = \min_i y_i(\varphi^t_{i,\mathrm{opt}_i}\boldsymbol{x}_{i,\mathrm{opt}_i}+(1-\varphi^t_{i,\mathrm{opt}_i})\boldsymbol{x}_{i,\tau})^{\top}\overline{\tilde{\boldsymbol{u}}}$$

$$\geq 0.5\min_i y_i(\boldsymbol{x}_{i,\mathrm{opt}_i}+\boldsymbol{x}_{i,\tau})^{\top}\overline{\tilde{\boldsymbol{u}}} \geq \frac{\alpha\rho}{8}\sqrt{\frac{d}{n}} = \omega_1. \tag{50}$$

Using Eq. (50) in Eq. (49), we get the first bullet point.

We show the second bullet point as follows. For any $\boldsymbol{\theta},\boldsymbol{\theta}'$, we have

$$|y\Phi(\boldsymbol{\theta},\boldsymbol{X})-y\Phi(\boldsymbol{\theta}',\boldsymbol{X})| = |\boldsymbol{u}^{\top}\boldsymbol{X}^{\top}\boldsymbol{\varphi}(\boldsymbol{X}\boldsymbol{W}^{\top}\boldsymbol{x}_1)-\boldsymbol{u}'^{\top}\boldsymbol{X}^{\top}\boldsymbol{\varphi}(\boldsymbol{X}\boldsymbol{W}'^{\top}\boldsymbol{x}_1)|$$

$$\leq |(\boldsymbol{u}-\boldsymbol{u}')^{\top}\boldsymbol{X}^{\top}\boldsymbol{\varphi}(\boldsymbol{X}\boldsymbol{W}^{\top}\boldsymbol{x}_1)|+|\boldsymbol{u}'^{\top}\boldsymbol{X}^{\top}(\boldsymbol{\varphi}(\boldsymbol{X}\boldsymbol{W}^{\top}\boldsymbol{x}_1)-\boldsymbol{\varphi}(\boldsymbol{X}\boldsymbol{W}'^{\top}\boldsymbol{x}_1))|. \tag{51}$$

These two terms can be bounded as follows.

$$|(\boldsymbol{u}-\boldsymbol{u}')^{\top}\boldsymbol{X}^{\top}\boldsymbol{\varphi}(\boldsymbol{X}\boldsymbol{W}^{\top}\boldsymbol{x}_1)| \leq \|\boldsymbol{u}-\boldsymbol{u}'\|\|\boldsymbol{X}^{\top}\|_{1,2} = \|\boldsymbol{X}\|_{2,\infty}\|\boldsymbol{u}-\boldsymbol{u}'\|, \tag{52}$$

$$\|\boldsymbol{X}^{\top}(\boldsymbol{\varphi}(\boldsymbol{X}\boldsymbol{W}^{\top}\boldsymbol{x}_1)-\boldsymbol{\varphi}(\boldsymbol{X}\boldsymbol{W}'^{\top}\boldsymbol{x}_1))\| \leq \|\boldsymbol{X}^{\top}\|_{1,2}\|\boldsymbol{\varphi}(\boldsymbol{X}\boldsymbol{W}^{\top}\boldsymbol{x}_1)-\boldsymbol{\varphi}(\boldsymbol{X}\boldsymbol{W}'^{\top}\boldsymbol{x}_1)\|_1$$

$$\leq 2\|\boldsymbol{X}\|_{2,\infty}\|\boldsymbol{X}\boldsymbol{W}^{\top}\boldsymbol{x}_1-\boldsymbol{X}\boldsymbol{W}'^{\top}\boldsymbol{x}_1\|_{\infty}$$

$$\leq 2\|\boldsymbol{X}\|_{2,\infty}^2\|(\boldsymbol{W}-\boldsymbol{W}')^{\top}\boldsymbol{x}_1\|$$

$$\leq 2\|\boldsymbol{X}\|_{2,\infty}^2\|\boldsymbol{x}_1\|\|\boldsymbol{W}-\boldsymbol{W}'\|$$

$$\leq 2\|\boldsymbol{X}\|_{2,\infty}^3\|\boldsymbol{W}-\boldsymbol{W}'\|. \tag{53}$$

Using Eqs. (52) and (53) in Eq. (51) for $\boldsymbol{\theta}_t$ and $\boldsymbol{\theta}_t+\lambda(\boldsymbol{\theta}_{t+1}-\boldsymbol{\theta}_t)$, we get

$$\max_{\lambda\in[0,1]}\frac{\ell(y\Phi(\boldsymbol{\theta}_t+\lambda(\boldsymbol{\theta}_{t+1}-\boldsymbol{\theta}_t),\boldsymbol{X}))}{\ell(y\Phi(\boldsymbol{\theta}_t,\boldsymbol{X}))}$$

$$\leq \max_{\lambda\in[0,1]}\exp(|y\Phi(\boldsymbol{\theta}_t+\lambda(\boldsymbol{\theta}_{t+1}-\boldsymbol{\theta}_t),\boldsymbol{X})-y\Phi(\boldsymbol{\theta}_t,\boldsymbol{X})|)$$

$$\leq \max_{\lambda\in[0,1]}\exp\left(2\lambda\|\boldsymbol{X}\|_{2,\infty}^3\|\boldsymbol{u}_t\|\|\boldsymbol{W}_{t+1}-\boldsymbol{W}_t\|+\lambda\|\boldsymbol{X}\|_{2,\infty}\|\boldsymbol{u}_{t+1}-\boldsymbol{u}_t\|\right)$$

$$\leq \exp\left(2\eta_t^W\|\nabla_{\boldsymbol{W}}\widehat{L}(\boldsymbol{\theta}_t)\|\|\boldsymbol{X}\|_{2,\infty}^3\|\boldsymbol{u}_t\|+\eta_t^u\|\nabla_{\boldsymbol{u}}\widehat{L}(\boldsymbol{\theta}_t)\|\|\boldsymbol{X}\|_{2,\infty}\right) \qquad \text{(using Eq. (2))}$$

$$\leq \exp\left(6\eta\|\boldsymbol{X}\|_{2,\infty}^3\frac{\eta t^{1/3}}{(1+t)}+\eta\|\boldsymbol{X}\|_{2,\infty}\right) \qquad \text{(using Lemma 10)}$$

$$\leq \exp\left(6\eta^2\|\boldsymbol{X}\|_{2,\infty}^3+\eta\|\boldsymbol{X}\|_{2,\infty}\right).$$

Since this is true for every $\boldsymbol{X}_i$, and $\eta \le \frac{\log(2)}{3\alpha\rho\sqrt{1.5d}} \wedge \frac{\log(3)}{3\alpha^2\rho^2 d}$, we get

$$\max_{\boldsymbol{\theta}' \in [\boldsymbol{\theta}_t, \boldsymbol{\theta}_{t+1}]} \widehat{L}(\boldsymbol{\theta}') \le \max_{i \in [n]} \exp\left(6\eta^2 \|\boldsymbol{X}_i\|_{2,\infty}^3 + \eta\|\boldsymbol{X}_i\|_{2,\infty}\right) \widehat{L}(\boldsymbol{\theta}_t)$$

$$= \exp\left(6\eta^2 B^3 + \eta B\right) \widehat{L}(\boldsymbol{\theta}_t) \le \exp\left(6\frac{\log(2)}{3}\frac{\log(3)}{2} + \frac{\log(2)}{3}\right) \widehat{L}(\boldsymbol{\theta}_t)$$

$$\le 8\widehat{L}(\boldsymbol{\theta}_t).$$

Next, we obtain the upper bound on the Hessian norm as follows. Using Prop. 3 from (Deora et al., 2023), we have

$$\|\nabla_{\boldsymbol{\theta}}\Phi(\boldsymbol{\theta}_t, \boldsymbol{X})\| \le \|\boldsymbol{X}\|_{2,\infty} + 2\|\boldsymbol{X}\|_{2,\infty}^2 \|\boldsymbol{X}\boldsymbol{u}_t\|_\infty,$$

$$\|\nabla_{\boldsymbol{\theta}}^2 \Phi(\boldsymbol{\theta}_t, \boldsymbol{X})\| \le 6d\|\boldsymbol{X}\|_{2,\infty}^2 \|\boldsymbol{X}\|_{1,\infty}^2 \|\boldsymbol{X}\boldsymbol{u}_t\|_\infty + 2\sqrt{d}\|\boldsymbol{X}\|_{2,\infty}^2 \|\boldsymbol{X}\|_{1,\infty}.$$

Using these, we get

$$\|\nabla_{\boldsymbol{\theta}}\Phi(\boldsymbol{\theta}_t, \boldsymbol{X}_i)\| \le \|\boldsymbol{X}_i\|_{2,\infty} + 2\|\boldsymbol{X}_i\|_{2,\infty}^3 \|\boldsymbol{u}_t\|,$$

$$\|\nabla_{\boldsymbol{\theta}}^2 \Phi(\boldsymbol{\theta}_t, \boldsymbol{X}_i)\| \le 6d\|\boldsymbol{X}_i\|_{2,\infty}^5 \|\boldsymbol{u}_t\| + 2\sqrt{d}\|\boldsymbol{X}_i\|_{2,\infty}^3,$$

$$\implies \|\nabla_{\boldsymbol{\theta}}\Phi(\boldsymbol{\theta}_t, \boldsymbol{X}_i)\|^2 + \|\nabla_{\boldsymbol{\theta}}^2 \Phi(\boldsymbol{\theta}_t, \boldsymbol{X}_i)\| \le \max_{i \in [n]} 4\|\boldsymbol{X}_i\|_{2,\infty}^6 \|\boldsymbol{u}_t\|^2 + 6d\|\boldsymbol{X}_i\|_{2,\infty}^5 \|\boldsymbol{u}_t\| + 2\sqrt{d}\|\boldsymbol{X}_i\|_{2,\infty}^3 + \|\boldsymbol{X}_i\|_{2,\infty}^2$$

$$= 4B^6\|\boldsymbol{u}_t\|^2 + 6dB^5\|\boldsymbol{u}_t\| + 2\sqrt{d}B^3 + B^2$$

$$\le 13(B \vee 1)^5 (B \vee d)(\|\boldsymbol{u}_t\| \vee 1)^2 =: \omega(\boldsymbol{\theta}_t).$$

Using this, we get

$$\max_{\boldsymbol{\theta}' \in [\boldsymbol{\theta}_t, \boldsymbol{\theta}_{t+1}]} \|\nabla_{\boldsymbol{\theta}}^2 \widehat{L}(\boldsymbol{\theta}')\| \le \max_{\boldsymbol{\theta}' \in [\boldsymbol{\theta}_t, \boldsymbol{\theta}_{t+1}]} \omega(\boldsymbol{\theta}') \widehat{L}(\boldsymbol{\theta}')$$

$$\le 8(\omega(\boldsymbol{\theta}_t) \vee \omega(\boldsymbol{\theta}_{t+1})) \widehat{L}(\boldsymbol{\theta}_t).$$

$\square$

## B.3  Proof of Theorem 2

We first restate Theorem 2, this time with the exact constants.

**Theorem 8** (Train loss convergence)**.** *Under Condition 1 and the data model DM, using the updates in Eq.* (10)*, for any $t > 0$,*

$$\widehat{L}(\boldsymbol{\theta}_{t+1}) \le \mathcal{O}\left(\exp\left(-\frac{\eta\omega_1}{2}(t+1)^{1/3}\right)\right),$$

*where $\omega_1 = \frac{\alpha\rho}{8}\sqrt{\frac{d}{n}}$.*

*Proof.* First, using Lemma 10, we have

$$\omega(\boldsymbol{\theta}_t) \vee \omega(\boldsymbol{\theta}_{t+1}) \le 13(B \vee 1)^5 (B \vee d)(\|\boldsymbol{u}_t\| \vee \|\boldsymbol{u}_{t+1}\| \vee 1)^2$$

$$\le 13(B \vee 1)^5 (B \vee d)(3\eta(t+1)^{1/3} \vee 1)^2$$

$$= 13(B \vee 1)^5 (B \vee d)(3\eta \vee (t+1)^{-1/3})^2 (t+1)^{2/3}$$

$$\le \omega_0 (t+1)^{2/3}, \tag{54}$$

where $\omega_0 = 13(B \vee 1)^5 (B \vee d)$, since $\eta \le 1/3$.

Next, for some $\boldsymbol{\theta}' \in [\boldsymbol{\theta}_t, \boldsymbol{\theta}_{t+1}]$, using the second order Taylor expansion of $\widehat{L}(\boldsymbol{\theta}_{t+1})$, we have

$$\widehat{L}(\boldsymbol{\theta}_{t+1}) = \widehat{L}(\boldsymbol{\theta}_t) + \langle \nabla_{\boldsymbol{\theta}} \widehat{L}(\boldsymbol{\theta}_t), \boldsymbol{\theta}_{t+1} - \boldsymbol{\theta}_t \rangle + \frac{1}{2}(\boldsymbol{\theta}_{t+1} - \boldsymbol{\theta}_t)^\top \nabla_{\boldsymbol{\theta}}^2 \widehat{L}(\boldsymbol{\theta}')(\boldsymbol{\theta}_{t+1} - \boldsymbol{\theta}_t)$$

$$\leq \widehat{L}(\boldsymbol{\theta}_t) + \langle \nabla_{\boldsymbol{\theta}} \widehat{L}(\boldsymbol{\theta}_t), \boldsymbol{\theta}_{t+1} - \boldsymbol{\theta}_t \rangle + \frac{1}{2}\|\boldsymbol{\theta}_{t+1} - \boldsymbol{\theta}_t\|^2 \max_{\boldsymbol{\theta}' \in [\boldsymbol{\theta}_t, \boldsymbol{\theta}_{t+1}]} \|\nabla_{\boldsymbol{\theta}}^2 \widehat{L}(\boldsymbol{\theta}')\|$$

$$\leq \widehat{L}(\boldsymbol{\theta}_t) - \eta_t^u \|\nabla_{\boldsymbol{u}} \widehat{L}(\boldsymbol{\theta}_t)\|^2 - \eta_t^W \|\nabla_{\boldsymbol{W}} \widehat{L}(\boldsymbol{\theta}_t)\|^2$$

$$+ \frac{1}{2}((\eta_t^u)^2 \|\nabla_{\boldsymbol{u}} \widehat{L}(\boldsymbol{\theta}_t)\|^2 + (\eta_t^W)^2 \|\nabla_{\boldsymbol{W}} \widehat{L}(\boldsymbol{\theta}_t)\|^2) \max_{\boldsymbol{\theta}' \in [\boldsymbol{\theta}_t, \boldsymbol{\theta}_{t+1}]} \|\nabla_{\boldsymbol{\theta}}^2 \widehat{L}(\boldsymbol{\theta}')\|$$

$$\leq \widehat{L}(\boldsymbol{\theta}_t) - \frac{\eta}{(t+1)^{2/3}} \|\nabla_{\boldsymbol{u}} \widehat{L}(\boldsymbol{\theta}_t)\| + \frac{1}{2}\left(\frac{\eta^2}{(t+1)^{4/3}} + \frac{\eta^2}{(t+1)^2}\right) \max_{\boldsymbol{\theta}' \in [\boldsymbol{\theta}_t, \boldsymbol{\theta}_{t+1}]} \|\nabla_{\boldsymbol{\theta}}^2 \widehat{L}(\boldsymbol{\theta}')\| \quad \text{(using Eq. (10))}$$

$$\leq \widehat{L}(\boldsymbol{\theta}_t) - \frac{\eta \omega_1}{(t+1)^{2/3}} \widehat{L}(\boldsymbol{\theta}_t) + \frac{8\eta^2}{(t+1)^{4/3}}(\omega(\boldsymbol{\theta}_t) \vee \omega(\boldsymbol{\theta}_{t+1})) \widehat{L}(\boldsymbol{\theta}_t) \quad \text{(using Lemma 13)}$$

$$\leq \left(1 - \frac{\eta \omega_1}{(t+1)^{2/3}} + \frac{8\eta^2 \omega_0 (t+1)^{2/3}}{(t+1)^{4/3}}\right) \widehat{L}(\boldsymbol{\theta}_t) \quad \text{(using Eq. (54))}$$

$$\leq \left(1 - \frac{\eta \omega_1}{2(t+1)^{2/3}}\right) \widehat{L}(\boldsymbol{\theta}_t) \leq \exp\left(-\frac{\eta \omega_1}{2}(t+1)^{-2/3}\right) \widehat{L}(\boldsymbol{\theta}_t) \quad \text{(using Condition 1)}$$

$$\leq \exp\left(-\sum_{t'=0}^{t} \frac{\eta \omega_1}{2}(t+1)^{-2/3}\right) \widehat{L}(\boldsymbol{\theta}_0) \quad \text{(by telescoping)}$$

$$\leq \exp\left(-\frac{\eta \omega_1}{2}(t+1)^{1/3}\right) \widehat{L}(\boldsymbol{\theta}_0).$$

$\square$

## B.4 Proofs of Theorems 3 and 4

We first restate Theorem 3, this time with the exact constants.

**Theorem 9** (IB Rate under Joint Training). *Under Condition 1 and the data model DM, using the updates in Eq. (10), for any $t \geq t_\epsilon := \exp\left(\frac{(10B\Lambda)^2}{\eta}\left(\frac{Cn^2\sqrt{2d}}{250\log\left(\frac{10n^2}{\delta}\right)} \vee 10B^2\Lambda\right)^{1/3} \epsilon^{-4/3}\right) \vee \exp\left(\frac{B^2\Lambda}{\eta}\right)$,*

$$\left\langle \frac{\boldsymbol{W}_t}{\|\boldsymbol{W}_t\|}, \frac{\boldsymbol{W}_{mm}}{\|\boldsymbol{W}_{mm}\|} \right\rangle \geq 1 - \epsilon - \frac{C(\eta, B, \Lambda, \epsilon)}{\log t},$$

*where $B = \alpha\rho\sqrt{1.5d}$, $C \geq 24$ is an absolute constant, and*

$$C(\eta, B, \Lambda, \epsilon) = \frac{2B^2\Lambda}{\eta}(1-\epsilon)\|\boldsymbol{W}_{t_\epsilon}\|\left(1 - (1-\epsilon)^{-1}\langle \overline{\boldsymbol{W}}_{t_\epsilon}, \overline{\boldsymbol{W}}_{mm}\rangle - 2\eta\|\boldsymbol{W}_{t_\epsilon}\|^{-1}\widehat{L}(\boldsymbol{\theta}_{t_\epsilon})\right).$$

*Proof.* In this case, Lemma 4 follows directly, using Lemma 11. Since $\|\boldsymbol{W}_0\| = 0$, using Eq. 10 and similar calculations as the proof of Lemma 5, for any $t > 0$,

$$\|\boldsymbol{W}_t\| \geq \eta(2B^2\Lambda)^{-1}\log t. \tag{55}$$

Next, using Lemma 11, we will show that for any $\epsilon \in (0, 1)$, there exists

$$R_\epsilon := 2\Lambda\epsilon^{-1}\left(\log\left(\frac{4Cn^2\sqrt{2d}}{\log\left(\frac{10n^2}{\delta}\right)}\epsilon^{-1}\right) \vee 5\log\left(40B^2\Lambda\epsilon^{-1}\right)\right), \tag{56}$$

such that for every $t$ where $\|\boldsymbol{W}_t\| \geq R_\epsilon$,

$$\left\langle -\nabla_{\boldsymbol{W}}\widehat{L}(\boldsymbol{\theta}_t), \frac{\boldsymbol{W}_{mm}}{\|\boldsymbol{W}_{mm}\|} \right\rangle \geq (1-\epsilon)\left\langle -\nabla_{\boldsymbol{W}}\widehat{L}(\boldsymbol{\theta}_t), \frac{\boldsymbol{W}_t}{\|\boldsymbol{W}_t\|} \right\rangle. \tag{57}$$

Using similar calculations as the proof of Lemma 7, in the first two cases, since $R' = R\|\boldsymbol{W}_{\mathrm{mm}}\|^{-1} \geq 10\log(40B^2\Lambda\epsilon^{-1})$, Eqs. (30) and (31) follow, respectively. Using Eq. (36), we can show that when

$$\exp(0.5\epsilon(1-\epsilon)^{-1}R') \geq 0.5\nu(n-1)(2)^2 \frac{\max_{i\in\mathcal{I}_3} \ell_{t,i}(\gamma_{i,\mathrm{opt}_i} - \gamma_i)}{\min_{i\in\mathcal{I}_1} \ell_{t,i}(\gamma_{i,\mathrm{opt}_i} - \gamma_i)}, \tag{58}$$

Eq. (29) is satisfied. Using Lemma 11, Eq. (58) is true when

$$\exp(0.5\epsilon(1-\epsilon)^{-1}R') \geq \frac{4Cn^2\sqrt{2d}}{\log\left(\frac{10n^2}{\delta}\right)}\epsilon^{-1},$$

which is satisfied by Eq. (56).

Next, using Eqs. (55) and (58), we can show that for any $t \geq t_\epsilon$, $\|\boldsymbol{W}_t\| \geq R_\epsilon \vee 1/2$. We have

$$\begin{aligned}
R_\epsilon &= 2\Lambda\epsilon^{-1}\left(\log\left(\frac{4Cn^2\sqrt{2d}}{\log\left(\frac{10n^2}{\delta}\right)}\epsilon^{-1}\right) \vee 5\log\left(40B^2\Lambda\epsilon^{-1}\right)\right) \\
&\leq 25\Lambda\left(\frac{Cn^2\sqrt{2d}}{250\log\left(\frac{10n^2}{\delta}\right)} \vee 10B^2\Lambda\right)^{1/3}\epsilon^{-4/3} \\
&\leq \eta(4B^2\Lambda)^{-1}\log t \leq \|\boldsymbol{W}_t\|.
\end{aligned}$$

In addition, $\|\boldsymbol{W}_t\| \geq 1/2$ for $t \geq \exp\left(\frac{B^2\Lambda}{\eta}\right)$. Combining these and using similar steps as the proof of Theorem 4 in (Tarzanagh et al., 2023a), we get

$$\begin{aligned}
\left\langle \frac{\boldsymbol{W}_t}{\|\boldsymbol{W}_t\|}, \frac{\boldsymbol{W}_{\mathrm{mm}}}{\|\boldsymbol{W}_{\mathrm{mm}}\|} \right\rangle &\geq 1 - \epsilon - \frac{\eta(2B^2\Lambda)^{-1}C(\eta,B,\epsilon)}{\|\boldsymbol{W}_t\|} \\
&\geq 1 - \epsilon - \frac{C(\eta,B,\epsilon)}{\log t},
\end{aligned}$$

where the last step follows by using Eq. (55). $\qquad\square$

Next, we restate Theorem 4 for convenience.

**Theorem 10** (IB Rate of $\boldsymbol{u}$). *Let* $\overline{\gamma} := \max_{\boldsymbol{u}:\|\boldsymbol{u}\|\leq 1} \min_i y_i\boldsymbol{u}^\top\boldsymbol{x}_{i,opt_i}$. *Under Condition 1 and the data model DM, using the updates in Eq. (10), for any* $t \geq t_\epsilon \vee \exp(C(\eta,B,\Lambda,\epsilon)(\epsilon^{-1} \vee (8B^2\Lambda)^4))$,

$$\min_i y_i\overline{\boldsymbol{u}_t}^\top\boldsymbol{x}_{i,opt_i} \geq \frac{\overline{\gamma}}{4} - \frac{1}{1 + \exp(\eta(8B^2\Lambda^2)^{-1}\log t)}.$$

*Proof.* First, by definition of $\overline{\gamma}$, we have

$$\|\nabla_{\boldsymbol{\theta}}\widehat{L}(\boldsymbol{\theta}_t)\| \geq \frac{\overline{\gamma}}{3}\widehat{L}(\boldsymbol{\theta}_t).$$

Then, following similar steps as the proof of Theorem 2, we have

$$\widehat{L}(\boldsymbol{\theta}_t) \leq \exp\left(-\frac{\eta\overline{\gamma}}{4}(t+1)^{1/3}\right)\widehat{L}(\boldsymbol{\theta}_0).$$

Using this, we have

$$\begin{aligned}
\min_i \exp(-y_i(\varphi_{i,\mathrm{opt}_i}\boldsymbol{x}_{i,\mathrm{opt}_i} + (1-\varphi_{i,\mathrm{opt}_i})\boldsymbol{x}_{i,\tau})^\top\boldsymbol{u}_t) &\leq \exp\left(-\frac{\eta\overline{\gamma}}{4}(t+1)^{1/3}\right), \\
\implies \min_i y_i(\varphi_{i,\mathrm{opt}_i}\boldsymbol{x}_{i,\mathrm{opt}_i} + (1-\varphi_{i,\mathrm{opt}_i})\boldsymbol{x}_{i,\tau})^\top\boldsymbol{u}_t &\geq \frac{\eta\overline{\gamma}}{4}(t+1)^{1/3}.
\end{aligned}$$

Then, using the proof of Lemma 10, we get

$$\min_i y_i(\varphi_{i,\mathrm{opt}_i}\boldsymbol{x}_{i,\mathrm{opt}_i} + (1 - \varphi_{i,\mathrm{opt}_i})\boldsymbol{x}_{i,\tau})^\top \overline{\boldsymbol{u}_t} \geq \frac{\overline{\gamma}}{4}$$

$$\implies \min_i y_i\varphi_{i,\mathrm{opt}_i}\boldsymbol{x}_{i,\mathrm{opt}_i}^\top \overline{\boldsymbol{u}_t} \geq \frac{\overline{\gamma}}{4} - \max_i(1 - \varphi_{i,\mathrm{opt}_i})\|\boldsymbol{x}_{i,\tau}\|$$

$$\geq \frac{\overline{\gamma}}{4} - \frac{\rho\sqrt{1.5d}}{1 + \exp(-\eta(8B^2\Lambda^2)^{-1}\log t)},$$

where for the last inequality, we use the following lower bound on the softmax scores,

$$\varphi_{i,\mathrm{opt}_i}^t \geq 1 - \frac{1}{1 + \exp(\eta(8B^2\Lambda^2)^{-1}\log t)}$$

which is obtained by using Theorem 3 and following similar steps as the proof of Lemma 1, as

$$(\boldsymbol{x}_{i,\tau} - \boldsymbol{x}_{i,\mathrm{opt}_i})^\top (\boldsymbol{W}_t - \overline{\boldsymbol{W}}_{\mathrm{mm}}\|\boldsymbol{W}_t\| + \overline{\boldsymbol{W}}_{\mathrm{mm}}\|\boldsymbol{W}_t\|)\boldsymbol{x}_{i,1} \leq 2B^2\|\boldsymbol{W}_t\|\|\overline{\boldsymbol{W}}_t - \overline{\boldsymbol{W}}_{\mathrm{mm}}\| - \frac{1}{\|\overline{\boldsymbol{W}}_{\mathrm{mm}}\|}\|\boldsymbol{W}_t\|$$

$$\leq 4B^2\sqrt{\epsilon \vee \frac{C(\eta, B, \Lambda, \epsilon)}{\log t}}(2\eta\log t) - \eta(4B^2\Lambda^2)^{-1}\log t$$

$$\leq -\eta(8B^2\Lambda^2)^{-1}\log t,$$

since $t \geq \exp(C(\eta, B, \Lambda, \epsilon)(\epsilon^{-1} \vee (8B^2\Lambda)^4))$.

Using $\varphi_{i,\mathrm{opt}_i} \leq 1$ then finishes the proof. □

## C  Additional Experiments and Settings

This section includes some additional experiments and details about the settings for the results included in this work. We use the PyTorch (Paszke et al., 2019) library for our code, which is licensed under the Modified BSD license.

**Fig. 1**  In this case, we use the MultiNLI (Williams et al., 2018) dataset, which contains sentence pairs belonging to one of three classes: entailment, neutral, contradiction. The task is to predict whether the second sentence entails, is neutral with, or contradicts the first sentence. It is released under the ODC-By license. We use the Hugging Face `pytorch-transformers` implementation of the BERT `bert-base-uncased` model, with pretrained weights (Devlin et al., 2019), released under Apache License 2.0. We use batch-size 32 to train all models. Learning rates are—Adam: $2e-5$, SGD: $1e-3$, SNGD: 0.01. The $\eta_{max}$ for SPS and SNGD is set to 0.1.

**Fig. 2**  In this case, the samples are generated following Example 1, with $U = 1$ and $\rho = 0.05$. We set $n = 20$, $d = 100$ and T = 6. We use a cap on the adaptive step-size, $\eta_{\max} = 100$ for NGD and NGD-mom, and $\eta_{\max} = 10$ for Polyak-step. The learning rate $\eta$ is set as 0.025 for NGD and NGD-mom, whereas for GD, it is set as 0.25. For NGD-mom, the momentum parameter is set to 0.9.

**Fig. 3**  In this case, the samples are generated based on the data model DM with $\rho = 0.1$ and $\alpha = 3$. We set $n = 10$, $d = 100$ and T = 2. The learning rate $\eta$ is set as 0.002 for NGD and NGD-joint, whereas for GD, it is set as 0.02.

**Fig. 4**  We use the CivilComments dataset (Borkan et al., 2019), which consists of online comments and the task is to classify if the comment is *toxic* or *non-toxic*. It is released under the CC BY-NC 4.0 license. We use the BERT `bert-base-uncased model`, with pretrained weights (Devlin et al., 2019) using the WILDS package (Koh et al., 2021). The learning rate for SGD is $10^{-3}$, $10^{-2}$ for NGD and $10^{-5}$ for Adam. The parameter $\eta_{max}$ is set to $10^{-2}$ for both SNGD and SPS. All models are trained with batch-size 32.

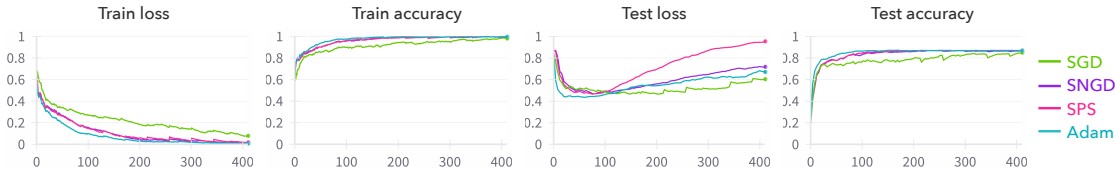

Figure 4: Comparison of train and test dynamics of various optimizers—SGD, SNGD, SPS, and Adam—while fine-tuning a pre-trained BERT model on the CivilComments dataset.

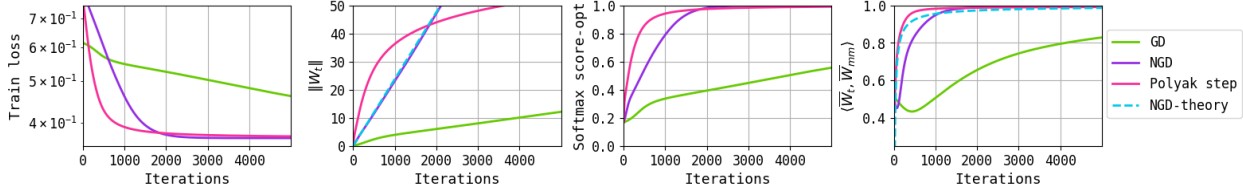

Figure 5: Training dynamics when optimizing only $W$ on synthetic data with antipodal `opt` tokens.

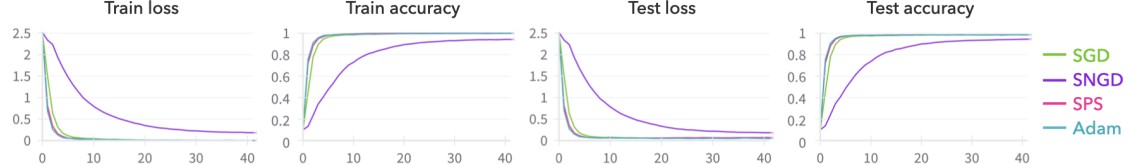

Figure 6: Comparison of train and test dynamics of various optimizers—SGD, SNGD, SPS, and Adam—for a ViT model on the MNIST dataset.

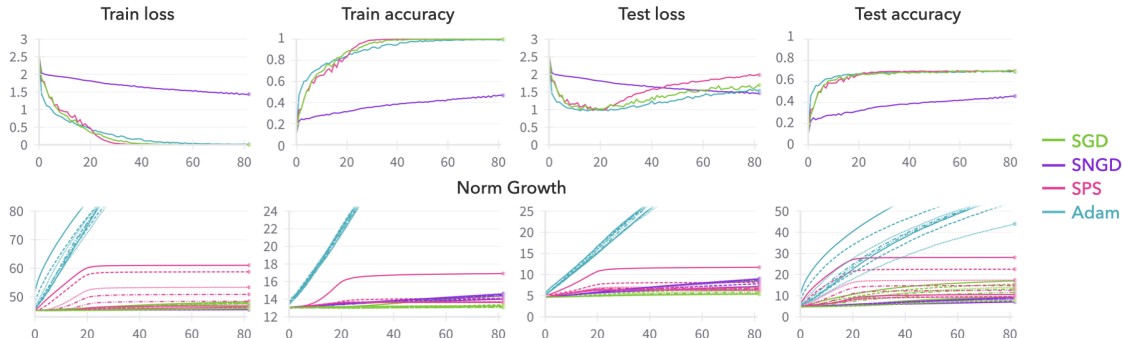

Figure 7: Comparison of train and test dynamics (top row) and parameter norm growth (bottom row) of various optimizers—SGD, SNGD, SPS, and Adam—for a ViT model on the CIFAR-10 dataset.

**Fig. 5** Training dynamics of a single-head self-attention model (Eq. (1)) when optimizing only $W$ on synthetic data (Example 1 with $\sigma = 0$). In this case, the `opt` tokens are antipodal instead of orthogonal. The remaining settings are the same as those for Fig. 2, except we set $\eta_{\max} = 5$ for Polyak-step.

**Fig. 6** We use the MNIST dataset (LeCun & Cortes, 2005), which contains gray-scale images of handwritten digit $0 - 9$. It is released under the CC BY-SA 3.0 license. We use the ViT for small-scale datasets proposed in (Lee et al., 2021), referred to as ViT-small hereon. The implementation is available at https://github.com/lucidrains/vit-pytorch under the MIT license. We use patch-size 4, and set depth as 2, number of heads as 8 and MLP width as 128. All models are trained with a batch-size of 100. Learning rates are set as follows. SGD: 0.1, SNGD: 0.001, Adam: 0.001. $\eta_{max}$ for SPS and SNGD is set to 0.01. We observe that unlike language datasets SNGD trains slower than SGD, and Adam achieves no significant gain

in terms of training speed over SGD (also reported in (Xie et al., 2023)), showcasing behaviour similar to CNNs (Kunstner et al., 2023).

**Fig. 7** We consider the CIFAR-10 dataset (Krizhevsky, 2009) which is a benchmark dataset for object recongnition tasks and contains colored images from 10 classes. It is released under the MIT license. We use the ViT-small model with patch-size 4, and set depth as 8, number of heads as 32 and MLP width as 512. All models are trained with batch-size 100. Learning rates are set as follows. SGD: 0.1, SNGD: 0.001, Adam: 0.001. The parameter $\eta_{max}$ is set to 10 for SPS and 0.1 for SNGD. As evident, SNGD portrays a slower training speed similar to the observations for MNIST. Additionally, we plot the norm growth for various layers of the ViT and see growth in parameters similar to our observations on synthetic datasets.

**Compute and Runtimes.** The experiments using synthetic data were run on Google Colab. The experiments on vision and language datasets were run on an internal cluster with two NVIDIA V100 GPUs with 32 GB memory each. We train for 40 epochs on MNIST and 80 epochs on CIFAR-10. The runtime for the latter is about 2 hours for each setting. We fine-tune the models on language datasets for 10 epochs, which takes about 32 hours for each run. In Figures 1 and 4, we plot the dynamics for every 200 iterations.

## C.1 Initializing in a Bad Stationary Direction

Fig. 8 shows a synthetic setting where we initialize in a bad stationary direction $\boldsymbol{W}_{init}$—one that gives a higher softmax score to the non-opt token. To be precise,

$$\boldsymbol{X}_1 = \begin{bmatrix} 1 & 0.2 \\ -1 & -0.2 \end{bmatrix}, \qquad \boldsymbol{X}_2 = \begin{bmatrix} -2.5 & 0.5 \\ 2.5 & -0.5 \end{bmatrix},$$

with labels $y_1 = -1, y_2 = 1$, respectively. We set $\boldsymbol{u}_* = [0, 1]^\top$. Using token optimality (Definition 1), this gives $\mathrm{opt}_1 = 2, \mathrm{opt}_2 = 1$; see Fig. 8 (left) for illustration. The chosen initialization $\boldsymbol{W}_{init}$ violates the (W-SVM) constraints for $i = 2$, that is

$$(\boldsymbol{x}_{2,\mathrm{opt}_2} - \boldsymbol{x}_{2,\tau \neq \mathrm{opt}_2})^\top \boldsymbol{W}_{init} \boldsymbol{x}_{2,1} < 0.$$

This can be seen in Fig. 8 (left), where the — line correlates more with the non-opt token than the opt one. This consequently forces the softmax score

$$\varphi^0_{2,\mathrm{opt}_2} = \frac{1}{1 + \exp((\boldsymbol{x}_{2,\tau \neq \mathrm{opt}_2} - \boldsymbol{x}_{2,\mathrm{opt}_2})^\top \boldsymbol{W}_{init} \boldsymbol{x}_{2,1})} < \frac{1}{2},$$

which can be seen in Fig. 8 (plot 2). We call $\boldsymbol{W}_{init}$ a "bad" stationary direction as starting with $\alpha \boldsymbol{W}_{init}$ and $\alpha \to \infty$ would result in $\varphi_{2,\mathrm{opt}_2}(\alpha \boldsymbol{W}_{init}) \to 0$, and $\nabla_{\boldsymbol{W}} \widehat{L}(\alpha \boldsymbol{W}_{init}) \to \boldsymbol{0}$. On the other hand, for $\boldsymbol{X}_1$, $\boldsymbol{W}_{init}$ behaves the opposite way (see plots 1-2 in Fig. 8).

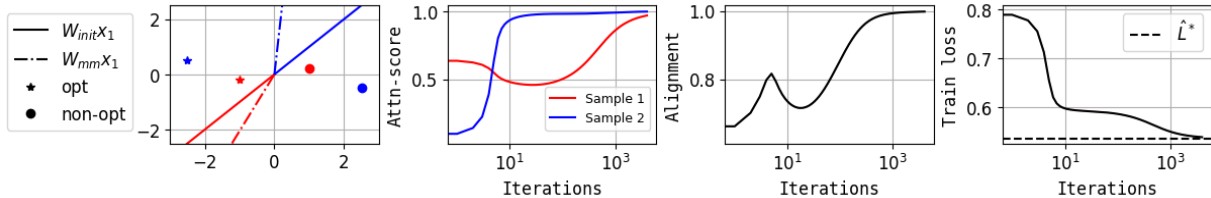

Figure 8: The training dynamics for a synthetic setting with $n = 2, d = 2, T = 2$. Attn-score denotes the opt-token softmax score $\varphi^t_{i,\mathrm{opt}_i}$, and alignment shows $\langle \overline{\boldsymbol{W}}_t, \overline{\boldsymbol{W}}_{\mathrm{mm}} \rangle$. Plot 1 shows the initialization $\boldsymbol{W}_{init}$, and the (W-SVM) solution $\boldsymbol{W}_{\mathrm{mm}}$ projected onto the first token, respectively. For sample 2, the $\boldsymbol{W}_{init}$ is more correlated to the non-opt token which consequently gives a small softmax score (plot 2). Plots 2-4 show that despite this "bad" initialization direction GD converges to $\boldsymbol{W}_{\mathrm{mm}}$ and achieves the loss minima $\widehat{L}^*$.

Despite this "bad" starting direction, plots 3-4 in Fig. 8 show that GD still globally convergences to $\boldsymbol{W}_{\mathrm{mm}}$, and minimizes the train loss. This behaviour aligns with our theoretical results (Thm. 1) showing global parametric convergence starting in any direction $\boldsymbol{W}_{init}$, with a small enough norm to avoid the bad stationary directions in the limit as $\|\boldsymbol{W}_{init}\| \to \infty$.

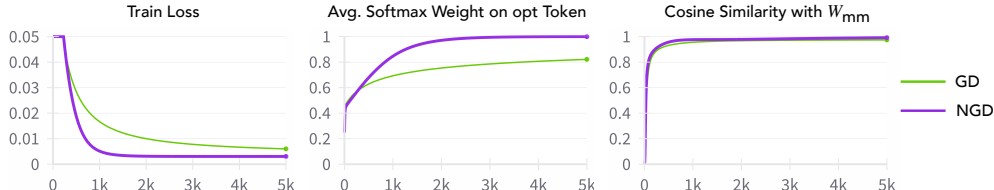

Figure 9: The training dynamics of self-attention parameters $\boldsymbol{W}_t$ for a binary classification on a simplified MNIST. For details on the task see Appendix C.2. Results show parametric convergence to $\boldsymbol{W}_{\mathrm{mm}}$, and fast convergence for NGD aligning with our theoretical findings.

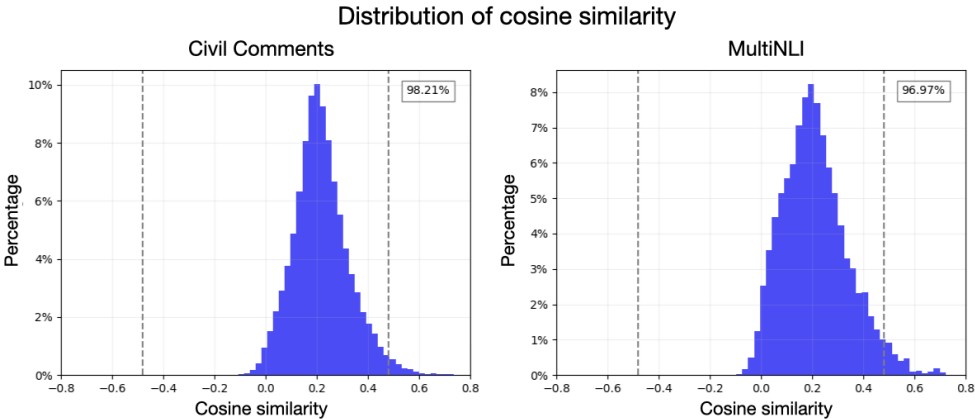

Figure 10: Validation of part 1 of Assumption 1. We compare cosine similarity of token embeddings for tokens within a sentence, and plot the distribution over multiple sentences. See details in Appendix C.3. We observe that a significant portion of the distribution lies around 0, indicating near orthogonality.

## C.2 One-layer Self-attention Model Trained on Binary MNIST

In this section, we conduct experiments using a simplified MNIST setting as follows: we created a binary classification task where each input consists of multiple patches, with one patch containing either a 0 or 1 digit (the 'signal' patch) and the remaining patches containing Gaussian noise. We trained a self-attention module that processes these patches before feeding them through a fixed linear classifier (pre-trained on 0/1 classification). As shown in Figure 9, the results in this setting empirically validate two key theoretical predictions: (1) the self-attention parameters $\boldsymbol{W}$ converge in direction to $\boldsymbol{W}_{\mathrm{mm}}$, and (2) normalized gradient descent achieves faster convergence than standard gradient descent. In this setup, the signal patch naturally corresponds to the token with the highest score, aligning with Ass. 2.

## C.3 Validation of Assumption 1

We conduct an experiment to verify the first part of Assumption 1 on two language-based datasets. To do this, we obtain token embeddings from a pre-trained BERT tokenizer (Devlin et al., 2019), and, for every sentence, we compute the cosine similarity between every pair of token embeddings. We consider 50k text examples from the Civil Comments dataset (Borkan et al., 2019) and plot the distribution of these similarity values. We repeat the experiment using 100k samples from the MultiNLI dataset (Williams et al., 2018). The results are shown in Figure 10. Values close to 0 indicate that the tokens are orthogonal.

We observe that a significant fraction of all the cosine similarities fall within a small range around 0, suggesting a nearly orthogonal behavior. Note that while Assumption 1 is only on the *similarity of non-optimal tokens with optimal or other non-optimal tokens*, whereas the experiment considers the similarity between every pair of token embeddings. Nevertheless, we see that the similarities are concentrated in a small region.

# D  Related Work

**Implicit bias of NNs.**  Since the first few works characterizing the implicit bias of linear predictors on separable data (see Introduction), there has been an abundance of work studying the implicit bias of gradient based methods for both linear predictors and NNs. Nacson et al. (2019); Ji & Telgarsky (2021); Ji et al. (2021) show fast convergence of GD-based methods to the max-margin predictor. For MLPs, early works study the implicit bias of GD/gradient flow using exponentially-tailed classification losses towards the KKT points of the corresponding max-margin problem in finite (Ji & Telgarsky, 2020; Lyu & Li, 2020) and infinite width (Chizat & Bach, 2020). Additionally, works by Phuong & Lampert (2021); Frei et al. (2022b); Kou et al. (2023b) study the implicit bias of GD trained ReLU/Leaky-ReLU networks on orthogonal data. Other works also study the implicit bias towards rank minimization with square loss in regression settings (Vardi & Shamir, 2021; Arora et al., 2019; Li et al., 2021). We encourage the reader to go through these works and a recent survey (Vardi, 2022) for a thorough understanding.

**Parametric convergence rates for NGD and PS in non-convex settings.** We compare the rates obtained in our results to the corresponding results for linear logistic regression (see Sec. 6). Contrary to these results, we prove the first parametric convergence for NGD and PS in non-convex settings. The work by Loizou et al. (2021) shows loss convergence results for PS. However, the rate of directional convergence might be different from the rate of loss convergence. This has been seen in the series of works analyzing gradient-based methods for linear logistic regression (Soudry et al., 2018; Nacson et al., 2019; Ji & Telgarsky, 2019). If the problem had a unique global minima, then loss convergence implies parameteric convergence. However, the problem of training self-attention does not have a unique minimum. For instance, using some other norm in place of $\ell_2$ norm in the objective of (W-SVM) would yield a different solution. While the norm diverges, the convergence is in the direction. In addition to this, there has been some work showing the benefit of adaptive step sizes such as NGD and PS for transformer optimization (Kunstner et al., 2023; Schaipp et al., 2024), which is connected to Q3. We next discuss related work on transformer theory.

**Transformers theory.**  Several studies, such as those by Baldi & Vershynin (2022); Dong et al. (2021); Yun et al. (2020a;b); Sanford et al. (2023); Bietti et al. (2023) have delved into exploring the expressivity of attention, while Baldi & Vershynin (2022); Dong et al. (2021); Yun et al. (2020a;b); Mahdavi et al. (2023) have initiated an investigation of its memory capacity. To gain insights into the optimization aspects of training attention models, Sahiner et al. (2022); Ergen et al. (2022) have explored convex relaxations of attention. Furthermore, a subdomain experiencing growing attention involves the theoretical exploration of in-context learning, as evidenced by recent studies (von Oswald et al., 2022; Akyürek et al., 2023; Zhang et al., 2023; Li et al., 2023c).

In this context, we discuss studies that seek to understand the optimization and generalization dynamics of transformers. Jelassi et al. (2022) show that Vision Transformers (ViTs) learn spatially localized patterns in a binary classification task using gradient-based methods. For a three-layer ViT starting from some structured initialization to mimic a pre-trained network, Li et al. (2023b) show sample complexity bounds to achieve zero generalization and show attention maps sparsify as SGD training proceeds. As a step towards understanding the training dynamics of the closely related prompt-attention, Oymak et al. (2023) study the initial trajectory of GD for one-layer attention. Further, there has also been work to obtain optimization and generalization guarantees of GD in multi-head attention models (Deora et al., 2023). Additionally, recent work by Tian et al. (2023a) attempts to understand SGD-dynamics for the task of next-token prediction for one-layer transformer with a linear decoder showing a similar sparsifying effect in the attention map. More recently, Tian et al. (2023b) extend this by analyzing the joint training dynamics of multi-layer transformer with an MLP. Other than this, there has been recent progress towards a theoretical understanding of next-token prediction dynamics under GD training (Li et al., 2024; Thrampoulidis, 2024). Furthermore, Makkuva et al. (2024) discussed the impact of the transformer architecture and distributional properties on the loss landscape when modeling the data as a Markov source, and Ildiz et al. (2024) linked the dynamics of self-attention to context-conditioned Markov chains.

