# OpenReview forum: "Implicit Bias and Fast Convergence Rates for Self-attention"
_TMLR — Accepted by TMLR_

### Review · Reviewer_jEuR · 2024-11-03

**Summary Of Contributions:**

This work advances the understanding of self-attention optimization by proving global convergence conditions and analyzing adaptive step-size strategies (normalized GD and Polyak step-size), demonstrating their finite-time convergence rates and attention map sparsification properties. The study shows these methods can accelerate parameter convergence compared to standard gradient descent in non-convex settings, while establishing stronger parallels between self-attention's implicit bias and linear logistic regression, despite self-attention's non-convex nature.

**Audience:**

Yes

**Claims And Evidence:**

Yes

**Requested Changes:**

1. Provide more discussions on the impact of different initialization schemes on the convergence properties, as this may affect practical implementation


2. Testing the validity of  Assumptions 1 and 2 in practical scenarios like in your experiments on language and vision datasets


3. Provide more discussions on the limitation of binary and single attention layer setting.

**Strengths And Weaknesses:**

Strengths:

1. Provides formal proofs for global convergence and finite-time convergence rates

2. Demonstrates concrete advantages of adaptive step-size strategies over standard gradient descent

3. Covers both theoretical foundations and some practical parts of optimization strategies


Weaknesses::


1. Analysis focuses only on binary classification with linear decoder, may not generalize to more complex tasks or multi-class settings


2. Studies single self-attention layer rather than full transformer architecture


3. Some theoretical results rely on specific conditions that may not hold in practical applications

---

> ### Author Response · Authors · 2025-01-28
> **Response to Reviewer jEuR**
>
> Thank you for your comments and suggestions.
>
>
> **W1-2, C3: Generalizing results to more complex tasks and architectures.**
>
> Please see the second and third point in the general response section.
>
> **W3, C2: Testing the validity of the data assumption**
>
> We thank the reviewer for the question on the data assumption, which has prompted us to explore this further. As suggested, we have conducted an experiment to verify the first part of Assumption 1 on two language-based datasets. In these experiments, we obtain token embeddings from a pre-trained BERT tokenizer, and, for every sentence, we compute the cosine similarity between every pair of token embeddings. We consider 50k text examples from the Civil Comments dataset and plot the distribution of these similarity values. We repeat the experiment using 100k samples from the MultiNLI dataset. The results are shown in Fig. 1 in the PDF. Values close to 0 indicate that the tokens are orthogonal.
>
> We observe that a significant fraction of all the cosine similarities fall within a small range around 0, suggesting a nearly orthogonal behavior. Note that our assumption is only on the similarity of non-optimal tokens with optimal or other non-optimal tokens, whereas the experiment considers the similarity between every pair of token embeddings. Nevertheless, we see that the similarities are concentrated in a small region.
>
> The figure can be found [here](https://anonymous.4open.science/api/repo/tmlr-exps-7364/file/tmlr-ib-exps.pdf?v=b44a42a1).
>
> **C1: Impact of different initialization schemes on the convergence properties.**
>
> Our theoretical results hold for any initialization as long as the initialization scale is small enough. As discussed below Ass. 1, a small initialization scale helps avoid saturated non-optimal tokens under “bad” stationary directions, i.e. directions $\overline{\bf{W}}$ such that $\nabla_{\bf{W}} \hat{L}(\alpha \overline{\bf{W}})\rightarrow \bf{0}$ and $\phi_{i,opt_i}(\alpha\overline{\bf{W}}) \rightarrow \bf{0}$ as $\alpha\rightarrow\infty$. This requirement is formalized in Lemma 2. Further, we have a synthetic experiment in App C.1 which shows the effect of starting in “bad” stationary directions. Specifically, we show that despite initializing along these directions, GD still globally converges directionally to ${\bf{W}}\_{mm}$, which aligns with our theoretical results.

---

### Review · Reviewer_zaLY · 2024-11-06

**Summary Of Contributions:**

This paper studies the convergence of gradient-based methods for self-attention modules. It shows that for normalized GD and Polyak step size GD, the normed iterates converge to the normed solution of a max-margin SVM problem. Empirical results show that these adaptive step sizes indeed achieve faster convergence than constant step sizes on synthetic data.


For transparency, I did not check the proofs of the submission in detail.

**Audience:**

Yes

**Broader Impact Concerns:**

None.

**Claims And Evidence:**

Yes

**Requested Changes:**

See weaknesses above, in particular the last bullet.

**Strengths And Weaknesses:**

*Strengths:*

* The paper nicely extends similar results for logistic regression, but here additional technical difficulties arise, most importanly a non-convex loss function.

* Adaptive step sizes are studied which are shown to converge faster (empirically) than constant step size GD.


*Weaknesses / Questions:*

* I could not understand the notation in Assumption 2. How is $\gamma_i$ defined? Can you describe this assumption informally?

* You assume that W-SVM is feasible. Is this assumption satisfiable under mild conditions on the data? Does it hold for the datasets you use in the experiments? For instance, matrices $W$ that have $x_{i,1}$ in their kernel are not feasible. For sufficiently many input examples which are linearly independent, would this imply that the solution has full rank?

* On page 12 you claim that this is the first iterate-convergence for Polyak stepsizes and NGD in non-convex settings. I am not sure whether this is correct: e.g. in Theorem 3.6 of Loizou, 2021 (on stochastic Polyak step sizes), objective function convergence is shown for the PL-condition (possibly non-covex). If one assumes a unique minimum, then this would imply also iterate convergence. Further, the statement is somewhat misleading, as your result does not proof that $W_t$ actually converges to a point (as I understood, the norm of $W_t$ diverges).

* I did not understand, why the W-SVM problem is using the first token always? Can you explain in more detail why this is chosen, and if/how this limits the results of the submission?

* On the experiments on real-world data: did you verify the convergence to the max-margin solution on real data? It seems that all the real-data plots in the paper only show train and test curves. I think it would be interesting (or even mandatory) to verify if the theoretical results hold on real data, especially given that the paper makes quite a lot of assumptions on the data model. The insight that adaptive step sizes help for training (on real data) is nice, but not particularly new, as it has been shown in multiple papers that Polyak-type step sizes and normalized GD can work well on transformer models (e.g. https://arxiv.org/abs/2305.07583, https://arxiv.org/abs/2304.13960).


*Minor:*

* The definition of $\bar W$ seems to be missing.

---

> ### Author Response · Authors · 2025-01-28
> **Response to Reviewer zaLY**
>
> Thank you for your careful reading and for your thoughtful comments and suggestions.
>
> **W1: Regarding Assumption 2.**
>
> Please see the first point in the general response section.
>
> **W2: Regarding the feasibility of (W-SVM).**
>
> Thank you for the question about W-SVM feasibility. You're right that we assume the feasibility of W-SVM. This assumption is analogous to linear separability assumptions in classical linear models, which are fundamental to understanding implicit bias in gradient-based methods. Our setting parallels the theory of implicit bias in overparameterized neural networks, where separability assumptions emerge naturally under sufficient overparameterization. For linear models, feasibility of hard-margin SVM is guaranteed when the ambient dimension exceeds the number of samples. Similarly, for our self-attention model, W-SVM feasibility is guaranteed when the ambient dimension exceeds both the number of samples and the context window size (as shown in Theorem 1 of Tarzanagh et al. (2023a)). We have explained this in the footnote on page 4 in the revision.
>
> **W3: Iterate convergence for adaptive step-sizes in non-convex settings.**
>
> The reviewer is correct that if the minima is unique, then results showing convergence of loss also imply convergence to the minima. However, we want to emphasize two points here. First, the rate of directional convergence might be different from the rate of loss convergence. This has been seen in the series of works analyzing gradient-based methods when training linear models on linearly separable data with exponentially-tailed loss functions [cite]. Second, the problem of training self-attention does not have a unique minimum. For instance, using some other norm in place of $\ell_2$ norm in the objective of (W-SVM) would yield a different solution. While the norm diverges, the convergence is in the direction. We hope these points address the reviewer’s concern about the claim. This is a subtle point worth elaborating further, which we have done in the revision.
>
>
> **W4: Using only the first token for (W-SVM).**
>
> As mentioned in the Preliminaries section footnote, using the first token is without loss of generality since we allow permutation of the tokens in the input, and the optimal token can be at any index. We could use any fixed token $\tau$ and the results would hold identically. In particular, what is important is that we do not fix the position of the optimal token (either by using different tokens for each sample or by enforcing a specific token ordering across samples), as this would actually make the problem significantly simpler by providing additional information. Thus, fixing the output token while allowing the optimal token to appear at any index is not limiting the contribution.
>
> **W5: Verifying convergence to the max-margin solution in real-world data.**
>
> Our theoretical results are applicable only for a single-layer self-attention model. Most real-world datasets require more complex architectures to get good performance and hence, we cannot verify these results for real-world datasets. That being said, we show the theory and empirics match closely for synthetic datasets which allows us to understand the training dynamics of self-attention. Please also see the second point in the general response section.
>
> Thanks for the reference, we missed that and have added it in the revised version.
>
> **W6: Definition of $\overline{\bf{W}}$.**
>
> We have added the definition in Section 2.

---

> > ### Comment · Reviewer_zaLY · 2025-02-03
> > **Response to authors**
> >
> > Dear authors,
> >
> > thank you for the clear and detailed response and the revised paper.
> >
> > I would add two follow-up questions/remarks:
> >
> > W1: Now that Assumption 2 is stated formally it is much easier to grasp it, thanks! However, it also shows how strong this assumption is: for all non-optimal tokens the score must be exactly equal (which is much stronger than e.g. uniqueness of the optimal token, correct?). Is there any hope that this assumption could hold in realistic scenarios?
> >
> > W5: I think the dataset and model need to be separated here: you could run your self-attention model on a real-world dataset instead of a synthetic. It would be interesting to see how realistic Assumptions 1 and 2 are, especially given how strong they appear (see remark above). Another interesting study would be to look at convergence if the assumptions *do not* hold: this could bring insight if the strength of the assumptions is inevitable or only necessary for the theoretical analysis.
> >
> > Some additions in these direction could fill in gaps in the connection theory-practice in my opinion.

---

> > > ### Author Response · Authors · 2025-02-15
> > >
> > > Thank you for your thoughtful comments. Regarding the assumption's strength: While we acknowledge this assumption may not hold in general, we hypothesize that similar global convergence behavior emerges under over-parameterization. Though we cannot formally prove this yet, our current proof represents the first demonstration of global convergence in this setting and we believe our simplified setting and transparent analysis have inherent value - they reveal fundamental mechanisms that might be obscured in more general formulations, potentially providing insights for understanding more complex scenarios.
> > >
> > > Following your suggestion about empirical validation, we conducted new experiments using a simplified MNIST setting as follows: we created a binary classification task where each input consists of multiple patches, with one patch containing either a 0 or 1 digit (the 'signal' patch) and the remaining patches containing Gaussian noise. We trained a self-attention module that processes these patches before feeding them through a fixed linear classifier (pre-trained on 0/1 classification). Our [results](https://anonymous.4open.science/api/repo/tmlr-exps-2-D99D/file/ib-sa-mnist.pdf?v=017f81ad) empirically validate two key theoretical predictions: (1) the self-attention parameters  $\mathbf{W}$ converge in direction to $\mathbf{W}\_{mm}$, and (2) normalized gradient descent achieves faster convergence than standard gradient descent. In this setup, the signal patch naturally corresponds to the token with the highest score, aligning with Assumption 2. We will include these experimental results in our revision.

---

> > > > ### Comment · Reviewer_zaLY · 2025-02-17
> > > > **Response to authors**
> > > >
> > > > These additional experiments further validate the theoretical findings, thank you for running them!

---

### Review · Reviewer_2WpF · 2024-12-23

**Summary Of Contributions:**

This work provides new theoretical results on convergence guarantees of single layered transformers. The results in the paper addresses limitations in prior work in this area that provided convergence guarantees that were (a) local, (b) asymptotic. In response to these limitations, this paper shows that (a) under some assumptions, global convergence is possible, (b) also provides finite convergence rate. Further, this paper shows that using normalized SGD and Polyak step-size, faster convergence can be obtained. This paper provides these results for two problem setting: (a) when the linear decoder after the attention layer is frozen, (b) when both the attention layer and the decoder layers are trained simultaneously. They also provide empirical evidence illustrating the convergence rates for various algorithms on synthetic as well as small vision and language datasets.

**Audience:**

Yes

**Broader Impact Concerns:**

N.A.

**Claims And Evidence:**

Yes

**Requested Changes:**

- Please address the points raised in the weaknesses.

**Strengths And Weaknesses:**

Strengths:
- The paper provides theoretical results in an important area: convergence rates of single layered transformers as well as their implicit bias to converge towards certain solutions
- The results addresses several limitations of prior works like asymptotic results and local convergence results, and also shows that adaptive step sizes in gradient descent can provide much faster convergence
- The paper also provides empirical results for convergence rates matching the intuitions obtained from the theoretical results


Weaknesses:
- I found several portions in the paper difficult to understand, especially, it seems that the author assumes that the reader is very familiar with prior works of Tarzanagh et al., which needs to be addressed. The writing should cover the background and problem setting better. E.g., “Our second technical assumption is similar to Tarzanagh et al. (2023a) with two key distinctions: Firstly, it applies to self-attention rather than their simplified attention model.” Please write in more detail what were the assumptions and results from previous work that you are improving upon. Also, in Assumption 2: what is “\gamma_i”? When using a notation, please refer to the definition where they are defined, its very difficult to find the definitions with so many notations.
- In eq 1, what is “x_1” and where is it defined? How does it fit into the attention equation? I am aware of self attention, but the way the equation is written confuses me. Can you please explain using dimensions of each matrix/vector and the outputs of the layers of the attention layer and the decoder layer.
- Similarly, in page 4, in eq (W-SVM), please provide more details about the equation, and how is it obtained (it is OK to provide details in the appendix, but the paper should have all the necessary preliminaries for a reader to understand all the details)
- How does the results translate to transformers with more layers? Are there any practical takeaways from this paper for training transformer models?

---

> ### Author Response · Authors · 2025-01-28
> **Response to Reviewer 2WpF**
>
> Thank you for your comments and suggestions.
>
> **W1: Assumptions and results from Tarzanagh et al. (2023a) that our work improves upon.**
>
> We believe we have covered the key problem formulation concepts from Tarzanagh et al. (2023a) necessary to understand our contributions. These concepts are described in detail in the *Preliminaries* section, including the setup, token score definition, optimal tokens, and the W-SVM problem. Our technical advances are clearly outlined in the *Introduction* and *Contributions* sections: i) demonstrating global directional convergence to the W-SVM solution, and ii) establishing finite-time rates for this global directional convergence. As thoroughly discussed, these results advance the local and asymptotic convergence results presented by Tarzanagh et al. (2023a).
>
> Regarding Assumption 2, it specifies that the margin over the non-optimal tokens is equal. Under the same assumption, the results of Tarzanagh et al. (2023a) are only local and asymptotic convergence results. Instead, we prove the first global convergence and finite-time rates for self-attention.
>
> We are happy to address any specific questions or further clarifications the reviewer might have.
>
> Thanks for pointing out the definition of $\gamma_i$, we have now added it in the preliminaries.
>
>
> **W2: Notation for the self-attention model.**
>
> As mentioned in the Contributions section, $\mathbf{x}_1$ is the first token of the input $\mathbf{X}$. We have also added this in the preliminaries section for clarification.
>
> We specify the dimensions of each element and how the model in Equation (1) is obtained in the paragraphs above and below Equation (1). If the reviewer has further specific questions about the variables involved or what steps are unclear, we would be happy to address them.
>
>
> **W3: More details about how (W-SVM) is obtained.**
>
> The intuition behind the constraints in (W-SVM) is as follows. Consider the loss for any sample:
>
> $\ell(y\mathbf{u}_*^\top\mathbf{X}^\top\phi(\mathbf{X}\mathbf{W}\mathbf{x}_1) =\ell(\mathbf{\gamma}^\top\phi(\mathbf{X}\mathbf{W}\mathbf{x}_1)) = \ell( \gamma\_{opt} \phi\_{opt} + \gamma(1- \phi\_{opt}))$.
>
> Since $\ell$ is a decreasing function, it is minimized when $\phi_{opt}=1$.
>
> Next, consider the softmax weight for the optimal token:
>
>
> $\frac{\exp({\bf{x}}\_{opt}^\top{\bf{W}}\_t{\bf{x}}_1)}{\sum\_{\tau} \exp({\bf{x}}\_{\tau}^\top{\bf{W}}_t{\bf{x}}_1)}=\frac{1}{1+\sum\_{\tau\neq opt}\exp(({\bf{x}}\_{\tau}-{\bf{x}}\_{opt})^\top{\bf{W}}_t{\bf{x}}_1)}$
>
>
> For loss minimization, this score should tend to 1 as the iterate norm $\|\|{\bf{W}}\_t\|\|$ tends to infinity. This can only happen when the exponent is negative, that is, $\mathbf{W}\_t$ satisfies $(\mathbf{x}\_{\tau}-\mathbf{x}\_{opt})^\top\mathbf{W}\_t\mathbf{x}\_1<0$. Then, analogous to translating the constraints to the SVM for training linear models on linearly separable data, we can obtain (W-SVM) for self-attention.
>
> We have added this discussion in Appendix A.1. Thanks for your question.
>
>
> **W4: Extension to more realistic architectures and practical takeaways.**
>
> Please see the second and third points in the general response section.

---

> > ### Comment · Reviewer_2WpF · 2025-03-03
> >
> > I thank the authors for providing detailed clarifications regarding the assumptions and results. Overall, I think the paper is in a good condition to be accepted.

---

### Author Response · Authors · 2025-01-28
**Global Response to Reviewers**

We sincerely thank the reviewers for their time and effort in reviewing our work, as well as for their positive feedback and helpful suggestions. We are encouraged to hear that the reviewers recognize our work for “providing theoretical results in an important area” (2WpF) by proving global convergence and finite-time convergence rates (jEuR), addressing “several limitations of prior works like asymptotic results and local convergence results” (2WpF). Reviewers (2WpF, zaLY, jEuR) also appreciated that our work demonstrates the superiority of adaptive step sizes by achieving faster rates for NGD and PS compared to GD, as well as the empirical results.

We respond to some common concerns below and address the other comments in the responses to each reviewer. We welcome additional queries the reviewers may have and would be happy to respond to them during the discussion period.

**1. Clarification about $\gamma_i$ and Ass. 2 (Reviewers 2WpF and zaLY)**:

We have added the explicit definition of $\gamma_i:= \min_{\tau \neq opt_i}\gamma_{i,\tau}$ in Section 3.1. Assumption 2 states that the margin for all non-optimal tokens for the $i^{th}$ sample is the same. Under the same assumption, previous results of Tarzanagh et al. (2023a) are only local and asymptotic convergence results. Instead, we prove the first global convergence and finite-time rates for implicit bias of gradient-based methods for self-attention.

**2. Regarding the experimental results (Reviewers 2WpF, zaLY, jEuR)**:

We would like to highlight that the primary focus of our work is to provide fundamental insights about the self-attention mechanism and its optimization properties. We support our theoretical results through experimental demonstrations which show that: a) in settings paralleling the theoretical setup, the observed rates closely match the analytical predictions, and, b) similar behaviors are observed on the language-based datasets MNLI (Fig. 1) and CivilComments (Fig. 4). For example, Fig. 2 shows that Polyak step-size and NGD significantly outperform SGD achieving fast rates as predicted by our analysis. That said, we refrain from making bold claims about things that are only observed empirically, acknowledging that they would need further investigation and may digress from the main message of the paper.


**3. How do the results translate to more complex settings? (Reviewers 2WpF and jEuR)**:

We agree with the reviewers that understanding the implicit bias of more complex transformer architectures is an important direction for future work. Our work takes a step in this direction by providing a precise characterization of the implicit bias and optimization dynamics of a single-layer self-attention model in both attention-only and joint optimization settings. While considering more complex architectures is indeed a worthwhile goal, it also significantly increases the difficulty of the problem.

Moreover, characterizing the implicit bias of overparameterized models is widely recognized as a challenging problem. The pioneering work by Soudry et al. (The implicit bias of gradient descent on separable data) focused on homogeneous linear models on separable data optimized with gradient descent. Subsequent research has explored the implicit bias of GD for deep linear networks and single-hidden-layer neural networks with LeakyReLU or ReLU activations. Although these settings are also far removed from the neural network architectures used in practice, they represent important steps toward developing similar theories for more complex models.
Similarly, to fully understand the implicit bias of the entire transformer architecture, it is crucial first to characterize the implicit bias of its core component—the self-attention mechanism. Despite the simplicity of our model and the assumptions about the data, the problem setting in our work remains non-convex. In this non-convex setting, we present the first results on global finite-time convergence and fast convergence rates.

We have updated the paper with the following modifications:

1. Added the definition of $\overline{ \bf {W}}$ and $\mathbf{x}_1$ in Section 2, and $\gamma_i$ in Section 3.1.

2. Added intuition about the constraints in (W-SVM) in Appendix A.1.

3. Added clarification on W-SVM feasibility in the footnote of page 4.

4. Added a discussion on the parametric convergence for NGD and PS, and relation to previous work in App. D.

---

### Decision · Action_Editor_2XT2 · 2025-03-03

**Recommendation:** Accept as is

**Comment:**

As all reviewers unanimously agreed on the great contributions of this paper, it should be accepted.

**Audience:**

Relevant to the vast majority of the machine learning community

**Claims And Evidence:**

This paper studies the implicit bias of self attention with accelerated optimization convergence by normalized GD and Polyak stepsize. In addition to the theoretical arguments, they are verified through numerical simulations.